# Ultraviolet radiation shapes dendritic cell leukaemia transformation in the skin

Gabriel K. Griffin[1,2,3 ✉], Christopher A. G. Booth[4], Katsuhiro Togami[4], Sun Sook Chung[4], Daniel Ssozi[2,5], Julia A. Verga[2,6], Juliette M. Bouyssou[4], Yoke Seng Lee[2,5], Vignesh Shanmugam[2,3], Jason L. Hornick[3], Nicole R. LeBoeuf[7], Elizabeth A. Morgan[3], Bradley E. Bernstein[2,6,8,9], Volker Hovestadt[2,10,11 ✉], Peter van Galen[2,5,9 ✉] & Andrew A. Lane[2,4,9 ✉]

Tumours most often arise from progression of precursor clones within a single anatomical niche. In the bone marrow, clonal progenitors can undergo malignant transformation to acute leukaemia, or differentiate into immune cells that contribute to disease pathology in peripheral tissues[1–4]. Outside the marrow, these clones are potentially exposed to a variety of tissue-specific mutational processes, although the consequences of this are unclear. Here we investigate the development of blastic plasmacytoid dendritic cell neoplasm (BPDCN)—an unusual form of acute leukaemia that often presents with malignant cells isolated to the skin[5]. Using tumour phylogenomics and single-cell transcriptomics with genotyping, we find that BPDCN arises from clonal (premalignant) haematopoietic precursors in the bone marrow. We observe that BPDCN skin tumours first develop at sun-exposed anatomical sites and are distinguished by clonally expanded mutations induced by ultraviolet (UV) radiation. A reconstruction of tumour phylogenies reveals that UV damage can precede the acquisition of alterations associated with malignant transformation, implicating sun exposure of plasmacytoid dendritic cells or committed precursors during BPDCN pathogenesis. Functionally, we find that loss-of-function mutations in *Tet2*, the most common premalignant alteration in BPDCN, confer resistance to UV-induced cell death in plasmacytoid, but not conventional, dendritic cells, suggesting a context-dependent tumour-suppressive role for TET2. These findings demonstrate how tissue-specific environmental exposures at distant anatomical sites can shape the evolution of premalignant clones to disseminated cancer.

Clonal expansions of cells containing somatic mutations are common in normal tissues, and often arise during ageing or in response to genotoxic stress[6–8]. Although most clones never progress, rare cells acquire additional alterations that confer a proliferative or survival advantage within the local tissue environment. In the bone marrow of ageing individuals, clonally expanded precursors with preleukaemic mutations give rise to a variety of differentiated immune cell populations that circulate throughout the body in the peripheral blood[1–3]. These cells can migrate into peripheral tissues and are increasingly recognized as mediators of organ-specific inflammation[9–11]. Whether tissue-specific mutational processes affect the clonal evolution of these immune cells is poorly understood. Here we used integrated genomic and single-cell analysis to examine progression to malignancy in BPDCN, an aggressive leukaemia that often presents as isolated skin tumours without clinically apparent blood or marrow involvement[12]. Thus, BPDCN presents a unique opportunity to study clonal evolution to cancer across anatomical sites, and to evaluate tissue-specific mutational processes in malignant transformation and progression.

## Phylogenomics of BPDCN across tissues

BPDCN is a unique acute leukaemia that often presents with malignant cells isolated to the skin[13,14] (Fig. 1a). Despite this localized presentation, most patients ultimately develop systemic disease[12,15]. Although these features suggest that BPDCN may arise in the skin, other findings favour a bone marrow origin. For example, BPDCN is frequently associated with underlying clonal haematopoiesis and can also present with concurrent bone marrow and blood involvement[12,14–16]. These observations raise fundamental questions about the anatomical origins, ontogeny and pathogenesis of BPDCN.

To define the relationship between clonal (premalignant) bone marrow and BPDCN skin tumours, we identified 16 patients with available

[1]Department of Pathology, Dana-Farber Cancer Institute, Boston, MA, USA. [2]Broad Institute of MIT and Harvard, Cambridge, MA, USA. [3]Department of Pathology, Brigham and Women's Hospital, Boston, MA, USA. [4]Department of Medical Oncology, Dana-Farber Cancer Institute, Boston, MA, USA. [5]Division of Hematology, Brigham and Women's Hospital, Boston, MA, USA. [6]Department of Cancer Biology, Dana-Farber Cancer Institute, Boston, MA, USA. [7]Department of Dermatology, Center for Cutaneous Oncology, Dana-Farber Cancer Institute and Brigham and Women's Hospital, Boston, MA, USA. [8]Department of Cell Biology, Harvard Medical School, Boston, MA, USA. [9]Ludwig Center at Harvard, Harvard Medical School, Boston, MA, USA. [10]Department of Pediatric Oncology, Dana-Farber Cancer Institute, Boston, MA, USA. [11]Division of Hematology/Oncology, Boston Children's Hospital, Boston, MA, USA. ✉e-mail: gabriel_griffin@dfci.harvard.edu; volker_hovestadt@dfci.harvard.edu; pvangalen@bwh.harvard.edu; andrew_lane@dfci.harvard.edu

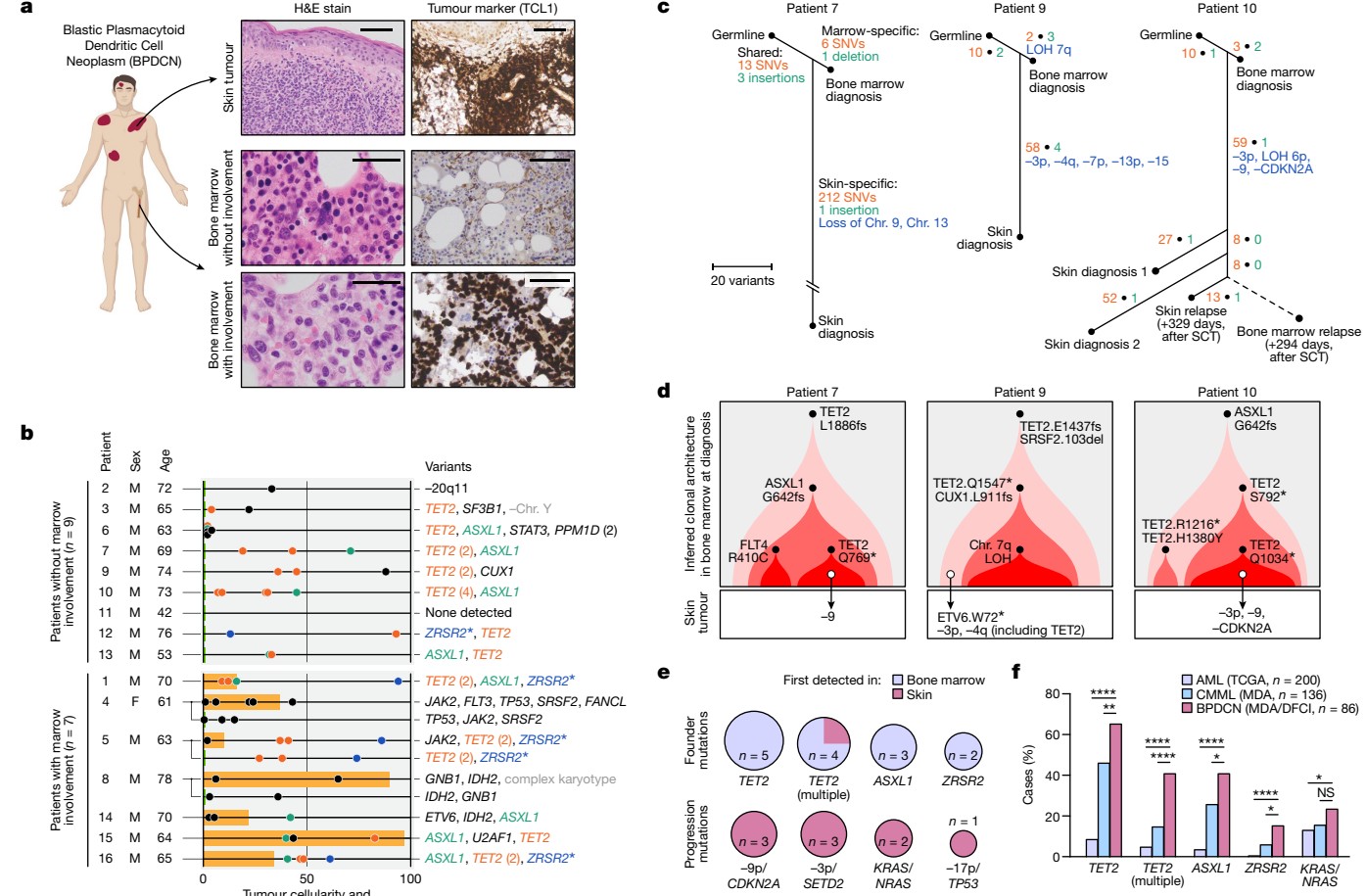

**Fig. 1 | Phylogenomics of BPDCN across tissues. a,** Skin tumour (top) from a representative patient with BPDCN showing infiltration by malignant cells (haematoxylin and eosin (H&E), left) expressing the pDC marker TCL1 (immunohistochemistry, right). Marrow can show normal haematopoiesis (middle) or involvement by malignant cells (bottom). Scale bars, 200 μm (top), 100 μm (middle right and bottom right), 50 μm (middle left and bottom left). **b,** Mutated genes identified by targeted sequencing of marrow samples from 16 patients with BPDCN, including nine without marrow involvement (top; limit of detection, 5%) and seven with involvement (bottom; the orange bars indicate tumour cellularity). Gene labels are ordered from left to right from low to high VAF (*x* axis). The parentheses indicate multiple mutations in one gene. Recurrent gene mutations (≥3 patients) are indicated in colour. The asterisks indicate genes on chromosome (Chr.) X. **c,** Tumour phylogenies for patients 7, 9 and 10. Single-nucleotide variants (SNVs) (red), insertions/deletions (green) and copy-number alterations (blue, minus symbol represents chromosome or

arm losses) are indicated (Methods). The dashed line indicates a post-transplant sample for which new alterations could not be detected owing to donor SNVs. SCT, stem cell transplant. **d,** Inferred clonal architecture in marrow samples for patients in **c**. Subclones directly related to BPDCN skin tumours are indicated by arrows, and are further annotated with skin-specific progression mutations. **e,** The frequency of mutations first detected in marrow (founder mutations, top) or skin tumours (progression mutations, bottom) for patients in **c** and Extended Data Fig. 1d,e. *n* values indicate the number of gene mutations across the five patients. **f,** The frequency of mutations in founder and progression genes in BPDCN, CMML and AML. Statistical significance was determined using two-sided Fisher's exact tests. DFCI, Dana-Farber Cancer Institute; MDA, MD Anderson Cancer Center; TCGA, The Cancer Genome Atlas. *P < 0.05, **P < 0.01, ****P < 0.0001; NS, not significant. The diagram in **a** was created using BioRender.

biopsy material for sequencing (Supplementary Table 1). This cohort comprised nine patients without bone marrow involvement and seven patients with concurrent skin and marrow involvement at diagnosis, including three who achieved remission after initial treatment (Fig. 1b). Skin tumours showed hallmark features of BPDCN, including purple nodules, plaques or bruise-like patches, and expression of canonical diagnostic markers (CD123, TCL1, CD4 and CD56; Extended Data Fig. 1a–c). Patients were predominantly male (15 out of 16, 94%) and over the age of 60 (14 out of 16, 88%), consistent with the known male bias and age association of BPDCN[13,14].

Targeted sequencing of 23 bone marrow samples from this cohort revealed pathogenic mutations in 15 out of 16 (94%) patients (Fig. 1b and Supplementary Table 2a). The mutations affected genes that are recurrently altered in clonal haematopoiesis, myeloid leukaemia and BPDCN, including *TET2* (11 out of 16 patients, 69%), *ASXL1* (9 out of 16 patients,

56%) and RNA splicing factors (9 out of 16 patients, 56%)[1–3,12,17,18]. Mutations in these genes were identified at a high variant allele fraction (VAF) in 8 out of 9 (89%) patients without marrow involvement, and in 3 out of 3 (100%) patients at the time of marrow remission, suggesting their presence in clonally expanded progenitors. There was no significant difference in VAF for samples with and without marrow involvement (34.9% and 29.2% for *TET2*, respectively, P = 0.635 by two-sided Student's *t*-test; 33.9% and 28.7% for *ASXL1*, respectively, P = 0.672).

We next defined the relationship between expanded premalignant haematopoietic clones in the bone marrow and BPDCN tumours in the skin at diagnosis using whole-exome (WES; Fig. 1c and Supplementary Table 2b–d), whole-genome (WGS; Extended Data Fig. 1d and Supplementary Table 2e) and targeted (Extended Data Fig. 1e) sequencing. This revealed a direct relationship between premalignant haematopoietic clones in the bone marrow and BPDCN skin

tumours in 9 out of 9 cases (100%) with matched sequencing. Reconstruction of tumour phylogenies showed a modest number of clonally expanded somatic mutations in bone marrow (WES range, 16–23; WGS range, 161–416) and a higher number in BPDCN skin tumours (WES range, 74–229; WGS range, 759–2,798). Most bone marrow mutations were found in matched skin tumours (WES range, 68.8–70.6%), while the majority of BPDCN skin tumour mutations were unique to the skin (range, 83.8–93.1%; Fig. 1c). Likewise, copy-number alterations, such as loss of CDKN2A, were detected in skin tumours but not matched bone marrow samples (Fig. 1c and Extended Data Figs. 1d,f, 2, 3a and 4a–c). These findings identify bone marrow as the source of premalignant clones in BPDCN, and indicate mutational evolution at multiple tissue sites during malignant transformation.

To examine clonal evolution over time, we extended our tumour phylogenies by sequencing six relapse samples from three patients (patients 1, 10 and 12; Fig. 1c and Extended Data Figs. 1d, 3b and 4d,e). Each of these patients experienced an initial response to systemic therapy that was followed (months later) by relapse in the skin and involvement of bone marrow and blood. Most mutations present in skin tumours from patient 10 at diagnosis were also detected in relapse skin tumour and bone marrow samples collected 1 year later (59.8–71.7%; Fig. 1c). Moreover, relapse samples accumulated additional alterations that were not detected at diagnosis. For example, out of 6,071 mutations detected in relapse samples from patient 1 (collected 2 years after diagnosis), 46.1% were shared with the initial skin tumour, while 30.4% were specific to the latest relapse sample (Extended Data Fig. 1d). This indicates that malignant cells present at diagnosis persist through initial therapy to initiate disease relapse.

### Clonal architecture and drivers of BPDCN

We next sought to define the clonal architecture and molecular drivers of BPDCN from precursor cells in the marrow to malignant tumours in the skin. We used VAFs to define founder clones and progression mutations in five patients (patients 1, 7, 9, 10 and 12) profiled by WES or WGS.

In contrast to clonal haematopoiesis of indeterminate potential[1–3], we observed complex clonal hierarchies in marrow samples from these patients. For example, marrow from patient 10 showed a truncal mutation in ASXL1 and two subclones defined by distinct pairs of TET2 mutations (subclone 1: S792* and Q1034*, subclone 2: R1216* and H1380Y; asterisk indicates gain of stop codon; Fig. 1d). Only mutations from subclone 1 were detected in skin tumours from patient 10, while subclone 2 remained isolated to the marrow. Likewise, marrow from patient 9 contained a single dominant clone with truncal TET2 (E1437fs; frameshift) and SRSF2 (P95–R102del; deletion) mutations, and subclonal events in TET2 (Q1547fs), CUX1 (L911fs) and 7q (CN-LOH; copy-number neutral loss of heterozygosity; Fig. 1d). Only the truncal TET2 and SRSF2 mutations were identified in the matched skin tumour, suggesting derivation from an ancestral clone that diverged before alterations that remained isolated to the marrow. Similar patterns of clonal evolution were observed in patients 1, 7 and 12 (Fig. 1d and Extended Data Fig. 1g). Overall, founder clones contained recurrent alterations in TET2 (5 out of 5 cases, 4 out of 5 cases with ≥2 alterations), ASXL1 (3 out of 5 cases) and splicing-factors (3 out of 5 cases), including ZRSR2 (2 out of 5 cases), suggesting privileged roles in premalignant evolution in BPDCN (Fig. 1e (top)).

We next examined progression events associated with transformation of founder clones in the marrow to BPDCN tumours in the skin. Activating RAS mutations were detected in skin tumours in 2 out of 5 patients (patients 1 and 12) but not founder clones in the marrow (Fig. 1e (bottom)). Likewise, copy-number loss of tumour suppressor regions was detected in skin tumours but not the marrow, including CDKN2A/chromosome 9p (loss of 9p harbouring the tumor suppressor CDKN2A) in 3 out of 5 patients (patients 1, 7 and 10), SETD2/chromosome 3p in 3 out of 5 patients (patients 1, 9 and 10) and TP53/chromosome 17p in 1

out of 5 patients (patient 1; Fig. 1e (bottom)). This supports key roles for these alterations in BPDCN transformation.

Finally, we compared the frequency of founder and progression-type mutations in two published BPDCN cohorts[12,17] to other myeloid neoplasms with similar clinicopathologic features, including chronic myelomonocytic leukaemia (CMML) and acute myeloid leukaemia (AML)[19–22]. Mutations in founder (TET2, ASXL1 and ZRSR2) and progression (NRAS; KRAS; loss of chromosome 9 or 3p harbouring CDKN2A or SETD2, respectively) genes were all more frequent in BPDCN than CMML and/or AML, supporting privileged roles in BPDCN pathogenesis (Fig. 1f and Extended Data Fig. 1h). Notably, TET2 alterations are present in nearly 70% of BPDCN tumours, including probable biallelic mutations in the majority of cases[12,17]. Moreover, approximately 10% of cases of BPDCN have IDH2 hotspot mutations, which typically occur in cases with wild-type TET2 and cause TET2 inhibition through the 2-HG oncometabolite[12,17,23–25]. This supports functional inactivation of TET2 in 70–80% of cases of BPDCN—to our knowledge, among the highest frequency of any cancer. This supports a unique role for TET2 inactivation during BPDCN development.

### Founder mutations across haematopoiesis

Our phylogenomic analysis indicated that BPDCN originates from clonally expanded progenitors in the bone marrow, consistent with recent findings[12,15,16]. To test this directly, we applied integrated single-cell RNA-sequencing (scRNA-seq) and genotyping analysis. We first generated a reference dataset from marrows of six healthy donors. This yielded 20,411 high-quality transcriptomes and represented the full spectrum of haematopoietic progenitor cells to differentiated myeloid, erythroid and lymphoid lineages (Fig. 2a and Supplementary Table 3). We next profiled marrow samples from five patients with BPDCN without marrow involvement (disease only in the skin) and six patients with involvement (disease in skin and the marrow; Fig. 2b). This yielded 66,600 high-quality transcriptomes.

To annotate cell types, we generated a single-cell classifier based on healthy donors using the random-forest machine learning algorithm[26] (Extended Data Fig. 5a). Predicted cell-type proportions in samples without BPDCN involvement were similar to healthy donors, and visualization of marker genes along the myeloid and erythroid differentiation hierarchies did not reveal major abnormalities, consistent with clinical evaluation (Fig. 2b and Extended Data Fig. 5b–d). In samples with known marrow involvement, cells classified as plasmacytoid dendritic cells (pDCs) were highly over-represented (5.3–90.5%, P = 0.016, Student's t-test), and application of a published BPDCN gene signature confirmed their identity as malignant BPDCN cells[27] (Fig. 2b and Extended Data Fig. 5c–h). This supports our single-cell approach for the detection of normal and malignant haematopoietic populations from BPDCN marrows.

We next sought to map mutations identified by phylogenomics onto haematopoietic differentiation hierarchies. We prioritized driver mutations and those with high expression by scRNA-seq (eXpressed variant sequencing, XV-seq), reasoning that these would be more efficiently genotyped. In total, we enriched 40 mutations from 11 single-cell libraries using our XV-seq approach and achieved efficient detection for many targets (for example, RAB9A, CDKN2A, and RPS24 in 37%, 20% and 99% of cells, respectively; Fig. 2c, Methods, Extended Data Fig. 6a–e and Supplementary Table 4). The XV-seq findings were in agreement with tumour phylogenies inferred from orthogonal assays, and donor/host annotations in samples after stem-cell transplantation (Extended Data Fig. 6f–h).

To evaluate the differentiation propensities of premalignant clones, we investigated 16 founder mutations in uninvolved marrows from 5 patients and detected a total of 10,245 wild-type and 1,204 mutated cells (Fig. 2d). Mutations were identified across progenitor, erythroid/myeloid and lymphocyte populations, with TET2 mutations showing

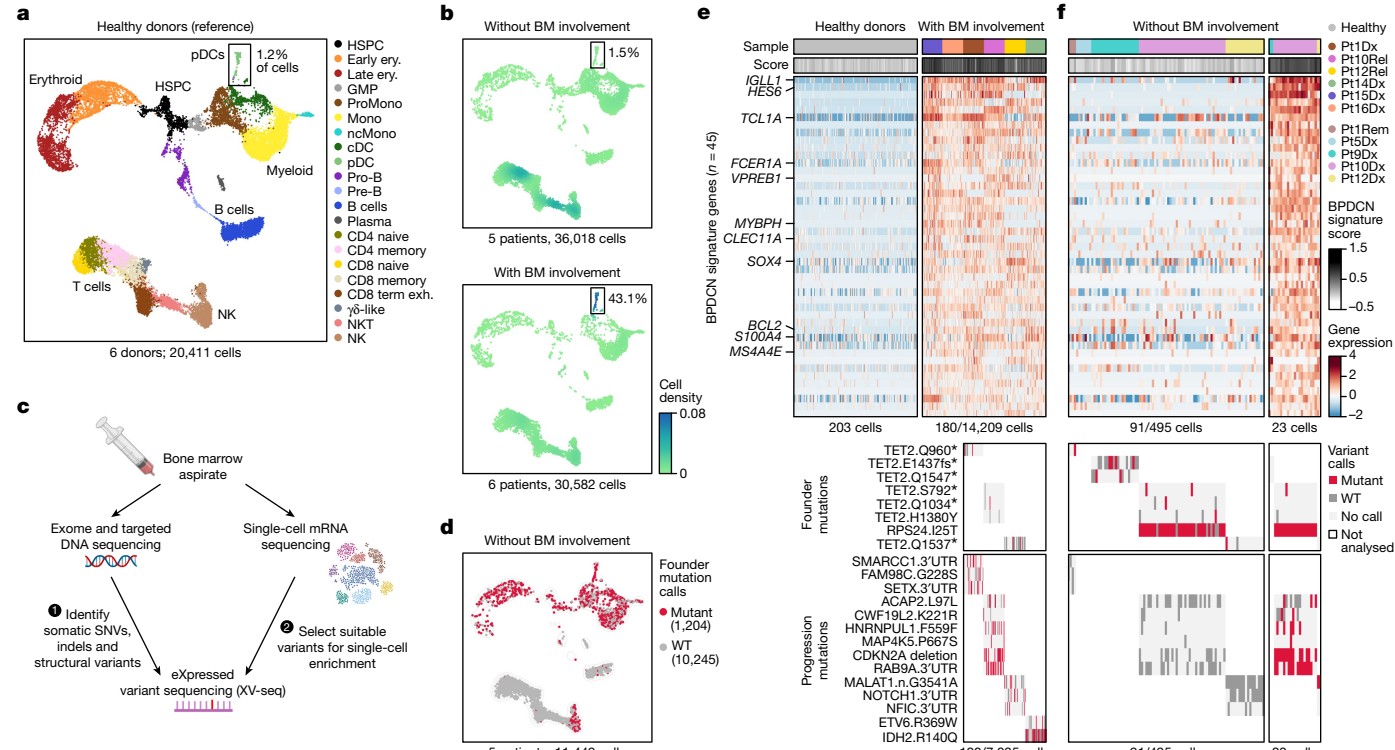

**Fig. 2 | Single-cell profiling resolves premalignant pDCs and BPDCN.**
**a**, Uniform manifold approximation and projection (UMAP) representation of the scRNA-seq analysis of marrow (*n* = 20,411 cells) from six healthy donors. Clusters show expected cell types, including progenitor, myeloid, erythroid and lymphocyte-lineage cells and pDCs (box). HSPC, hematopoietic stem and progenitor cell; ery, erythroid; GMP, granulocyte macrophage progenitor; ProMono, promonocyte; Mono, monocyte; ncMono, non-classical monocyte; cDC, conventional dendritic cell; pDC, plasmacytoid dendritic cell; Pro-B, pro-B cell; Pre-B, pre-B cell; CD8 term exh, CD8 terminally exhausted; NKT, natural killer T cell; NK, natural killer cell. **b**, UMAP representation of the density of cells from marrow samples of patients with BPDCN (*n* = 11) projected by transcriptional similarity to the cell types defined in **a** and coloured by two-dimensional kernel density estimation. The samples include those without known involvement by BPDCN cells (top; *n* = 5 patients, *n* = 36,018 cells) and those with involvement (bottom; *n* = 6 patients, *n* = 30,582 cells). BM, bone marrow. **c**, The XV-seq procedure. Mutations identified by DNA sequencing were selected for enrichment on the basis of their detection in matched scRNA-seq data. **d**, UMAP representation of the XV-seq results for enrichment of 16 founder

mutations from 5 patients with BPDCN without known marrow involvement. Cells are projected onto the clusters defined in **a** and coloured according to whether mutant (red; *n* = 1,204 cells) or wild-type (grey; *n* = 10,245 cells) transcripts were detected. Cells without mutant or wild-type calls are not shown. **e**, The expression of BPDCN signature genes (rows; *n* = 45) in cells classified as pDCs from six healthy donors (left; *n* = 203 cells) and six marrow samples from patients with BPDCN involvement (right; *n* = 14,209) (top). Malignant cells are downsampled to 30 cells per sample with genotyping information. The annotation bars (top) indicate the sample identifiers and BPDCN signature scores. Bottom, founder and progression mutations detected by XV-seq. **f**, Expression of BPDCN signature genes (top) and XV-seq mutations (bottom) in cells classified as pDCs from patients without known marrow involvement, as in **e**. Premalignant pDCs (left; *n* = 91 cells with genotyping information) show low signature scores and founder mutations exclusively. Occult BPDCN cells (right; *n* = 23) show high signature scores and a mix of founder and progression mutations. Pt, patient; Dx, diagnosis; Rem, remission; Rel, relapse; WT, wild-type. The diagram in **c** was created using BioRender.

erythroid/myeloid bias in 4 out of 5 cases (*P* < 0.05; Fig. 2d and Extended Data Fig. 6i). This confirms the origin of founder mutations in multi-potent progenitors rather than lineage-committed pDCs.

## Resolving premalignant pDCs and BPDCN

Our analyses suggest a model in which clonal (premalignant) haematopoietic precursors give rise to a spectrum of differentiating blood cell populations, including pDCs. These pDC-committed cells then acquire additional mutations during malignant transformation to BPDCN. To test this, we analysed founder and progression mutations in pDCs from healthy donors and patient marrow samples with and without known involvement by BPDCN.

First, we analysed cells classified as pDCs on the basis of our random-forest approach (203 from healthy donors, 518 from patient samples without marrow involvement and 13,709 from patient samples with marrow involvement). pDCs from healthy donors and patients without marrow involvement clustered together, whereas putative malignant BPDCN cells from patients with marrow involvement clustered

separately (Extended Data Fig. 7a). We next generated a BPDCN transcriptional signature by comparing BPDCN cells from cases with marrow involvement to pDCs from healthy donors and patients without marrow involvement (Extended Data Fig. 7b and Supplementary Table 3c). This signature comprised 45 upregulated genes, including *BCL2*, *IGLL1*, *TCL1A* and *HES6*, and showed strong correlation with a published BPDCN signature[27] (Pearson's *r* = 0.89; Extended Data Fig. 7c).

Next, we used single-cell transcriptional and mutation analysis to characterize the stepwise changes leading to pDC transformation. Samples with marrow involvement contained pDCs with high BPDCN signature scores and both founder and progression mutations, consistent with classification as malignant BPDCN cells (Fig. 2e and Extended Data Fig. 7d–h). By contrast, samples without known marrow involvement contained pDCs with low BPDCN signature scores and founder mutations only, probably representing premalignant pDCs or pDC-like cells (*n* = 495 cells; Fig. 2f (left)). Compared with healthy pDCs, premalignant pDCs adopted some but not all of the changes in gene expression observed in BPDCN cells, including upregulation of the *TCL1A* oncogene and downregulation of the *CXCR3* and *CXCR4*

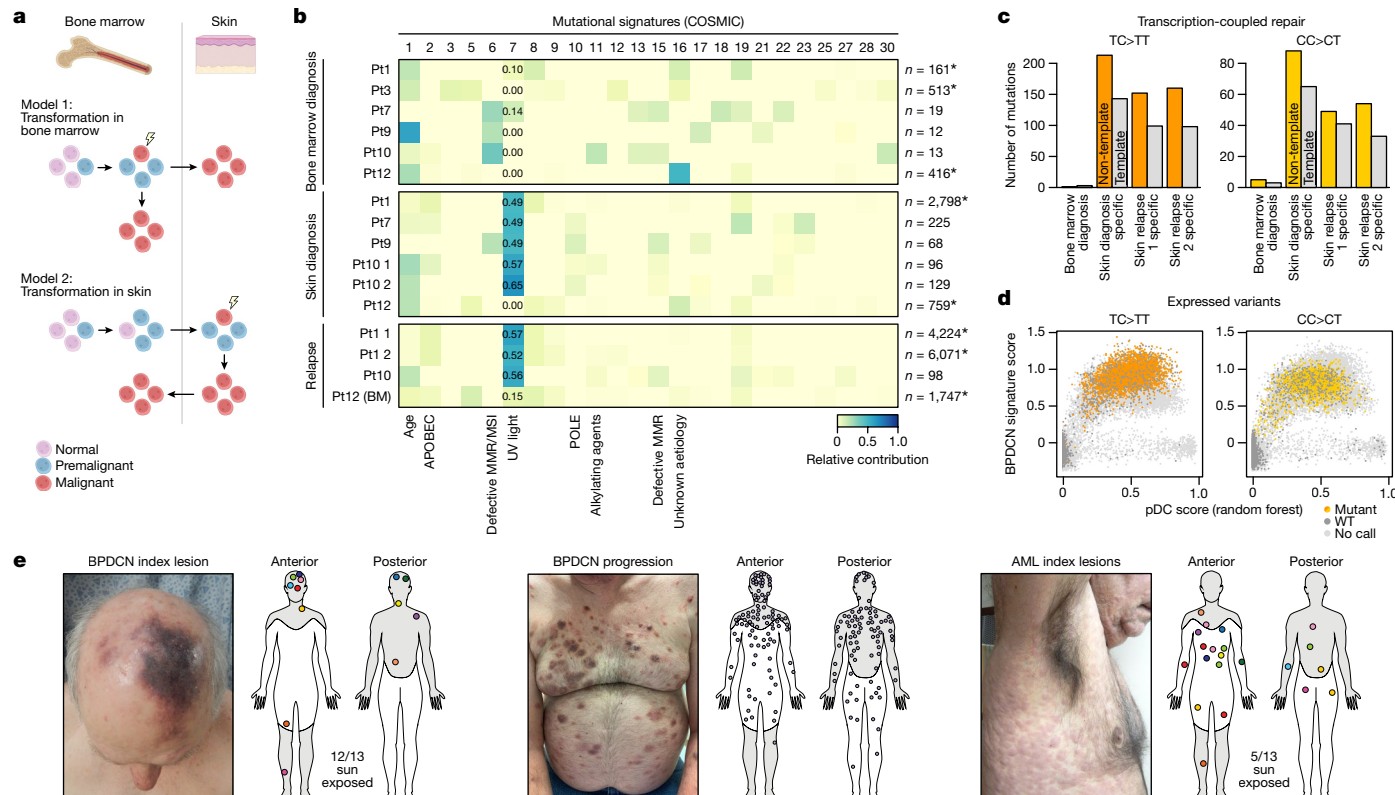

**Fig. 3 | UV damage localizes malignant progression of BPDCN to the skin. a**, Two models of clonal progression to malignancy in BPDCN. In model 1 (top), clonal precursors (blue) transform to malignant cells (red) in the bone marrow, and then spread to the skin. In model 2 (bottom), clonal precursors from the marrow (blue) seed the skin, transform to malignant cells (red) and then spread in a retrograde manner back to bone marrow. **b**, Mutational signature analysis of uninvolved patient marrows (top), BPDCN skin tumours at diagnosis (middle) and relapse samples (bottom; all skin tumours except for patient 12 (Pt12), which is marrow). Blue heat indicates the relative contribution of each signature. The total SNVs per sample is indicated on the right. *n* values marked by asterisks indicate samples profiled by WGS; all others represent WES. **c**, UV-associated

TC>TT and CC>CT mutations in samples from patient 1, separated according to their presence on the template (transcribed, indicated in grey) or non-template (non-transcribed, indicated in colour) strands of annotated genes. **d**, Single cells (*n* = 66,600) from all marrow samples, showing the random-forest pDC prediction score (*x* axis) and the BPDCN signature score (*y* axis). The colours indicate cells with UV-associated progression mutations (TC>TT, orange; CC>CT, yellow) detected by XV-seq. **e**, The anatomical distribution of BPDCN skin tumours at the time of diagnosis (left) and disease progression (middle). Skin lesions in patients with AML at diagnosis (right). Grey shading indicates areas of chronic or intermittent UV exposure. Representative clinical photos are shown. The diagram in **a** was created using BioRender.

chemokine receptors involved in homing to bone marrow and other tissues (Extended Data Fig. 8a,b).

Notably, we also identified rare pDCs with high BPDCN signature scores (*n* = 23) in samples without known marrow involvement, including 2 out of 4,593 cells (0.04%) from patient 9; 19 out of 10,106 cells (0.19%) from patient 10; and 2 out of 6,862 (0.03%) from patient 12 (Fig. 2f (right) and Extended Data Fig. 7d). Although the clinical significance of these rare cells requires further investigation, XV-seq revealed progression mutations in 19 out of 23 (82.3%) pDCs with high scores but in only 1 out of 495 other pDCs, verifying their malignant identity (*P* = 1.1 × 10⁻⁸⁴; Extended Data Fig. 8c–e). Thus, integrated expression and genotyping analysis resolves premalignant pDCs and BPDCN cells, including identification of rare malignant cells not detected by routine histopathology.

## UV radiation mutagenesis in BPDCN

The presence of rare malignant cells in bone marrow that were occult to clinical evaluation prompted us to consider two models of BPDCN transformation: (1) transformation of premalignant cells in the marrow followed by spread to the skin; or (2) homing of premalignant cells to the skin, followed by transformation and spread back to bone marrow (retrograde dissemination; Fig. 3a).

We reasoned that mutational signature analysis of founder and progression mutations could define the anatomical site(s) at which transformation probably occurred. We therefore evaluated our genomic sequencing data (WES and WGS) for 30 mutational signatures defined in the Catalogue of Somatic Mutations in Cancer (COSMIC). We found a marked enrichment for UV-radiation-associated signatures (signature 7) in BPDCN tumours from 4 out of 5 patients (relative contribution, 0.49–0.65) but not in matched (uninvolved) marrow samples (relative contribution, 0–0.14, *P* < 0.005, Wilcoxon rank-sum test; Fig. 3b). We also detected UV signatures in 10 out of 21 (47.6%) samples from three additional BPDCN cohorts[16,17,28] (using a stringent threshold of 0.2; Extended Data Fig. 9a).

We next examined the nucleotide changes that drive these UV signatures. UV-induced DNA damage is characterized by C>T transitions in dipyrimidine contexts, including UV-associated TC>TT and CC>CT mutations, and UV-specific CC>TT dinucleotide mutations[29]. UV-associated mutations were enriched in BPDCN skin tumours (43.1–55.6% TC>TT, 13.3–37.7% CC>CT; *n* = 5 from patients 1, 7, 9 and 10) but not in matched (uninvolved) marrow samples (5.3–15.4% TC>TT, *P* < 0.001, Wilcoxon rank-sum test; Extended Data Fig. 9b,c). Moreover, UV-specific CC>TT dinucleotide mutations were exclusively detected in BPDCN tumour samples but not uninvolved marrows (Extended Data Fig. 9d). Finally, UV-associated TC>TT and CC>CT mutations

were enriched on the non-template strand of coding genes (60.7% and 57.9%, P < 0.005, binomial test; Fig. 3c). This probably reflects transcription-coupled repair of mutations on the template strand—another hallmark of UV damage[30,31].

In previous studies, UV mutations in sun-exposed skin from healthy donors have been identified at low a VAF, reflecting background damage to skin cells that do not show clonal expansion beyond minute foci[6]. By contrast, UV mutations in BPDCN were detected at a high VAF, suggesting their presence as clonally expanded alterations within malignant cells. To validate this, we revisited our scRNA-seq with genotyping data (XV-seq) and assessed the distribution of UV mutations. In total, we identified 4,263 cells in the bone marrow with UV mutations. Among these, 4,100 (96.2%) were classified as malignant BPDCN cells, including the population of rare (occult) BPDCN cells (n = 19) from patient 10 described above (Fig. 3d and Extended Data Fig. 9e). The presence of UV-associated mutations in BPDCN cells in the marrow is consistent with their previous transit through the skin.

On the basis of these findings, we further hypothesized that BPDCN tumours at the time of initial clinical presentation would localize to sun-exposed rather than sun-protected skin. Indeed, index skin tumours from our cohort were typically solitary lesions at sun-exposed sites, including head and neck, upper back and central chest (n = 13; Fig. 3e (left)). By contrast, skin lesions from patients with BPDCN at disease progression (Fig. 3e (middle)) or AML patients at diagnosis tended to be disseminated and involve both sun-exposed and sun-protected sites (Fig. 3e (right); 12 out of 13 initial BPDCN versus 5 out of 13 AML skin lesions in sun-exposed sites; P = 0.0112, Fisher's exact test). Together, these findings indicate that UV exposure is a recurrent feature of BPDCN.

## UV radiation damage before BPDCN transformation

We next examined the temporal order of UV damage during BPDCN development. Specifically, we examined whether UV exposure begins before or after the occurrence of progression mutations associated with malignant transformation. Accordingly, we overlaid UV mutations onto a detailed tumour phylogeny for patient 10, which was constructed from multiple samples and clinical timepoints. Skin tumours taken from different anatomical sites at diagnosis contained two distinct malignant clones that arose from a common TET2-mutated precursor in the marrow (Figs. 1c and 4a,b). Progression mutations in these clones included parallel CDKN2A deletions with unique breakpoints (Fig. 4c), and loss of different chromosome 3p alleles in regions containing the tumour suppressor SETD2 (Fig. 4d and Extended Data Fig. 10a), indicating convergent evolution during malignant transformation. The majority of UV-associated TC>TT mutations (32 out of 60 alterations, 53.3%) were shared between these two malignant clones (Figs. 3c and 4a,b). This indicates that UV damage began before the loss of CDKN2A and SETD2.

To further validate this finding, we compared the VAFs of founder and UV-associated progression mutations in patient 10. In the bone marrow, founder mutations (n = 16) showed a wide range of VAFs (coefficient of variation, 0.38), consistent with their gradual acquisition over time (Fig. 4b). By contrast, mutations common to marrow and both skin samples showed a narrow distribution of VAFs (0.13–0.15), consistent with their acquisition in a clonal precursor before transformation. Similar ranges of VAFs were observed for founder mutations in bone marrow (0.31–0.46) and skin tumour samples (0.09–0.10) from patients 7 and 9 (Extended Data Fig. 10b).

Together, these data indicate that UV damage can precede malignant transformation, and further nominate a clonal (premalignant) pDC or pDC-like cell in the skin as the precursor of BPDCN in many cases (Fig. 3a (model 2)). Moreover, our data indicate that transformed BPDCN cells in the skin can then spread to other anatomical sites, including through retrograde dissemination to previously uninvolved bone marrow.

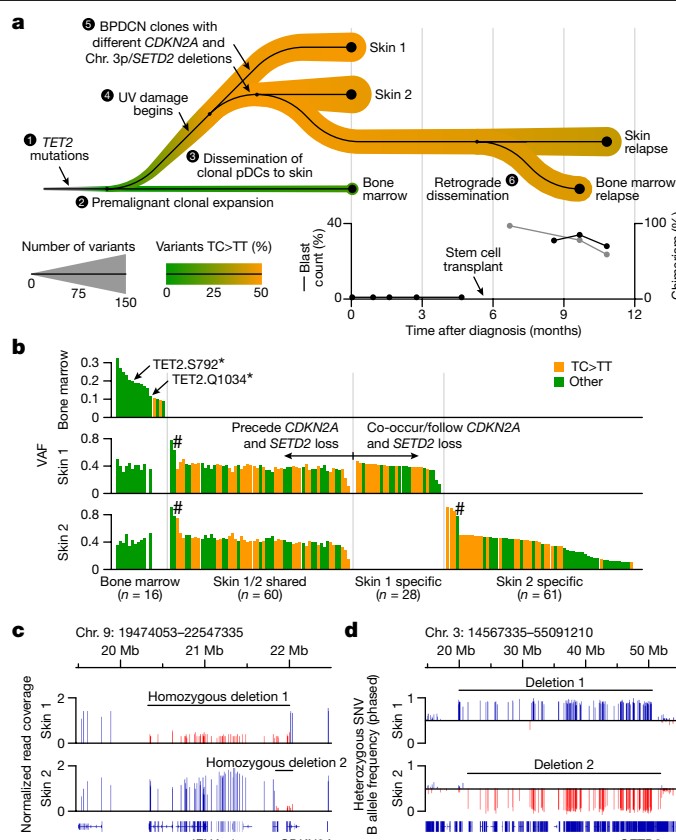

**Fig. 4 | UV damage begins before malignant transformation. a**, Subway plot showing clonal dynamics, clinical features and the disease course of patient 10. Samples profiled by WES (n = 5) are indicated by black dots and connected according to phylogenomic relationships. The line width indicates the total number of detected mutations in each sample, and the colour indicates the percentage of UV-associated TC>TT mutations from green (0%) to orange (50%). The plots at the bottom show the bone marrow blast count (left y axis, black lines) from pathology assessment, and donor chimerism (right y axis, grey line) after allogeneic stem-cell transplant. **b**, VAFs (y axis) of somatic mutations (x axis) detected in uninvolved bone marrow (top) and two BPDCN skin tumours from distinct anatomical sites (middle and bottom) in patient 10 at diagnosis, as in **a**. Mutations are grouped according to the sample in which they were first detected (left, bone marrow; middle left, shared skin 1 and 2; middle right, unique to skin 1; right, unique to skin 2). UV-associated TC>TT mutations are indicated in orange, and other mutations are indicated in green. The hashes indicate mutations of which VAFs are affected by copy-number alterations or location on chromosome X. **c**, Normalized read coverage on chromosome 9 for the two BPDCN skin tumours presented in **a** and **b**. Separate homozygous deletions affecting the CDKN2A tumour suppressor gene are indicated, with vertical bars showing the read coverage in the deleted (red bars) and non-deleted (blue bars) regions. **d**, Phased B allele frequencies (y axis) of heterozygous SNVs on chromosome 3 for the two BPDCN skin tumours presented in **a**–**c**. The vertical bars indicate the allele frequencies in a region containing the SETD2 tumour suppressor gene. Blue bars, A allele lost; red bars, B allele lost.

## Loss of Tet2 protects pDCs from UV effects

We next evaluated the role of UV damage in malignant transformation and disease evolution. Initially, we reasoned that UV-induced DNA damage would trigger malignant transformation in pDCs through disruption of tumour suppressors or activation of oncogenes. However, as has been demonstrated for skin cancer, assigning UV damage as the causal mechanism for isolated mutations is challenging, unless they happen to involve UV-specific CC>TT substitutions[32]. Among 12 such CC>TT mutations identified in our cohort, one affected a known leukaemia driver gene (ETV6 p.R369W, patient 14; Fig. 5a (left)) and

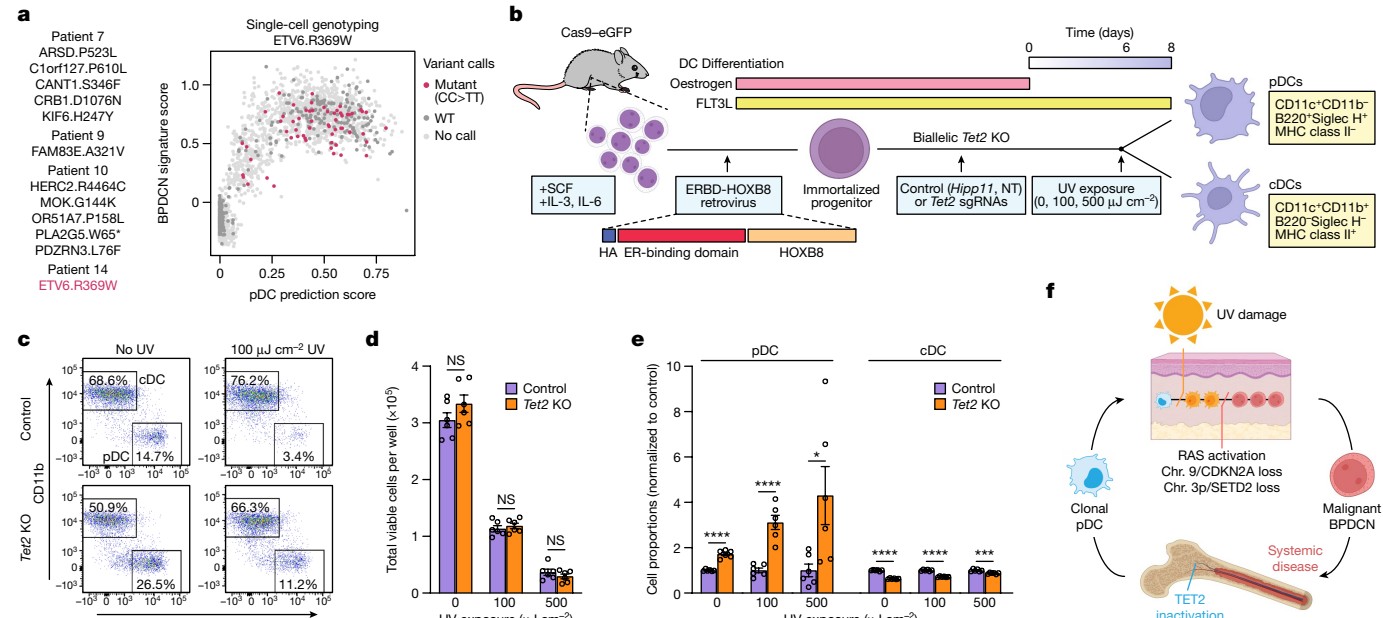

**Fig. 5 | *Tet2* loss protects pDCs from UV-induced cell death. a**, All detected UV-specific (CC > TT) gene mutations, including ETV6.R369W (red) in patient 14 (left). Right, XV-seq analysis of the bone marrow from patient 14 (*n* = 7,374 cells) showing the random-forest pDC prediction (*x* axis) and BPDCN signature (*y* axis) scores. The colours indicate cells with ETV6.R369W mutant calls (red), wild-type transcripts only (dark grey) and no variant calls (light grey). **b**, Ex vivo differentiation of dendritic cells from mouse bone marrow. Transduction of oestrogen-responsive HOXB8 generates progenitors capable of stable propagation and gene editing in vitro. Oestrogen withdrawal triggers differentiation over 6–8 days into pDCs and cDCs. UV exposure was performed on day 6, and cells were further differentiated until day 8. ER, estrogen receptor. **c**, Representative flow cytometry analysis of cDC (CD11b⁺B220⁻) and pDC (CD11b⁻B220⁺) populations in the control, *Tet2*-knockout and UV-exposed conditions. Gating was performed on viable CD11c⁺ cells, as in Extended Data

Fig. 10d. **d**, Viable cells (*y* axis) in control and *Tet2*-knockout cells on day 8 after UV exposure on day 6. **e**, The proportion of viable cells (*y* axis) classified as pDCs or cDCs in control (purple) or *Tet2*-knockout (orange) conditions at the indicated UV doses. Data are normalized to the 0 UV condition. **f**, Proposed model for BPDCN development in UV-associated cases. Clonal (premalignant) pDCs/pDC-like precursors arise in the marrow and seed the skin. These cells are then exposed to UV, undergo clonal selection and acquire additional mutations during malignant transformation. Malignant cells then spread systemically, including through retrograde dissemination back to the bone marrow. For **d** and **e**, data are mean ± s.e.m., and include two control and two *Tet2* gRNAs performed in triplicate, representative of two independent experiments. Statistical analysis in **d** and **e** was performed using two-sided Student's *t*-tests; ***$P$ < 0.001. The diagrams in **b** and **f** were created using BioRender.

XV-seq confirmed its presence exclusively within BPDCN cells (Fig. 5a (right) and Extended Data Fig. 10c). Nonetheless, definitive UV-induced oncogenic events could not be identified for the remaining cases in our cohort.

We next considered another hypothesis whereby UV exposure might provide a selective pressure for premalignant pDCs. To test this, we applied an ex vivo culture system of primary mouse bone marrow progenitors transduced with an oestrogen-responsive form of *HOXB8*[33] (Fig. 5b). In the presence of oestrogen and FLT3 ligand, this system enables stable propagation and genome editing of myeloid progenitor cells in vitro. Oestrogen withdrawal can then be used to trigger synchronized differentiation over 6–8 days into mature pDCs and conventional DCs (cDCs)[33] (Extended Data Fig. 10d,e).

We exposed these cultures to UV radiation on day 6 of differentiation after oestrogen withdrawal. By titrating UV exposure down to a low level, we identified a dose range (0, 100, 500 µJ cm⁻²) that induced a reproducible gradient of cell death over a 24–48 h period[34] (Fig. 5c,d). We next investigated whether premalignant pDCs were more resistant to UV-induced cell death compared with normal pDCs or cDCs. We used CRISPR–Cas9 to generate HOXB8 progenitors with biallelic *Tet2* mutations—the most common premalignant alteration in BPDCN—and then differentiated these cells into mature pDCs and cDCs (Extended Data Fig. 10f). In the absence of UV, *Tet2* knockout enhanced the proportion of pDCs (1.72-fold, $P$ < 0.0001), consistent with a role for TET2 in DC differentiation[17,35] (Fig. 5c). After UV exposure (100, 500 µJ cm⁻²), *Tet2* knockout caused a further increase in the proportion of surviving pDCs, but had no protective effect on cDCs (Fig. 5e). This suggests a

tumour suppressor role for TET2 in UV-exposed pDCs, and may explain the strong association between *TET2* inactivation, skin localization and UV-associated mutations in BPDCN.

## Discussion

Here we studied the development of BPDCN, an unusual acute leukaemia that originates from clonal precursors in the bone marrow but often presents with malignant cells isolated to the skin. Using tumour phylogenomics and single-cell profiling, we found that premalignant clones in the bone marrow give rise to BPDCN tumours in the skin. BPDCN cells are distinguished from clonal precursors by numerous UV-associated mutations that are absent from other blood-cell lineages. We found that UV-associated mutations can begin accumulating before malignant transformation and that BPDCN skin tumours first appear at sun-exposed sites. Taken together, this implicates UV exposure of premalignant pDCs or committed precursors in the skin during tumour pathogenesis. Moreover, we show that malignant cells from the skin can spread through retrograde dissemination to previously uninvolved bone marrow, and integrate single-cell gene expression and mutation analysis to identify rare circulating UV-damaged BPDCN cells (Fig. 5f).

Although pDCs are not present at high numbers in normal skin, they are recruited during antiviral, autoimmune and wound-healing responses after skin injury or UV damage[36–38]. pDCs recruited to the skin can be eliminated by UV phototherapy, which may contribute to the benefit of this treatment for inflammatory dermatoses[36,39]. Consistent with this, we show that normal pDCs are highly sensitive

to UV-induced cell death, whereas *Tet2*-mutated pDCs are relatively protected. These data suggest that biallelic inactivation of *TET2*, a highly recurrent premalignant alteration in BPDCN, facilitates pDC expansion and mutagenesis. We propose that UV exposure contributes to the anatomical localization and selective pressures faced by clonal pDCs or precursors during the early stages of BPDCN development in the skin, and may explain the high fraction of *TET2* alterations observed in this malignancy. Areas for future research include determining how TET2 affects the DNA damage response, and whether UV mutations create neoantigens that could sensitize to immunotherapy.

Finally, our findings demonstrate how exposure of circulating (clonal) immune cells to tissue-specific environmental exposures can shape mutational evolution and disease progression. Although BPDCN is relatively rare, UV signatures have also been observed in normal and malignant T cell populations and in a subset of paediatric B lymphoblastic leukaemia/lymphoma[40–42]. Together, these data support a model for clonal selection and premalignant evolution in the blood system that can be influenced by more than one tissue environment. More broadly, the role of tissue-specific mutational processes in shaping the evolution of clonal disorders across anatomical sites warrants further investigation.

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

## Methods

### Patient samples

Patients with BPDCN seen at the Dana-Farber Cancer Institute provided informed consent to an IRB-approved research protocol permitting tissue collection and sequencing analysis. The demographic characteristics of the patient cohort are provided in Supplementary Table 1a. Healthy control participants for single-cell sequencing consented to IRB-approved research protocols from Brigham and Women's Hospital or Lonza Bioscience that cover all of the study procedures; demographics are provided in Supplementary Table 3a.

### Histopathology

Histological processing and immunohistochemical staining of patient bone marrow and skin tumour biopsies was performed according to routine clinical procedures in the Department of Pathology at the Brigham and Women's Hospital, as previously described[26,43,44]. Results are included in Supplementary Table 1a.

### Targeted DNA sequencing and analysis

Targeted sequencing of fresh bone marrow samples using a 95-gene leukaemia panel (Rapid Haem Panel, $n$ = 23 samples) and formalin-fixed, paraffin-embedded archival skin tumour samples using a 282-gene pan-cancer panel (Oncopanel, $n$ = 9 samples) was performed for genes recurrently altered in myeloid malignancies and BPDCN[45–48]. Mutation calls were manually inspected and verified in sequence alignment files for each sample. Combined mutation calls and VAFs across all samples are provided in Supplementary Table 2a. For the patient 2 bone marrow sample, we observed a lower read coverage for amplicons covering the *ASXL1* gene relative to other samples and controls (67%, corresponding to a VAF of 0.33, located on chromosome band 20q11) and, accordingly, included this copy-number alteration in Fig. 1b.

### WES analysis

WES analysis of cryopreserved bone marrow, skin tumour and germline samples (uninvolved skin for patient 7, bone-marrow-derived fibroblasts for patients 9 and 10) was performed using the Illumina HiSeq 4000 (2 × 150 bp, patient 7) and the BGISEQ-500 (2 × 100 bp, patients 9 and 10) platforms, as previously described[49]. An overview of all of the profiled samples is provided in Supplementary Table 1b. Sequencing data for a total of 12 samples were mapped to the human genome reference (hg19; https://www.ncbi.nlm.nih.gov/data-hub/genome/GCF_000001405.13/) using BWA (v.0.7.15)[50]. The resulting BAM files were further analysed and recalibrated with Picard (v.2.5.0)[51] and the GATK toolkit (v.4.0.0.0)[52]. Somatic mutations were identified using Mutect2[53] by comparing to patients' germline variations. Initial calls were filtered by estimated cross-sample contamination and artifacts related to orientation bias. Calls were then merged for each patient and further filtered by removing calls that showed a VAF lower than 0.1 in all samples, with the exception of mutations that were also identified by targeted sequencing (that is, mutations in *TET2* in patient 10). We also excluded calls that showed a germline VAF that was greater than one-fifth of the highest VAF of other samples from the same patient, and mutations that were detected across multiple patients (with the exception of hotspot mutations in *ASXL1* in patients 7 and 10). Resulting high-confidence mutation calls for each sample are provided in Supplementary Table 2b–d. For the patient 10 relapse bone marrow sample, which was collected after the patient received a stem cell transplant, we were unable to define sample-specific mutations owing to the high proportion of donor DNA. However, we could quantify mutations in this sample identified in other samples from the same patient.

Combined mutation calls were the basis for the inference of tumour phylogenies (Figs. 1c and 4a) and the inference of putative clonal architectures in patient bone marrow samples (Fig. 1d). Mutation calls were further analysed for mutational signatures defined in the COSMIC database[54] by applying the R package MutationalPatterns[55]. The relative contribution of 30 different mutational signatures was calculated and scores for UV-light-associated signature 7 are indicated (Fig. 3b). For copy-number analysis, SNVs were jointly identified for all samples from each patient using bcftools (v.1.10.2; commands mpileup and call). The B (minor) allele frequency and read coverage (relative to germline samples) for each SNV was used to infer copy-number alterations (Extended Data Figs. 2 and 3). Public datasets (Extended Data Fig. 8a) were analysed from provided mutation calls, with the exception of data from ref. 28, which were processed from raw sequencing reads (Sequence Read Archive: SRP301976) and analysed using Mutect2.

### WGS analysis

WGS analysis of germline, bone marrow and skin tumour samples at diagnosis and relapse (13 samples from patients 1, 3 and 12) was performed using the BGI DNBSEQ platform (2 × 100 bp). Bone marrows were profiled from cryopreserved samples, skin tumours were profiled from formalin-fixed paraffin-embedded (FFPE) samples and germline samples were profiled from both sample types (Supplementary Table 1b). Sequencing data were mapped to the human genome (hg19) reference using BWA (v.0.7.17)[50]. The resulting BAM files for all of the samples from each patient were jointly analysed using Mutect2[53] by comparing to the matched germline sample, supplying both a germline resource (somatic-b37_af-only-gnomad.raw.sites.vcf) and a panel of healthy individuals (somatic-b37_Mutect2-exome-panel.vcf). Variant calls were filtered using GATK[52] (v.4.2.3.0; commands LearnReadOrientationModel and FilterMutectCalls) and annotated using the Funcotator command (funcotator_dataSources.v1.6.20190124s). The resulting calls were further filtered by retaining only variants that had a coverage greater than 20 for cryopreserved samples and greater than 10 for FFPE samples in all samples per patient. Final mutation calls were defined at the latest timepoint per patient: in patient 1, we considered variants detected in both skin relapse 2 and bone marrow relapse (collected after the patient received a stem cell transplant) with a VAF of greater than 0.25. In patient 3, we considered variants detected in the diagnostic bone marrow with a VAF of greater than 0.1. In patient 12, we considered variants detected in bone marrow relapse with a VAF of greater than 0.25. These filtering steps were deemed to be appropriate owing to the lower quality of FFPE-derived samples, and challenges due to high proportions of donor DNA in patients who received a stem cell transplant. Variants were attributed to the first sample in which they were detected with a VAF greater than 0.1, and all subsequent samples.

The resulting mutation calls were visualized in tumour phylogenies (Extended Data Fig. 1d) and were analysed for mutational signatures similar to as described for the WES dataset above (Fig. 3b). Summaries of mutation calls for each sample are provided in Supplementary Table 2e. For copy-number analysis, single-nucleotide variants were jointly identified for all of the samples from each patient using bcftools (v.1.10.2; commands mpileup and call). The B allele frequency and read coverage (relative to germline samples, not shown) for each SNV was used to infer copy-number alterations (Extended Data Fig. 4).

### scRNA-seq

scRNA-seq was performed on cryopreserved bone marrow aspirates. Cells were stored in liquid nitrogen, thawed using standard procedures and viable (propidium iodide negative) cells were sorted on the Sony SH800 flow cytometer. Next, 10,000–15,000 cells were loaded onto a Seq-Well array or 10x Genomics chip. Further processing was performed using the recommended procedures for the Seq-Well S[3] (http://shaleklab.com/resource/seq-well/)[56] or the 10x Genomics 3′ v3/v3.1 chemistry. Seq-Well S[3] libraries were sequenced on the NextSeq system (20 + 8 + 8 + 57 cycles) and 10x libraries were sequenced on the NovaSeq system (28 + 8 + 91 cycles for single-index or 28 + 10 + 10 + 75 cycles for dual index). Some of the data were previously reported[57,58] (Supplementary Table 3a). Serial samples from the same patient were loaded onto

separate sequencing runs to avoid erroneous assignment of reads by index swapping between diagnosis/remission/relapse samples (this is particularly relevant for the identification of rare malignant cells).

## XV-seq

We developed an improved method for targeted enrichment of genetic variants from scRNA-seq libraries that is compatible with Seq-Well S[3] and 10x 3′ gene expression platforms. Compared to previous methods by us and others[26,59], we incorporated a number of computational and experimental steps for increased sensitivity and specificity: (1) we considered all mutations detected by WES, including synonymous mutations and mutations affecting untranslated regions (UTR). These mutations do not result in changes in the protein sequence, but can be used to infer clonal relationships. (2) We quantified detection of these mutations in the regular scRNA-seq data before enrichment. For example, of the 186 mutations detected across samples for patient 10, only 16 (8.6%) were detected in at least one transcript (Supplementary Table 2d). We found that detection in the regular scRNA-seq data is a good predictor of enrichment efficiency (Extended Data Fig. 6c). (3) We specifically considered loci of which only a single allele is present in the genomes of healthy and/or malignant cells. For these mutations, detection of the wild-type allele is as informative as the presence of the mutant allele (that is, if the wild-type is detected, the mutant must be absent; for heterozygous mutations, the mutant could remain undetected). In our dataset, this included a mutation in the X-chromosomal gene *RAB9A* (in a male patient), a focal deletion of *CDKN2A/B*, which occurred in cells already carrying loss of chromosome 9, and 3′ UTR mutations in *SETX* and *SMARCC1*, which also occurred in cells with loss of the other allele on chromosome 9 and chromosome 3. (4) Finally, we incorporated technical optimizations such as inclusion of dual indices, as outlined below.

## XV-seq for Seq-Well S[3]

Compared with single-cell genotyping of Seq-Well S[3] libraries that we previously reported[26], we adjusted primer designs to generate dual-indexed libraries. This increases the confidence that reads are assigned to the correct library, particularly when using Illumina instruments with patterned flow cells. We first designed biotinylated mutation-specific primers to detect each of the known mutations in a given sample (Supplementary Table 4a). As a starting material, we used amplified cDNA from the Seq-Well S[3] protocol (also known as whole-transcriptome-amplified material). We then set up a biotin-PCR reaction to add a biotin tag and Nextera adapter to our gene of interest while retaining the unique molecular identifier (UMI) and cell barcode, as follows. We created a mixture containing a standard reverse primer at 3 μM (SMART-AC), and mutation-specific primers at a combined concentration of 3 μM. To prepare the template for the biotin-PCR reaction, we pooled and diluted whole-transcriptome-amplified products from the same sample and timepoint to 10 ng in a total volume of 10 μl. We next added 2.5 μl of primer mix (final concentration, 0.3 μM) and 12.5 μl of 2× KAPA HiFi Hotstart Readymix (Roche, NC0465187) to the template. We performed PCR using the following conditions: initial denaturation at 95 °C for 3 min; followed by 12 cycles of 90 °C for 20 s, 65 °C for 15 s and 72 °C for 3 min; and final extension at 72 °C for 5 min. After amplification, we purified the PCR product with 0.7× AMPure XP beads and captured biotinylated fragments using Streptavidin beads.

To add Illumina adapters, dual-indexed barcodes and a custom read primer binding sequence to the fragments, we performed a second PCR using the Streptavidin-bound product as a template (23 μl), with 2 μl of a 5 μM primer mix (N70D primers, Supplementary Table 4b) and 25 μl PFU Ultra II HS 2× Master Mix (Agilent, 600850). The parameters used for the second biotin-PCR were as follows: initial denaturation at 95 °C for 2 min; then 4 cycles of 95 °C for 20 s, 65 °C for 20 s and 72 °C for 2 min; followed by 10 cycles of 95 °C for 20 s and 72 °C for 2 min and 20 s; and then final extension at 72 °C for 5 min. After the second PCR,

we magnetized the Streptavidin beads and saved/purified DNA from the supernatant with 0.7× AMPure XP beads. After eluting in 20 μl of TE, we magnetized the beads and saved the supernatant for sequencing on the Illumina NextSeq system.

## XV-seq for 10x

We adjusted the Genotyping of Transcriptomes[59] protocol by (1) omitting staggered handles on gene-specific primers and (2) incorporating dual 10 bp library indices, which minimizes the chance of barcode swapping and ensures compatibility with 10x Genomics scRNA-seq v3.1 libraries and Illumina v1.0 and v1.5 chemistry. The starting material for transcript genotyping were the full-length cDNA libraries generated according to the 10x Genomics 3' v3 or v3.1 scRNA-seq protocol. If cDNA quantities were limited, we performed a full-length cDNA PCR amplification using generic primers that bind to all transcripts (primers: PartialRead1 and PartialTSO; Supplementary Table 4c). The pre-enrichment PCR was set up by mixing 10 ng of cDNA template, forward and reverse primers at 0.3 μM each, 2× Kapa HiFi HotStart ReadyMix and H$_2$O up to 50 μl. The PCR was performed under the following conditions: initial denaturation at 95 °C for 3 min; followed by 6 cycles of 98 °C for 20 s, 67 °C for 15 s, 72 °C for 3 min; and a final extension of 72 °C for 3 min. After amplification, we purified the PCR product with 0.6× AMPure XP beads (Beckman Coulter Life Sciences, A63881).

The enrichment for loci of interest consists of two PCR reactions. For PCR1, to enrich for loci of interest (determined by targeted or exome sequencing), we designed primers to amplify specific regions (Supplementary Table 4a). We downloaded the transcript sequence in Geneious Prime 2020, and annotated the mutation of interest. We designed mutation-specific primers within 50 bases upstream of the mutation site (so that the mutation site would be captured in read 2 of the sequencing data). To add a read 2 sequence to this mutation-specific primer, we appended CACCCGAGAATTCCA at the 5′ end. PCR1 was performed using these mutation-specific primers and a generic forward primer (PartialRead1; Supplementary Table 4c). We mixed up to six mutation-specific primers per PCR1 reaction, as long as they targeted different transcripts. We prepared PCR1 reactions as follows: 100 ng cDNA was added to 0.25 μM forward primer and 0.25 μM mutation-specific primer(s), 20 μl 2× Kapa HiFi HotStart ReadyMix and H$_2$O up to 40 μl. The PCR was performed under the following conditions: a denaturation step at 95 °C for 3 min; followed by 10 cycles of 98 °C for 20 s, 67 °C for 15 s, 72 °C for 3 min; and a final extension 72 °C for 3 min. After amplification, we purified the PCR product with 1× AMPure XP beads. We next performed PCR2 to generate indexed libraries compatible with the Illumina NextSeq and NovaSeq machines.

For PCR2, we used a P5 sequence followed by a 10 bp index barcode and a read 1 sequence as a forward primer (XV-P5-i5-BCXX) and a P7 sequence followed by a 10 bp index barcode and a read 2 sequence as a reverse primer (XV-P7-i7-BCXX; Supplementary Table 4c). The PCR was set up as follows: 18 μl of the PCR1 product was added to 2 μl primers (0.25 μM each) and 20 μl Kapa HiFi HotStart ReadyMix. The PCR was performed under the following conditions: 95 °C for 3 min; followed by 6 cycles of 98 °C for 20 s, 67 °C for 15 s, 72 °C for 3 min; and a final extension 72 °C for 3 min. After amplification the PCR product was purified with 1× AMPure XP beads. Elution in 20 μl buffer TE typically yielded 5–50 ng μl$^{-1}$ with an average size of 300–1,500 bp, which was pooled for sequencing on the Illumina NextSeq or NovaSeq instruments with the goal of generating 10 million reads per library.

## scRNA-seq computation analysis

Data from the Seq-Well protocol (healthy donor 6 and patient 9) were processed as described previously[26]. In brief, demultiplexed fastq files were processed to maintain only cell barcodes with 100 reads and to append the cell barcode and UMI, derived from read 1, to the read identifier of read 2. The hg38 reference genome and annotations were downloaded from Ensembl (release 99), extended with *RNA18S* and

*RNA28S* genes from UCSC, and finalized using Cell Ranger mkgtf with the recommended settings and the additional gene biotypes Mt_rRNA and rRNA. We then used STAR (v.2.6.0c) to align processed fastqs to hg38 and created a count matrix. Data from the 10x Genomics 3′ v3 and v3.1 platform (the remaining 15 samples) were processed using Cell Ranger (v.7.0.0) using the default settings and the same hg38 reference. Count matrices from both the Seq-Well and the Cell Ranger pipelines were processed to retain only cells with >2,000 UMIs, >1,000 genes and <20% mitochondrial alignments. From the count matrix, we removed mitochondrial genes (^MT-*), genes of the biotype rRNA (defined in the reference gtf file) and *RNA18S*/*RNA28S*. We maintained X- and Y-chromosomal genes, including *ZRSR2* and *IL3RA*.

## XV-seq computational analysis

To quantify detection of mutations in regular scRNA-seq libraries, we assessed every mutation detected by exome sequencing in the genome alignments for the respective sample. Mutations were quantified using samtools mpileup. For each base, information for cell barcode and UMI was obtained by setting the --output-extra option, and subsequently collapsed using R and the data.table package. Mutations that were most efficiently detected or that were of special interest were selected for XV-seq enrichment.

For analysis of XV-seq data, fastq files were processed using IronThrone-GoT (v.2.1) using the recommended set-up (https://github.com/landau-lab/IronThrone-GoT)[59]. For patient 9, we used --bclen 12, --umilen 8 and a whitelist of cell barcodes that passed RNA-seq quality controls. For all of the other samples, we used --bclen 16, --umilen 12 and the whitelist 3M-february-2018.txt. For every mutation, we generated custom configuration files to distinguish between wild-type and mutant transcripts by one or several differing bases. If the mutation site was directly adjacent to the primer, the 3′ end of the primer was used as a shared sequence and additional bases were added to the wild-type/mutant sequences, taking into account that IronThrone-GoT allows for 20% of the bases in the analysed reads to be mismatched from the provided sequences. For *MTAP*, five configuration files were used, one for each of the potential splicing products indicating the *CDKN2A/B* deletion (Extended Data Fig. 6b). IronThrone-GoT jobs were submitted in Linux using the Sun Grid Engine with the options -pe smp 4 -binding linear:4 -l h_vmem=32g -l h_rt=96:00:00. After completion of the IronThrone-GoT run, we processed the generated information (summTable) by plotting the number of wild-type and mutant calls for different sequencing reads of each transcript (UMI). We used only transcripts that were supported by ≥3 reads and with at least threefold more wild-type than mutant calls or vice versa. For MALAT1.n.G3541A in Fig. 2e,f and Extended Data Fig. 8, we reduced the read threshold to 1. In the case of heterozygous mutations, cells in which a wild-type transcript is detected are not necessarily wild-type cells, as the mutated allele may have been missed. In the case of multiple mutations within the same gene (as is observed for *TET2* in BPDCN), transcripts may show a wild-type result at one site while still harbouring a mutation in *cis* at a different position in the same transcript/allele. We added the genotyping information as metadata to Seurat objects with scRNA-seq expression data by joining based on cell barcodes.

To check the accuracy of our single-cell mutation calls, we validated the ASXL1.G642fs mutation in Pt10Dx using two different enrichment primers. This known oncogenic guanine insertion, resulting in ATCG GAGGGGGGGGT>ATCGGAGGGGGGGGGT, can be challenging to call. We enriched the mutation site from 10,106 high-quality single-cell transcriptomes using two different primers: ASXL1-1886 (CACCCGAG AATTCCA**GTCACCACTGCCATAGAGAGG**) and ASXL1-1898 (CACCC GAGAATTCCA**ATAGAGAGGCGGCCACCA**; the first one is included in Supplementary Table 4a) (transcript-binding sequences are in bold). In the first experiment, we detected mutated *ASXL1* transcripts in nine cells. In the second experiment, we detected mutated transcripts in eight cells. Seven of the cells overlapped between the two attempts,

indicating striking concordance. We also detected wild-type *ASXL1* transcripts in 33 and 32 cells in the two experiments, respectively. There was perfect overlap in 32 wild-type cells that were called between the two experiments with different *ASXL1* enrichment primers. The agreement between these experiments, together with the orthogonal targeted DNA sequencing, which identified the same mutation, attests to the reliability of our mutation calls.

## Dimensionality reduction and cell type annotation

Count matrices from healthy donors were imported into R (v.4.2.1) using Seurat (v.4.1.1) on a MacBook Pro with an M1 Max chip. Normalization, variable feature identification and data scaling were performed using the Seurat defaults[60]. After principal component analysis, Harmony was used to integrate data from Seq-Well S[3] and 10x v3 3′ scRNA-seq[61]. We then used Harmony reduction to determine clusters and UMAP coordinates. Integration from Seq-Well and 10x platforms generated clusters that were driven by biological (rather than technical) differences between cells (Fig. 2a). Clusters were annotated by expression of canonical marker genes such as *CD34* (progenitors), *CD14* (monocytes), haemoglobin (erythroid), *IRF8* and *TCF4* (pDCs; Supplementary Table 3b). This yielded 21 healthy reference cell types. One cluster (1.04% of healthy donor cells) was classified as doublets on the basis of co-expression of marker genes.

To annotate cell types from the samples of patients with BPDCN, we used the random-forest algorithm[26]. Specifically, we used the R package randomForest (v.4.7-1.1) to generate a classification forest using marker genes of healthy donors (determined by Seurat's FindAllMarkers function); we previously showed that this approach performs similarly to other reference-based classification algorithms[57]. The confusion matrix and fivefold cross-validation both indicated 89.7% accuracy (Extended Data Fig. 5a). We next used the classification forest to assign each cell from the patient samples with prediction/probability scores for each reference cell type (function predict() with randomForest object and type = "prob"; see 3_RandomForest.R at https://github.com/petervangalen/Single-cell_BPDCN/). The reference cell type with the maximum prediction score was used for the patient cell classification. Cells that were classified as doublets (up to 2.34%) were excluded from further analysis. Projection of patient cells onto the UMAP of healthy donor cells was performed by plotting each patient cell at the coordinates of the normal cell with the highest prediction score correlation (Fig. 2b).

## Annotation of host and donor single cells

To annotate single cells from the patient 10 relapse bone marrow sample for their origin (this patient received an allogeneic stem cell transplantation prior to relapse), we quantified SNVs specific to the host or donor genome in each single cell. We first identified all SNVs in the relapse bone marrow exome sequencing dataset, which represents a mixture of both genomes ($n$ = 127,916; Extended Data Fig. 3b; see also the copy-number analysis above). For each SNV, we then quantified its B allele frequency in both the germline and relapse bone marrow sample. By applying thresholds on both frequencies, we identified variants that are informative for each genome (Extended Data Fig. 5e). We further removed variants that were located within broad copy-number alterations on chromosomes 3, 6 and 9, as well as on chromosomes X and Y. A total of 56,155 SNVs were identified in this manner, with 5,989 (10.7%) being homozygous in both genomes (that is, host A/A and donor B/B, or host B/B and donor A/A) and therefore informative for both alleles. These SNVs were quantified in the single-cell transcriptome data of the diagnostic and relapse sample using samtools mpileup. For each base, information for cell barcode and UMI was obtained by setting the --output-extra option. We then aggregated coverage for all host- and donor-specific alleles across the genome for each single cell. Cell annotations for the patient 10 relapse bone marrow sample were obtained for cells with a coverage of at least 10 and a donor-specific allele coverage of less than 10% (host cells, $n$ = 4,453) or greater than 90% (donor cells,

$n$ = 2,664; illustrated in Extended Data Fig. 5f). A small fraction of cells with donor-specific allele coverage between 10% and 90% potentially reflect cell multiplets and were removed from further analysis. In total, 94.9% of patient 10 relapse cells were annotated for their host/donor origin. As a control, none of the single cells from the patient 10 diagnostic bone marrow sample were classified as donor cells.

**BPDCN signature generation and single-cell gene expression analysis**

To generate a single-cell transcriptional signature specific for malignant BPDCN cells, we made two groups of cells: (1) cells classified as pDCs from healthy donors and (2) cells classified as pDCs from patients with marrow involvement. Cells classified as pDCs without progression mutations from patients without marrow involvement were also included in group 1 as they were similar to pDCs from healthy donors (Extended Data Fig. 7a). Cells classified as pDCs with progression mutations from patients without marrow involvement were not included in differential gene expression analysis because they were suspected circulating malignant BPDCN cells based on mutation and gene expression patterns. We randomly selected at most 50 cells per sample, so that the analysis would not be dominated by samples with a high number of pDCs. We then compared the two groups using the Seurat function FindMarkers and selected genes with twofold higher expression in the second group ($log_2$-transformed fold change > 1) and an adjusted $P < 1 \times 10^{-30}$ (ref. 62; Wilcoxon Rank Sum test, 45 genes; Extended Data Fig. 7b and Supplementary Table 3c). To identify malignant cells, we scored all 87,011 single-cell transcriptomes for this 45-gene signature using the Seurat function AddModuleScore (Extended Data Fig. 7c,d). We then defined malignant BPDCN cells as all cells classified as pDCs in patients with bone marrow involvement, as well as all cells (regardless of their initial classification) with a BPDCN signature score exceeding 0.5 (Extended Data Fig. 7f–h). To ensure that reclassification of a small proportion of cells as malignant BPDCN cells was justified, we checked marker gene expression, reasoning that the absence of canonical markers would support reclassification. Indeed, reclassified pro-B cells lacked CD19 and reclassified plasma cells lacked CD138 (Extended Data Fig. 7g). Using Seurat objects with scRNA-seq expression data and metadata (including cell type annotations and XV-seq mutation calls joined based on cell barcodes), we performed all downstream single-cell analyses in R with extensive use of the tidyverse[63].

**In vitro differentiation of dendritic cells and UV exposure**

HOXB8-FL cells were derived as described previously[33] from bone marrow cells of mice constitutively expressing Cas9 (Jackson Laboratory, 026179). HOXB8-FL cells were cultured in RPMI-1640 (Gibco, 11875093) supplemented with 10% FBS (Sigma-Aldrich, F2442), 1% penicillin–streptomycin (Corning, 30002CI), 50 ng μl$^{-1}$ mouse FLT3L (BioLegend, 550706) and 1 μM oestrogen (Sigma-Aldrich, E2758). Cells were resuspended in fresh medium every 2–3 days. For DC differentiation, HOXB8-FL cells were washed once in RPMI-1640, then resuspended in the same medium without oestrogen. For UV exposure on day 6 after-oestrogen withdrawal, cells were resuspended in PBS and exposed to the indicated doses of UV using an XL-1500 Spectrolinker. Cells were then resuspended in differentiation medium and analysed by flow cytometry on day 8 after oestrogen withdrawal. For CRISPR-mediated knockout of *Tet2*, Cas9-expressing HOXB8-FL cells underwent lentiviral transduction of sgRNA using the pLKO5.sgRNA.EFS.GFP vector (Addgene, 57822). Data were combined from two Tet2-targeting sgRNAs (GAATACTATCCTAGTTCCGAC and GAACAAGCTCTACATCCCGT). For controls, data were combined from non-transduced cells and cells transduced with sgRNA targeting a safe harbour region (ATGTACAACACAAACGAAGT). *Tet2*-sgRNA-induced indels were validated using PCR amplicon next-generation sequencing (Extended Data Fig. 10f). For flow cytometry analysis, differentiated HOXB8-FL cells were incubated for 10 min in Fc block (BD, 553141), then stained for

CD11b Alexa Fluor 700 (BioLegend, 101222), CD11c PE/Cyanine7 (BioLegend, 117317), B220 APC/Cyanine7 (BioLegend, 103224), Siglec-H PE (BioLegend, 129605) and MHC-II PerCP/Cyanine5.5 (BioLegend, 107626). cDCs were defined as CD11c$^+$CD11b$^+$B220$^-$MHC-II$^+$ and pDCs were defined as CD11c$^+$CD11b$^-$B220$^+$Siglec-H$^+$ (Extended Data Fig. 10d,e). DAPI staining was used to exclude non-viable cells.

**Reporting summary**

Further information on research design is available in the Nature Portfolio Reporting Summary linked to this article.

## Data availability

The WES/WGS data are available in the dbGaP under accession number phs003228. The single-cell sequencing data and gene expression matrices are available at the Gene Expression Omnibus under accession number GSE227690. Source data are provided with this paper.

## Code availability

Single-cell code is shared at GitHub and Zenodo (https://github.com/petervangalen/Single-cell_BPDCN and https://doi.org/10.5281/zenodo.7746255), and Seurat objects are available online (https://vangalenlab.bwh.harvard.edu/resources/).

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

**Acknowledgements** We thank all of the patients and families for participation in this research study; A. Kreso and A. S. Nam for discussions; and P. Rogers for flow cytometry support. G.K.G. is supported by the Damon Runyon Cancer Research Foundation; V.H. by the Charles H. Hood Foundation, the Children's Cancer Research Fund and the V Foundation; P.v.G. by the National Cancer Institute (R00 CA218832), the Starr Cancer Consortium, the Gilead Sciences Research Scholars Program in Hem/Onc, the Harvard Medical School Epigenetics & Gene Dynamics Initiative, the Leukaemia & Lymphoma Society, the Glenn Foundation for Medical Research,

the American Federation for Aging Research (AFAR) and the William Guy Forbeck Research Foundation; P.v.G. and A.A.L. by the Ludwig Center at Harvard and the Bertarelli Rare Cancers Fund; and A.A.L. by the NCI (R37 CA225191), Department of Defense (W81XWH-20-1-0683) and the Mark Foundation for Cancer Research. A.A.L is a Scholar of the Leukaemia & Lymphoma Society. None of the funding bodies were directly involved in the design of the study, nor in the collection, analysis or interpretation of the data, or writing of the manuscript. Some of the figure panels (Figs. 1a, 2c, 3a, 5b & 5f) were generated using BioRender.

**Author contributions** G.K.G., C.A.G.B., K.T., S.S.C., D.S., J.A.V., J.M.B., Y.S.L., V.S., J.L.H., N.R.L., E.A.M., B.E.B., V.H., P.v.G. and A.A.L. conducted experiments and analysed data. G.K.G., K.T., S.S.C., D.S., J.A.V., Y.S.L., B.E.B., V.H., P.v.G. and A.A.L. conducted sequencing experiments. G.K.G., V.S., J.L.H., N.R.L. and E.A.M. conducted clinical and pathological analyses. G.K.G., C.A.G.B., K.T., J.M.B. and A.A.L. generated mouse engineered haematopoietic cells and conducted in vitro differentiation experiments. G.K.G., V.H., P.v.G. and A.A.L. designed the study, interpreted the data and wrote the paper. All of the authors participated in editing the paper.

**Competing interests** B.E.B. discloses financial interests in Fulcrum Therapeutics, Chroma Medicine, HiFiBio, Arsenal Biosciences, Cell Signaling Technologies and Design Pharmaceuticals. A.A.L. received research funding from Abbvie and Stemline therapeutics, and consulting fees from Cimeio Therapeutics, IDRx, Jnana Therapeutics, N-of-One and Qiagen. P.v.G. received consulting fees from ManaT Bio and Immunitas.

**Additional information**
**Correspondence and requests for materials** should be addressed to Gabriel K. Griffin, Volker Hovestadt, Peter van Galen or Andrew A. Lane.

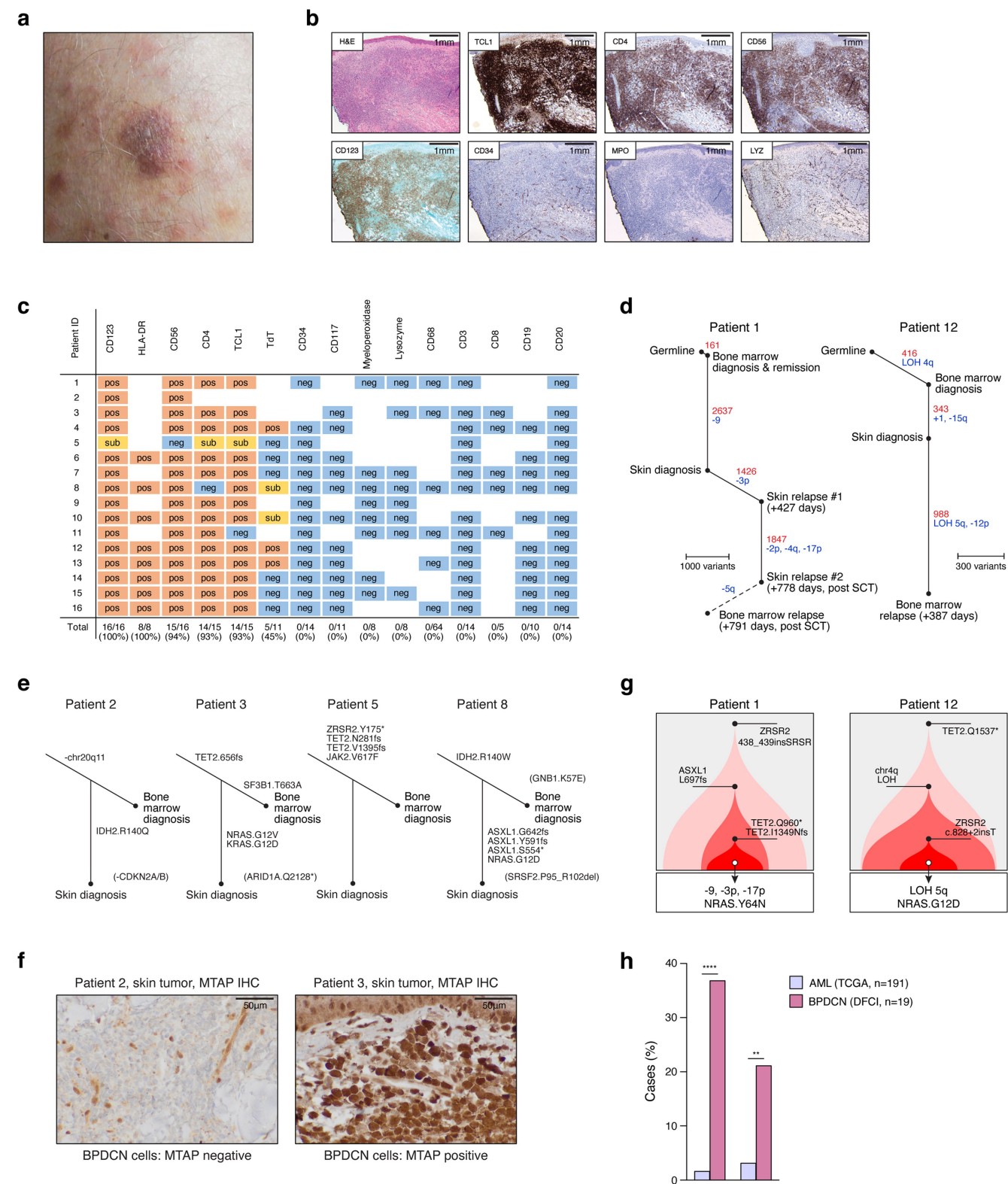

**Extended Data Fig. 1 | See next page for caption.**

**Extended Data Fig. 1 | Clinical evaluation of bone marrow involvement in BPDCN. a**, Clinical photo of a representative BPDCN skin lesion at initial diagnosis showing a violaceous nodule that occurred on a background of diffuse pink, brown and violaceous papules, plaques, and tumours. **b**, Low-power histologic images of a representative BPDCN skin tumour showing extensive dermal infiltration by malignant BPDCN cells with characteristic immunophenotype, including positivity for TCL1, CD4, CD56, and CD123, and negativity for CD34, myeloperoxidase (MPO), and lysozyme (LYZ). These features distinguish BPDCN in the skin from histologic mimics, including acute myeloid leukaemia with monocytic differentiation and chronic myelomonocytic leukaemia. **c**, Immunophenotypic features of the sixteen BPDCN cases included in the cohort, as determined by immunohistochemistry and/or flow cytometry performed during initial pathology evaluation. This shows the presence of BPDCN markers (TCL1, CD4, CD56, CD123) and the absence of myeloid, T, or B-cell lineage markers that define other forms of acute leukaemia. **d**, Tumour phylogenies and sample-specific alterations for Patients 1 and 12 as determined by whole-genome sequencing. Somatic SNVs (red) and copy-number alterations

(blue) were defined at the latest relapse time point, and then assessed in prior skin tumour and bone marrow diagnosis samples. **e**, Tumour phylogenies and sample-specific alterations for Patients 2, 3, 5, and 8 as determined by targeted sequencing (RHP and Oncopanel, see Methods). Mutations that are not covered in both the RHP (bone marrow samples) and Oncopanel (skin samples) assays are indicated in parentheses. **f**, Immunohistochemistry for MTAP protein (brown) as a marker of chr 9p21.3 loss in BPDCN skin tumours. MTAP is lost in skin tumour cells from patient 2, which contain a focal deletion on chr9p21.3, and retained in skin tumour cells from patient 3, which do not have chr9p21.3 loss. **g**, Inferred clonal architecture in diagnostic marrows for Patients 1 and 12 in (d). Premalignant subclones related to BPCN skin tumours are indicated with arrows, and further annotated with skin-specific progression mutations. **h**, Frequency of copy-loss in CDKN2A/chr9p and SETD2/chr3p in BPDCN and AML. Statistical significance by two-sided Fisher's exact test. TCGA, The Cancer Genome Atlas. DFCI, Dana-Farber Cancer Institute. **P < 0.01, ****P < 0.0001. Related to Fig. 1.

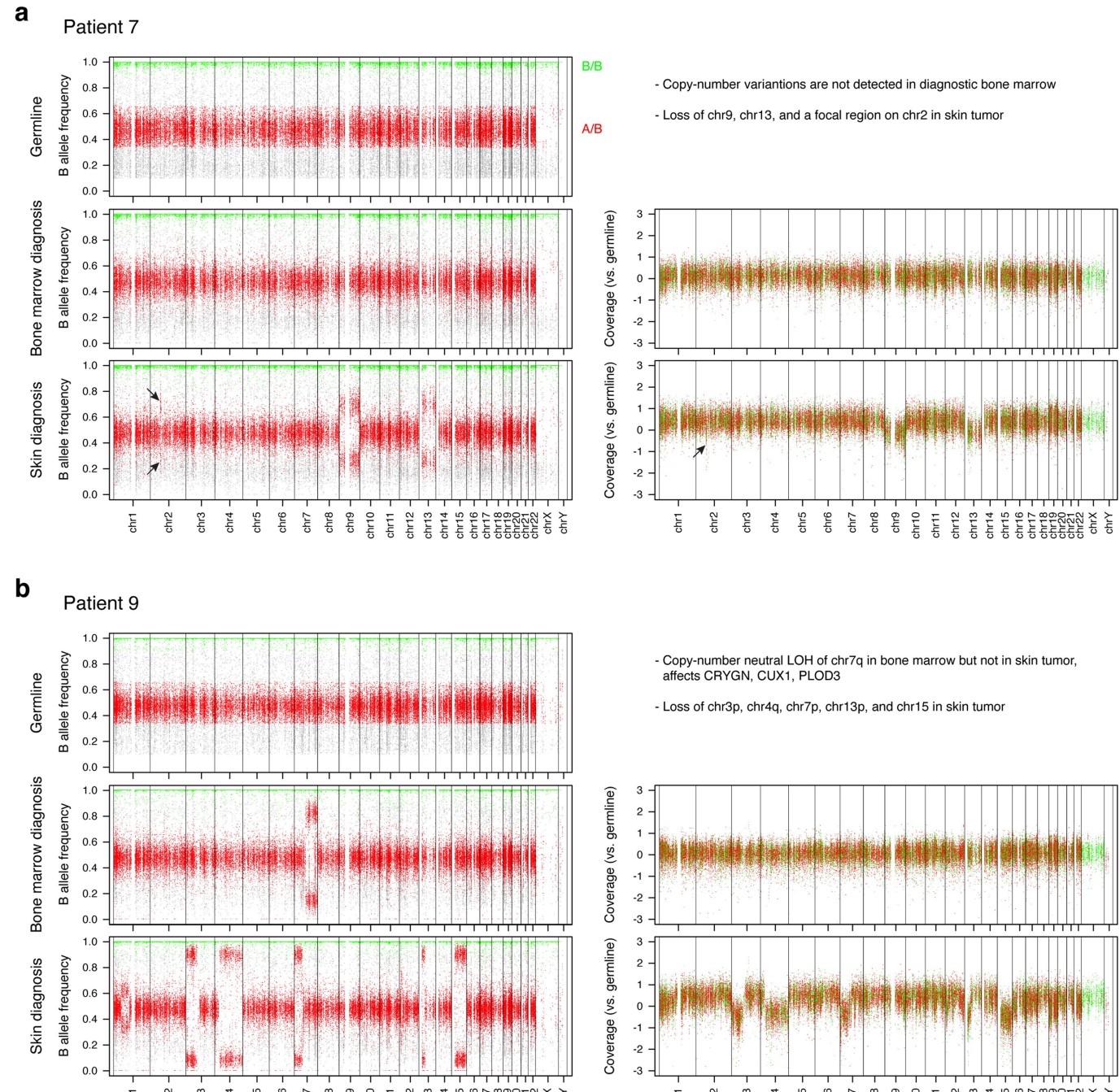

**Extended Data Fig. 2 | Copy-number alteration analysis of Patient 7 and 9 whole-exome sequencing data. a**, Genome plots show B (minor) allele frequencies (left) and read coverage (right) for single-nucleotide variants (SNVs) detected in the Patient 7 germline sample using whole-exome sequencing. Heterozygous SNVs (A/B) are indicated in red, homozygous SNVs (B/B) are indicated in green. The same SNVs are shown for the bone marrow and skin

tumour samples, indicating copy-number alterations specific to the skin tumour. A focal loss detected on chr2q is indicated by arrows. **b**, Genome plots show similar analysis for germline, bone marrow, and skin tumour samples of Patient 9. A sub-clonal copy-number neutral loss of heterozygosity (LOH) on chr7q was detected in the uninvolved bone marrow sample, which was not detected in the skin tumour. Related to Fig. 1.

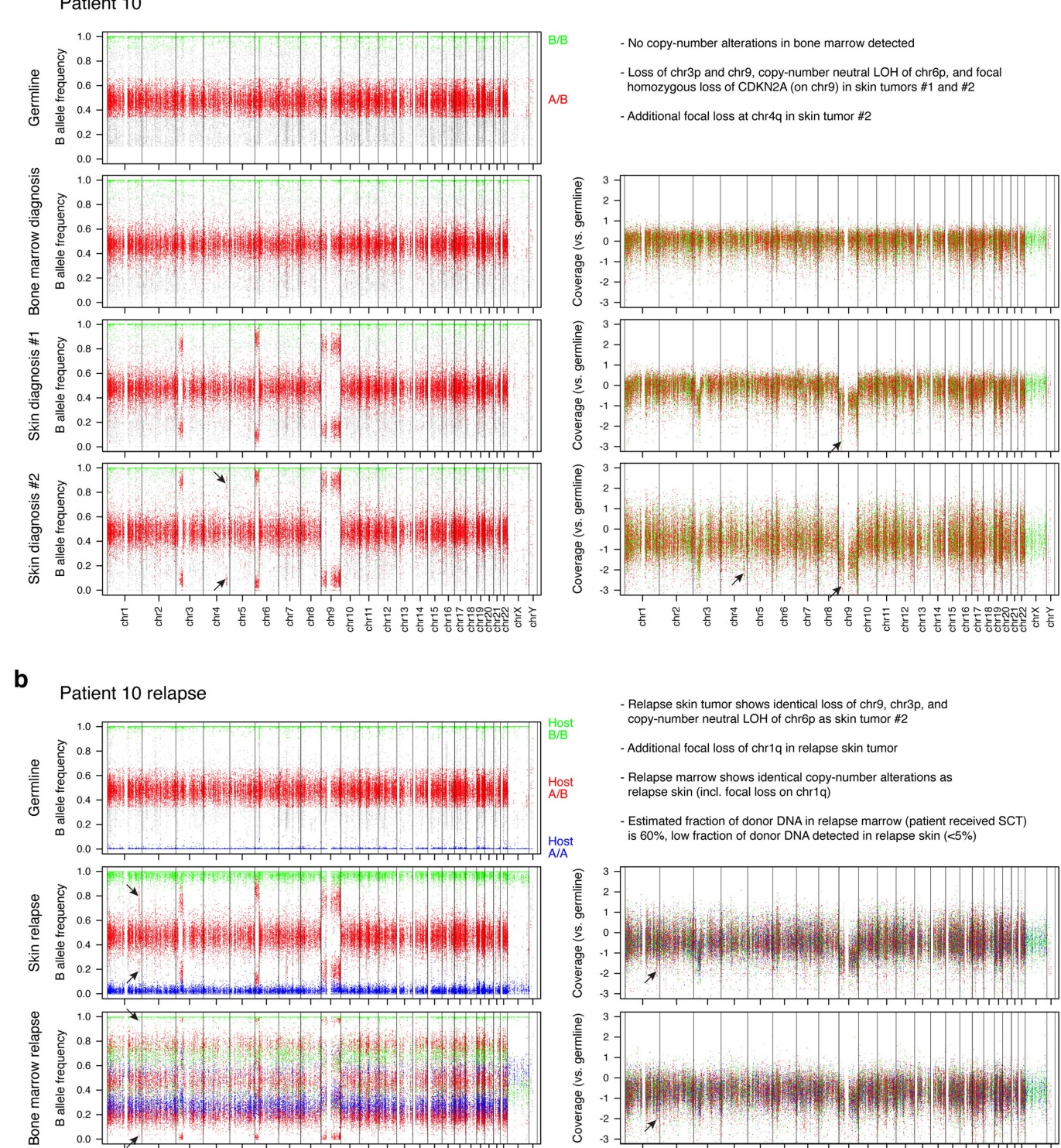

**Extended Data Fig. 3 | Copy-number alteration analysis of Patient 10 whole-exome sequencing data. a**, Genome plots show B (minor) allele frequencies (left) and read coverage (right) for single-nucleotide variants (SNVs) detected in the Patient 10 germline sample using whole-exome sequencing. Heterozygous SNVs (A/B) are indicated in red, homozygous SNVs (B/B) are indicated in green. The same SNVs are shown for bone marrow and skin tumour sample #1 and #2, indicating copy-number alterations specific to the skin tumours. A homozygous deletion of the *CDKN2A* gene locus is indicated by arrows. An additional focal loss on chr4q in skin tumour #2 is also indicated. **b**, Genome plots show SNVs detected in the relapse skin tumour and bone marrow samples of Patient 10 collected after receiving an allogeneic stem cell transplant. SNVs homozygous in the host germline sample are indicated in blue (A/A) and green (B/B). Heterozygous SNVs are indicated in red (A/B). A low level of donor DNA is detected in the relapse skin tumour sample (<5%). The estimated percentage of donor DNA is 60% in the relapse bone marrow sample. Copy-number alterations specific to the skin tumours (i.e. loss of chr9, chr3p, and copy-number neutral LOH of chr6p) are detected in both relapse samples. The only additional alteration is a focal loss on chr1q. Related to Fig. 1.

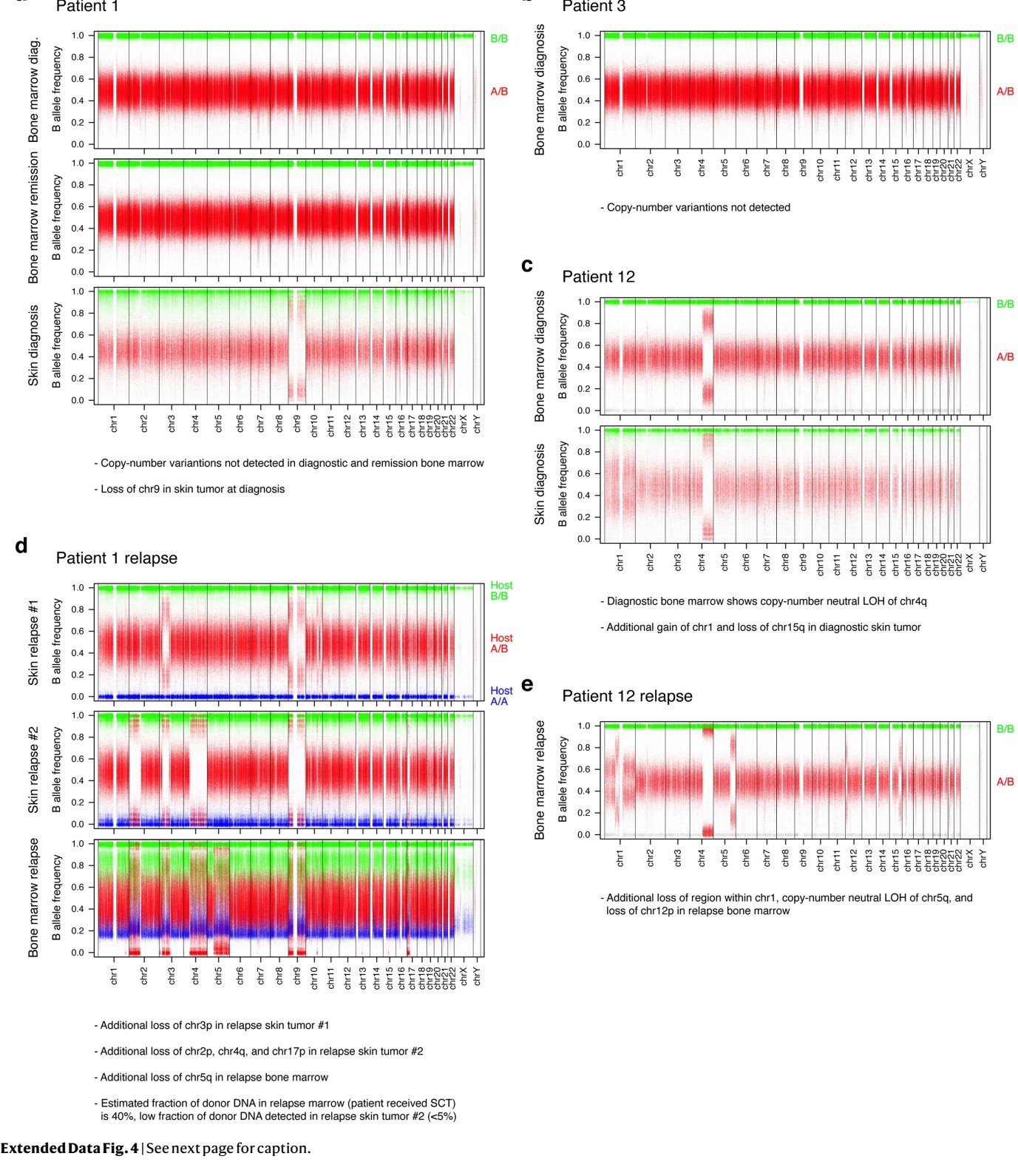

**a** Patient 1

Bone marrow diag. — B allele frequency

Bone marrow remission — B allele frequency

Skin diagnosis — B allele frequency

- Copy-number variantions not detected in diagnostic and remission bone marrow

- Loss of chr9 in skin tumor at diagnosis

**b** Patient 3

Bone marrow diagnosis — B allele frequency

- Copy-number variantions not detected

**c** Patient 12

Bone marrow diagnosis — B allele frequency

Skin diagnosis — B allele frequency

- Diagnostic bone marrow shows copy-number neutral LOH of chr4q

- Additional gain of chr1 and loss of chr15q in diagnostic skin tumor

**d** Patient 1 relapse

Skin relapse #1 — B allele frequency

Skin relapse #2 — B allele frequency

Bone marrow relapse — B allele frequency

- Additional loss of chr3p in relapse skin tumor #1

- Additional loss of chr2p, chr4q, and chr17p in relapse skin tumor #2

- Additional loss of chr5q in relapse bone marrow

- Estimated fraction of donor DNA in relapse marrow (patient received SCT)
  is 40%, low fraction of donor DNA detected in relapse skin tumor #2 (<5%)

**e** Patient 12 relapse

Bone marrow relapse — B allele frequency

- Additional loss of region within chr1, copy-number neutral LOH of chr5q, and
  loss of chr12p in relapse bone marrow

**Extended Data Fig. 4** | See next page for caption.

**Extended Data Fig. 4 | Copy-number alteration analysis of Patient 1, 3, and 12 whole-genome sequencing data. a**, Genome plots show B (minor) allele frequencies for all single-nucleotide variants (SNVs) detected in the Patient 1 germline sample using whole-genome sequencing. SNVs are shown for bone marrow samples at diagnosis and remission, and a skin tumour sample. SNVs that are heterozygous (A/B) in the germline sample are indicated in red, homozygous SNVs (B/B) are indicated in green. Loss of chr9 specific to the skin tumour sample is indicated. **b**, Genome plot shows similar analysis for a bone marrow sample of Patient 3, not harbouring any copy-number alterations. **c**, Genome plots show similar analysis for a bone marrow and skin tumour sample of Patient 12. Copy-number neutral loss of heterozygosity (LOH) on chr4q was detected in both samples. Additional loss of chr15q and gain of chr1 was detected in the skin tumour sample. **d**, Genome plots show single-nucleotide variants (SNVs) detected in the relapse bone marrow sample of Patient 1 collected after receiving an allogeneic stem cell transplant. SNVs are shown for skin relapse sample #1 and #2, as well as a bone marrow relapse. Samples were collected at consecutive time points. SNVs homozygous in the host germline sample are indicated in blue (A/A) and green (B/B), heterozygous SNVs are indicated in red (A/B). In addition to the loss of chr9 in the skin sample at diagnosis, additional copy-number alterations are detected in each sample: Loss of chr3p is first detected in skin relapse #1, loss of chr2p, chr4q, and chr17p are first detected in skin relapse #2, and loss of chr5q is first detected in the bone marrow relapse. Donor DNA is detected in both skin sample #2 and bone marrow relapse sample. Estimated fraction of donor DNA is <5% and 40%, respectively. **e**, Genome plot shows similar analysis for a bone marrow relapse sample of Patient 12, who did not receive an allogeneic stem-cell transplant. Additional alterations in chr1, chr5q, and chr12p are indicated. Related to Fig. 1.

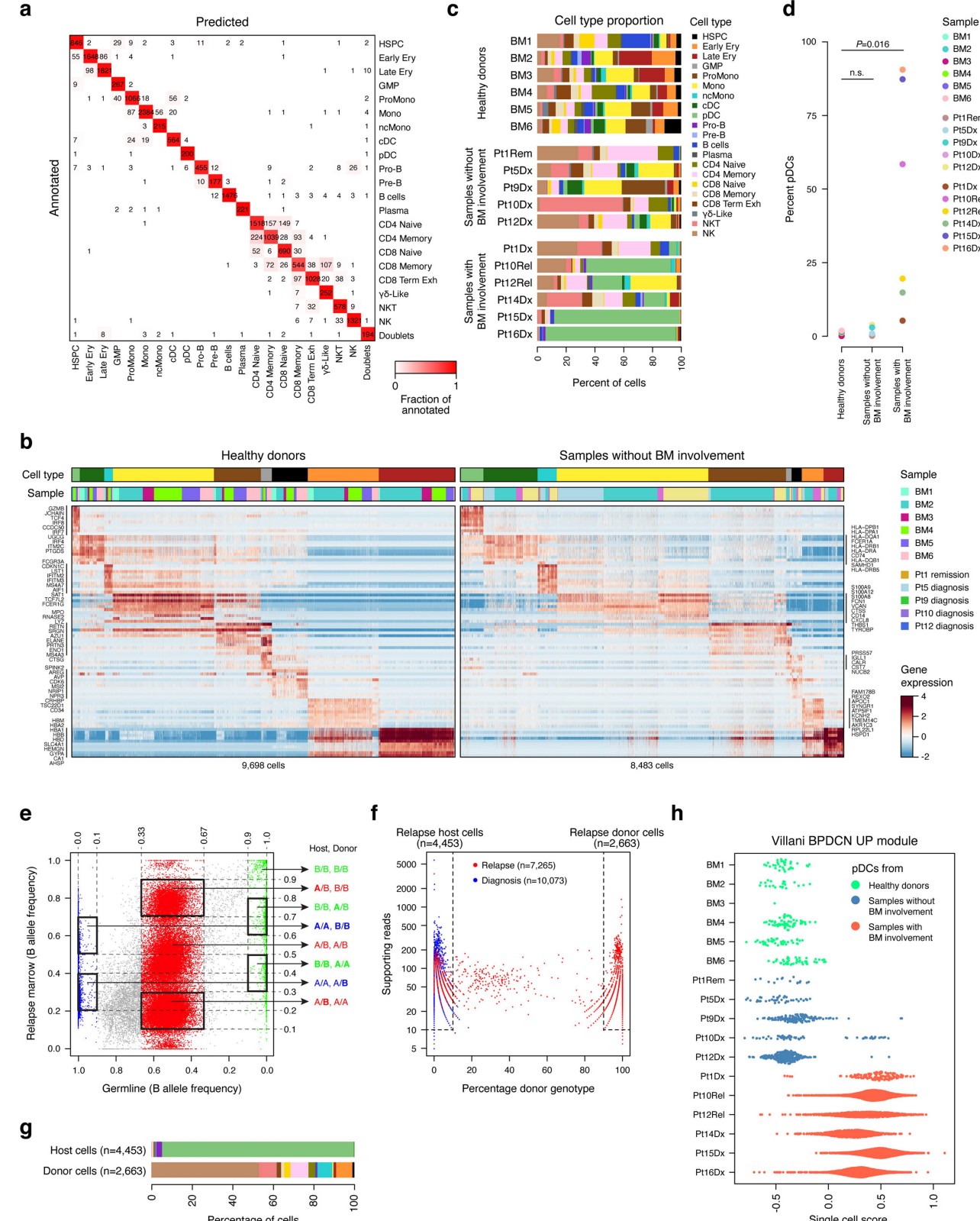

**Extended Data Fig. 5** | See next page for caption.

**Extended Data Fig. 5 | Single-cell transcriptome analysis of healthy donor and BPDCN patient bone marrow samples. a**, Heatmap depicts five-fold cross validation of the random forest classifier comprising 22 classes corresponding to the cell types identified in healthy bone marrow samples (including inferred cell doublets). The healthy donor cells were split such that 80% were used as a reference to predict the classification of the remaining 20%. This process was repeated five times. Cells that fall on the diagonal are classified according to their annotation. Cells that do not fall on the diagonal are mis-classified as a different cell type. **b**, Heatmaps show expression of known marker genes (rows) in cells (columns) of the myeloid and erythroid differentiation trajectories in bone marrow samples from healthy donors (left) and from BPDCN patients without known bone marrow involvement (right). Top annotation bars show cell types (colour legend provided in panel c) and sample identity. **c**, Barplot shows the proportion of cell types within each of the 17 analysed single-cell samples. Healthy donor cells were annotated by clustering and assessment of marker gene expression. Patient cells were annotated using the random forest classifier, using healthy donors as a reference. The high proportion of T cells in Patient 10 was consistent with flow cytometry. **d**, Dot plot shows percent of cells classified as pDCs in each healthy donor and patient sample. High percentage of pDCs in samples with known marrow involvement reflects malignant BPDCN cells. Data is shown for all samples that were analysed by scRNA-seq (n = 6 healthy donors, n = 5 samples without known marrow involvement, and n = 6 samples with marrow involvement). Statistical significance between sample groups is indicated (two-sided Student's t-test). **e**, Scatterplot shows B (minor) allele frequencies of Patient 10 SNVs in relapse bone marrow sample (post allogeneic stem cell transplant, y-axis) and germline sample (x-axis). SNVs homozygous in the host germline sample are indicated in blue (A/A) and green (B/B), heterozygous SNVs are indicated in red (A/B). This analysis allows for the identification of alleles that are specific for host and donor cells (indicated in bold). Thresholds that were used for subsetting these SNVs are indicated. **f**, Scatterplot shows quantification of host- and donor-specific alleles in the scRNA-seq data of Patient 10 relapse bone marrow samples (red). The fraction of SNVs specific to the donor genome is indicated on the x-axis. Genotypes could be assigned for the majority of cells (98.0%), with 62.6% of those annotated as host-derived, and 37.4% annotated as donor-derived. Cells from the diagnostic bone marrow sample (blue), for which no cells were annotated as donor-derived, are shown as comparison. **g**, Barplot shows the proportion of cell types within Patient 10 relapse cells genotyped as host- and donor-derived. Most host cells classify as pDCs, likely reflecting malignant BPDCN cells. **h**, Violin plot shows scores of a published BPDCN gene signature for pDCs from each of the 17 analysed single-cell samples. Related to Fig. 2.

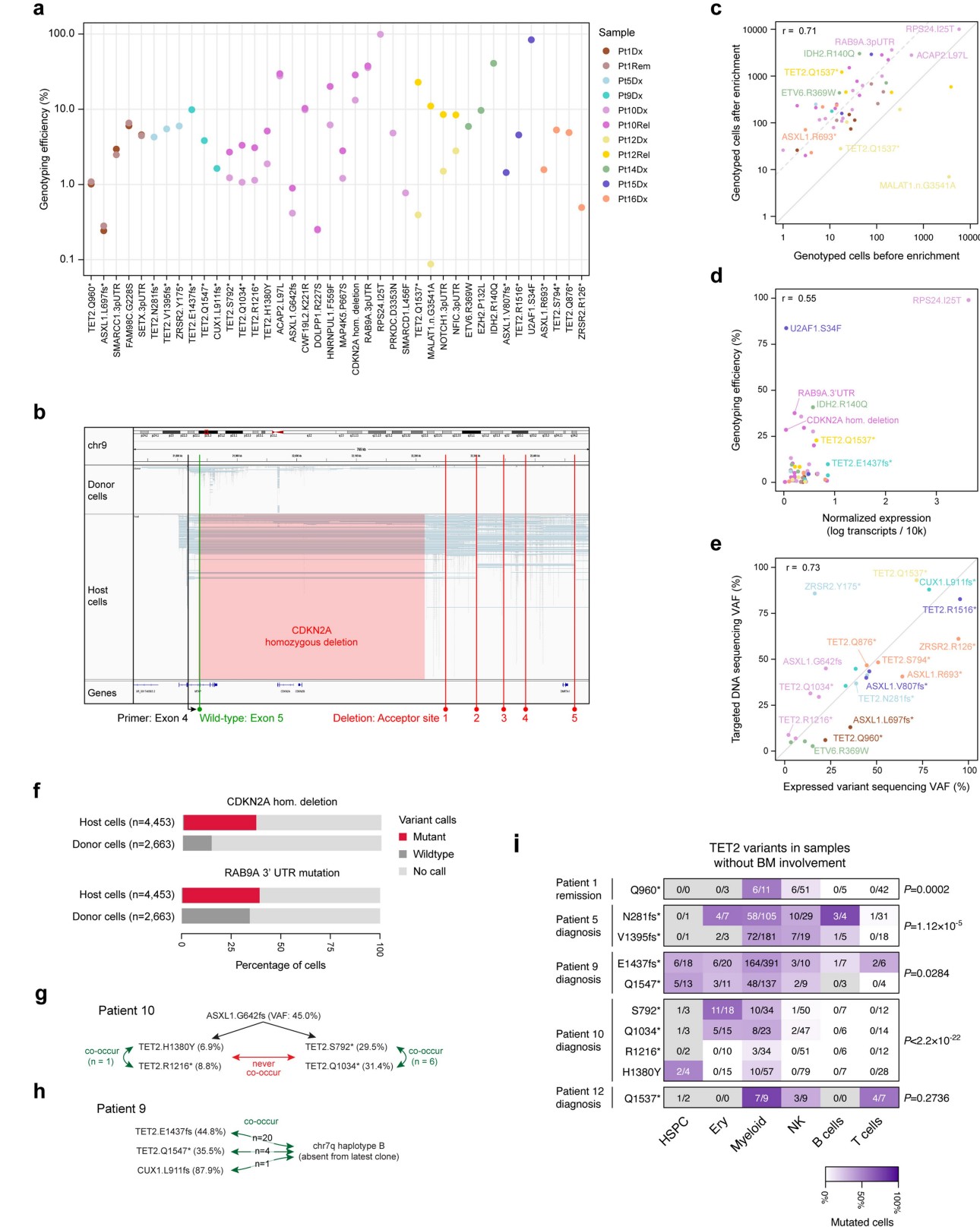

**Extended Data Fig. 6** | See next page for caption.

**Extended Data Fig. 6 | Single-cell genotyping of bone marrow samples.**
**a**, Scatterplot shows overview of the genotyping efficiency of all 40 mutations targeted by XV-seq across 11 samples. Some mutations were analysed in multiple samples collected from the same patient. **b**, Genome plot shows combined 10x scRNA-seq reads for Patient 10 relapse bone marrow donor cells and host cells over the *CDKN2A* gene locus on chromosome 9. The focal homozygous deletion observed in (malignant) host cells results in atypical splicing of the upstream *MTAP* gene to five different acceptor sites downstream of *CDKN2A*. This enabled the generation of a genotyping primer specific to exon 4 of *MTAP* that can be used to detect the *CDKN2A* deletion in single cells. **c**, Scatterplot compares the number of genotyped cells detected in raw scRNA-seq data (x-axis) with the number of genotyped cells detected by XV-seq (y-axis, r = 0.71). Median enrichment across targets is 11.1-fold (indicated by dashed line). These data demonstrate XV-seq target enrichment and the utility of selecting suitable mutations based on raw scRNA-seq data. **d**, Scatterplot shows the genotyping efficiency of XV-seq targets (y-axis) compared to the normalized expression level of the transcript (x-axis, r = 0.55). **e**, Scatterplot shows agreement between VAFs from bulk targeted sequencing using the Rapid Haem Panel (y-axis) and single-cell genotyping using XV-seq (x-axis, r = 0.73). For the latter, the VAF was calculated as the number of mutated transcripts / number of total transcripts captured. **f**, Barplot shows the percentage of cells from the Patient 10 relapse sample (post stem cell transplant) for which genotype information was obtained (Extended Data Fig. 5e–g). *RAB9A* is located on chromosome X (male patient) and *CDKN2A* is located on chromosome 9 of which one copy is lost in addition to the focal deletion of the locus (Extended Data Fig. 3a). The exclusive detection of wild-type and mutated transcripts in the expected cell populations supports accuracy of cell type annotation, host/donor classification, and XV-seq mutation detection. **g**, Illustration of supporting evidence for subclonal structure obtained from single-cell XV-seq of the Patient 10 uninvolved bone marrow. *TET2* mutations S792* and Q1034* co-occur in the same cell (major subclone). Similarly, *TET2* mutations H1216* and H1380Y also co-occur in the same cell (minor subclone). Mutations specific to the two subclones were not detected in the same cell. The sample is karyotypically normal, further supporting the existence of two subclones, as truncating mutations in *TET2* are unlikely to affect the same allele. VAFs from targeted sequencing are indicated between parentheses. **h**, Illustration of supporting evidence for subclonal structure obtained from single-cell XV-seq of the Patient 9 uninvolved bone marrow. This sample is characterized by a subclonal loss of heterozygosity (LOH) on chr7q. Detection of the lost haplotype in n=25 single cells indicates that mutations in *TET2* and *CUX1* occurred before the LOH of chr7q. VAFs from targeted sequencing are indicated between parentheses. The high VAF of the mutation in *CUX1* is explained by its location on chr7q. **i**, Heatmaps show proportion of *TET2*-mutated cells in each major hematopoietic cluster. Ten patient-specific *TET2* mutations were assessed in five marrow samples. *P*-values indicate mutant-cell enrichment in HSPC/Erythroid/Myeloid vs. B/T/NK cells by Pearson's Chi-square test with Yates' correction. Related to Fig. 2.

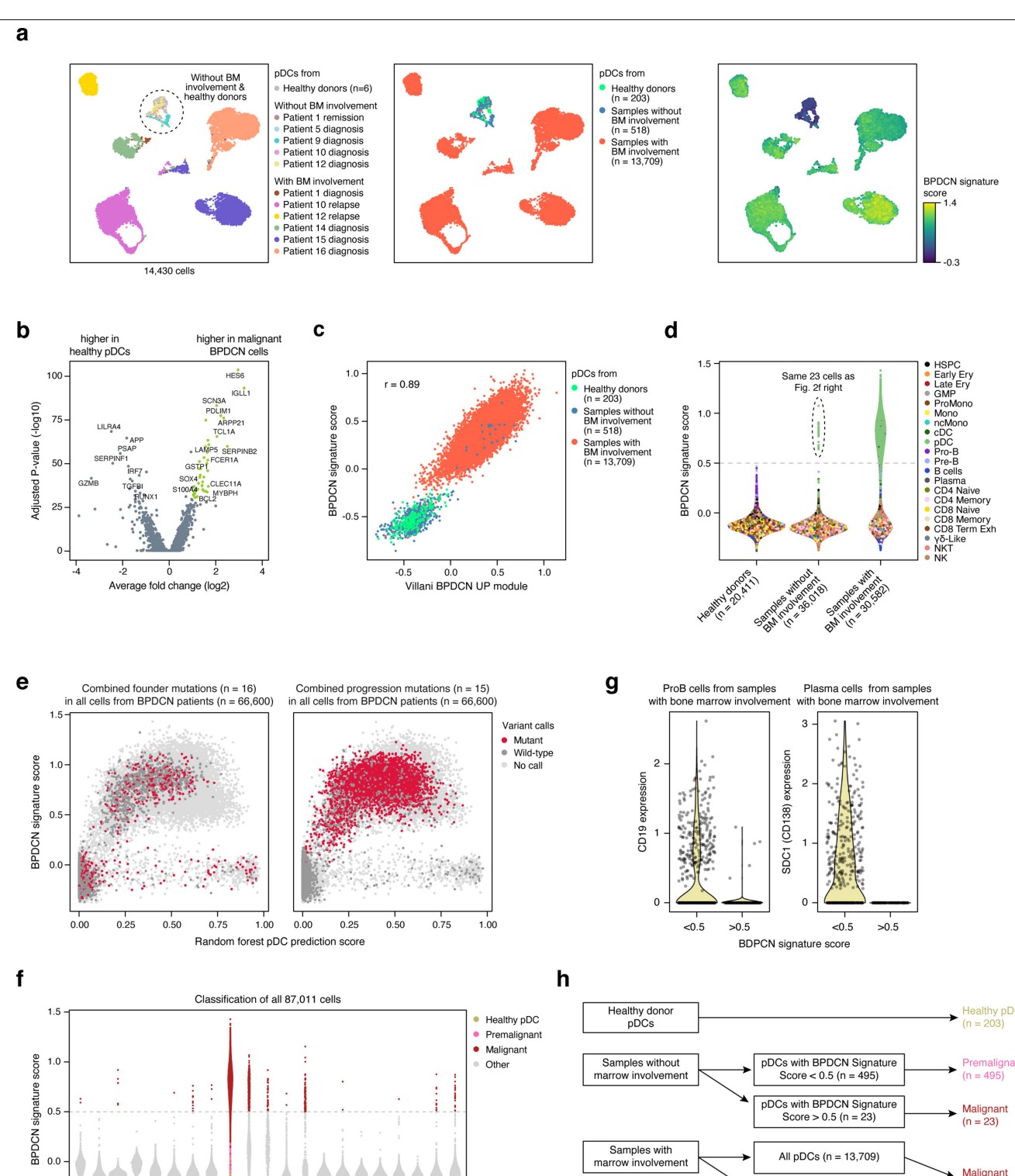

**Extended Data Fig. 7** | See next page for caption.

**Extended Data Fig. 7 | Single cell-derived BPDCN signature enables malignant cell classification. a**, UMAPs show all cells classified as pDCs across the entire scRNA-seq dataset (n = 14,430 cells), coloured according to patient/sample (left), samples with and without known BPDCN involvement (middle), and BPDCN signature score (right). **b**, Volcano plot shows differentially expressed between healthy pDCs and malignant BPDCN cells. We used 45 genes with log2 fold change > 1 and adjusted *P*-value < 1E-30 (green symbols), including *BCL2* and *TCL1A*, to calculate the BPDCN signature score in downstream analyses. All 45 genes are provided in Supplementary Table 3c. *P*-values were calculated using a two-sided Wilcoxon Rank Sum test and adjusted using Bonferroni correction as implemented in the Seurat function FindMarkers. **c**, Scatterplot shows signature scores in all cells that were classified as pDC (n = 14,430). We calculated scores using a previously published BPDCN module[27] (x-axis) and using the 45 signature genes we defined in panel a (y-axis). **d**, Sinaplot shows BPDCN signature scores in all single cells (n = 87,011) that we profiled in this study, split by donor type. The colour indicates cell type classification by the RandomForest algorithm. **e**, Scatterplots show single cells from all patient samples (n = 66,600) according to their random forest pDC prediction score (x-axis) and BPDCN signature score (y-axis). Red dots indicate detection of mutant transcripts (n = 16 founder mutations, left; n = 15 progression mutations, right), grey dots indicate detection of wild-type transcripts. **f**, Sinaplot shows BPDCN signature scores in all single cells (n = 87,011), split by cell type and coloured by final classification. **g**, Violin plots show expression of canonical marker genes in cells that were originally classified as proB cells and plasma cells. Cells with a BPDCN signature score exceeding 0.5 were reclassified as malignant cells. The absence of CD19 in reclassified proB cells and the absence of CD138 in reclassified plasma cells supports our reclassification. **h**, Flow chart illustrates classification of healthy pDCs, premalignant pDCs, and malignant BPDCN cells for single cells across the dataset. Related to Fig. 2.

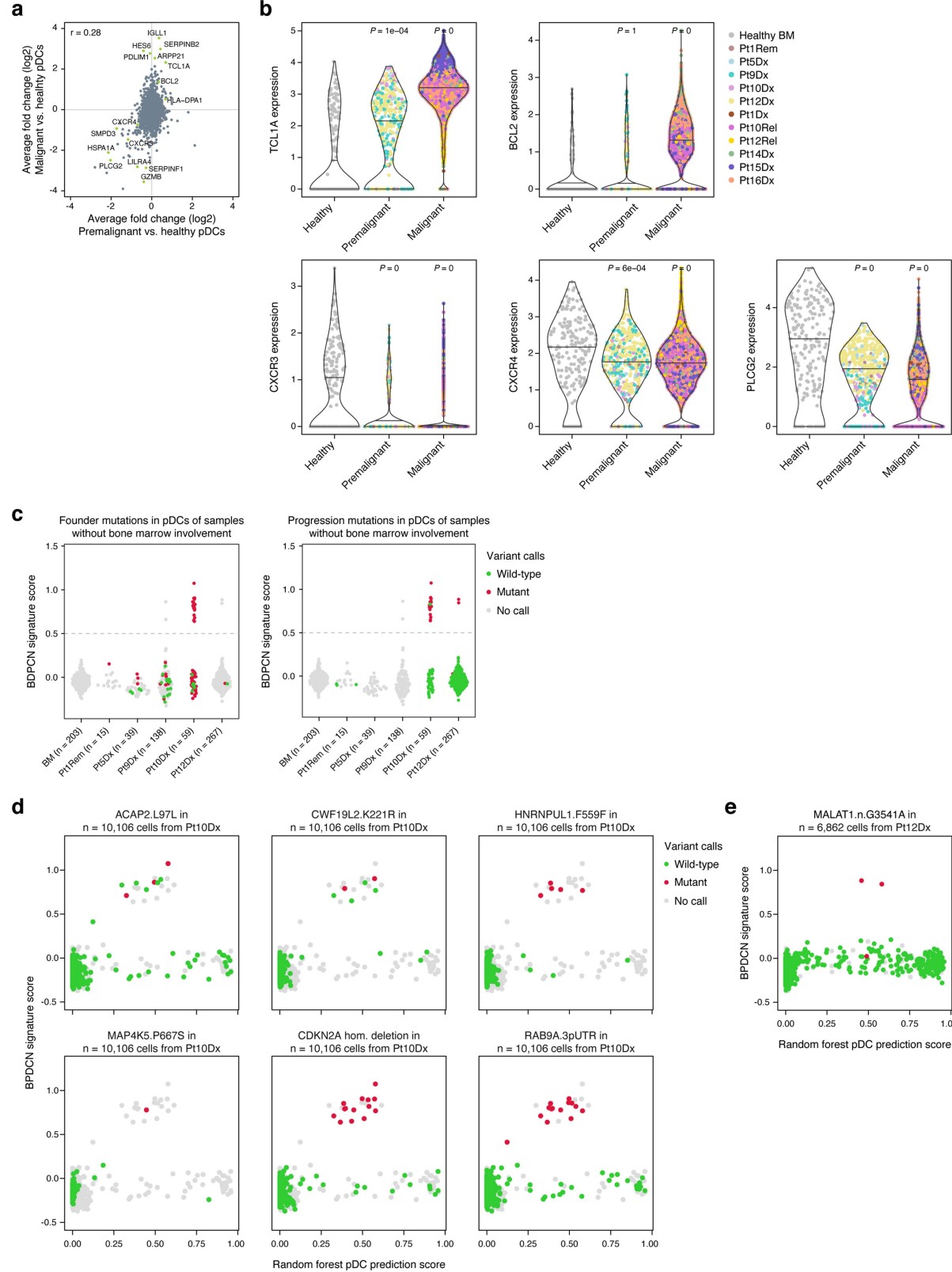

**Extended Data Fig. 8 | See next page for caption.**

**Extended Data Fig. 8 | Gene expression and mutation analysis outlines BPDCN disease progression. a**, Scatterplot shows gene expression fold changes between premalignant vs. healthy pDCs (x-axis) and malignant BPDCN cells vs. healthy pDCs (y-axis). The changes are positively correlated (r = 0.28) and more pronounced for the malignant BPDCN cell comparison. **b**, Violin/sina plots show expression of selected genes that are differentially expressed between healthy pDCs (n = 203), premalignant pDCs (n = 495), and malignant BPDCN cells (n = 14,232). Cells are grouped according to cell annotations defined in Extended Data Fig. 7f. Adjusted *P*-values indicate a comparison to healthy pDCs and were calculated using a two-sided Wilcoxon Rank Sum test and adjusted using Bonferroni correction as implemented in the Seurat function FindAllMarkers. **c**, Sina plots show the BPDCN signature score in all cells classified as pDC in bone marrow samples from healthy donors and patients without involvement. Red dots indicate cells with detection of mutant transcripts, green dots indicate cells with detection of wild-type transcripts (n = 16 founder mutations, left; n = 13 progression mutations, right). Progression mutations were nearly exclusively captured in cells with a BPDCN signature score exceeding 0.5, consistent with the malignant identity of these cells (*P* = 1.1E-84 by Pearson's Chi-square test with Yates' correction). **d**, Scatterplots show single cells from the Patient 10 diagnostic sample (n = 10,106) according to their random forest pDC prediction score (x-axis) and BPDCN signature score (y-axis). Red dots indicate detection of mutant transcripts, green dots indicate detection of wild-type transcripts for six indicated genes. Integrated analysis of gene expression and mutations concertedly identified rare circulating malignant BPDCN cells (n = 19, 0.19%). **e**, Scatterplot shows single cells from the Patient 12 diagnostic sample (n = 6,862) according to their random forest pDC prediction score (x-axis) and BPDCN signature score (y-axis). Red dots indicate detection of mutant transcripts, green dots indicate detection of wild-type transcripts for *MALAT1*. The detection of mutated transcripts in two cells with a high BPDCN signature score supports their malignant identity. Related to Fig. 2.

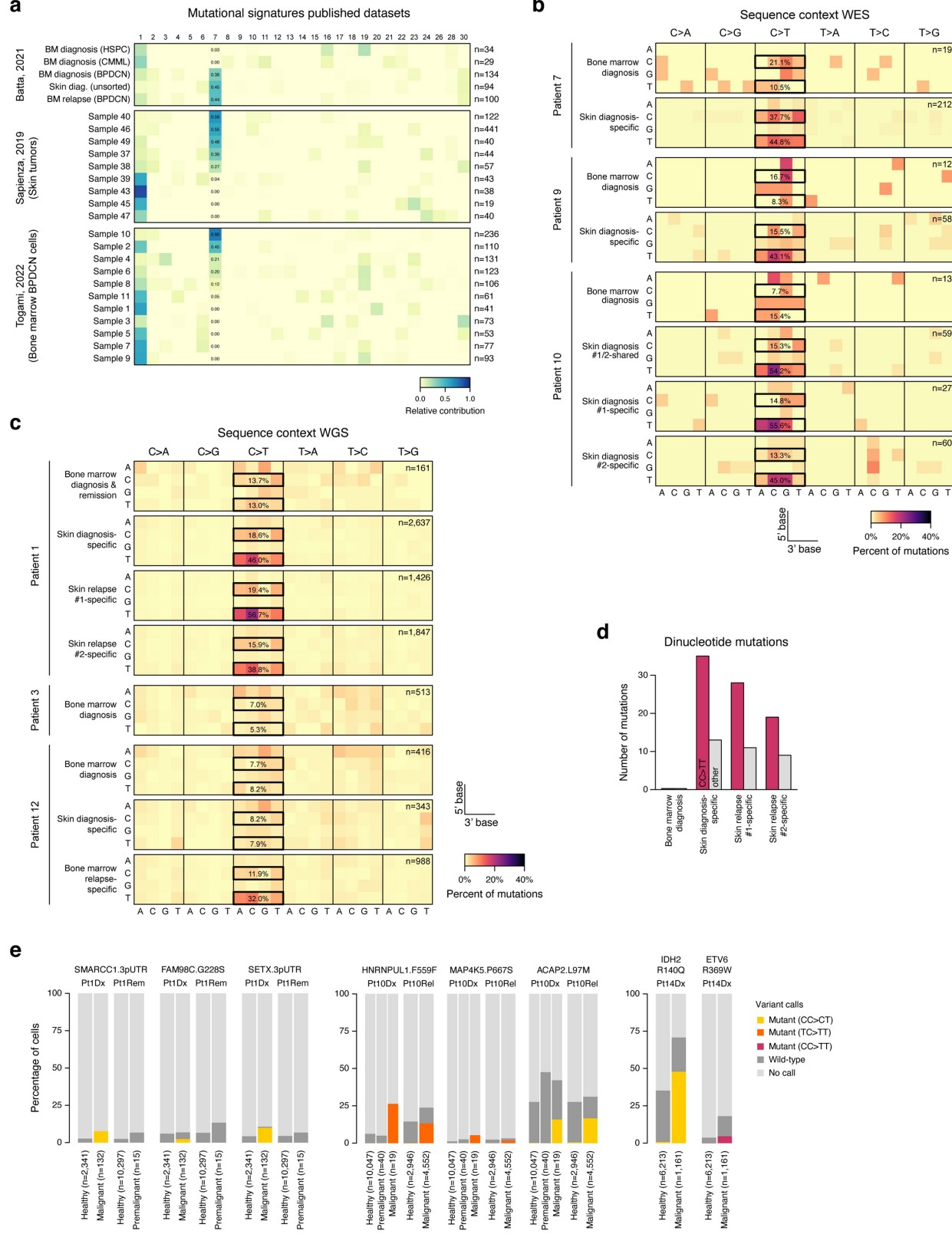

**Extended Data Fig. 9 | See next page for caption.**

**Extended Data Fig. 9 | Mutational signature analysis of BPDCN patient samples. a**, Heatmaps show mutational signature analysis of BPDCN patient samples analysed by whole-exome sequencing from three published datasets[16,17,28]. Blue heat indicates the predicted relative contribution of the mutational signature. Relative contribution of signature 7 (UV damage signature) is indicated. Samples from Batta et al. were generated from indicated bone marrow populations at diagnosis (n = 3) and relapse (n = 1), and a skin tumour sample, all from the same patient. The UV damage signature is detected in all samples containing malignant cells (relative contribution ≥0.38). Samples from Sapienza et al. were generated from skin tumours of nine patients. The UV damage signature is detected in five samples (≥0.27). Samples from Togami et al. were generated from sorted malignant cells of bone marrows from 11 patients. The UV damage signature is detected in four samples (≥0.20). **b**, Heatmaps show sequence context of somatic SNVs detected in samples analysed by whole-exome sequencing. Mutations are only shown in the sample in which they were first detected, and not in subsequent samples. Column headers indicate the base substitution. X- and y-axis labels indicate the bases upstream and downstream of the mutated base, respectively. Red heat indicates the relative contribution of mutations within the given nucleotide context. UV-associated CC > CT and TC > TT mutations are indicated by bold horizontal boxes and their total percentages. **c**, Heatmaps show sequence context of somatic SNVs detected in samples analysed by whole-genome sequencing. Annotations are similar to panel b. **d**, Barplot shows the number of dinucleotide mutations detected in samples from Patient 1, grouped by UV-specific CC > TT and all other dinucleotide mutations. Mutations are only shown in the sample in which they were first detected. **e**, Barplots indicate the percentage of cells for which wild-type or mutant transcript were detected using XV-seq. Selected UV-associated (CC > CT and TC > TT) as well as UV-specific (CC > TT, ETV6) progression mutations are shown. Cells are grouped according to cell annotations defined in Extended Data Fig. 7f. Notably, in the diagnostic sample of Patient 10 (without known marrow involvement), UV-associated mutations are specifically detected in rare malignant cells (n = 19), consistent with our model of retrograde dissemination. Related to Fig. 3.

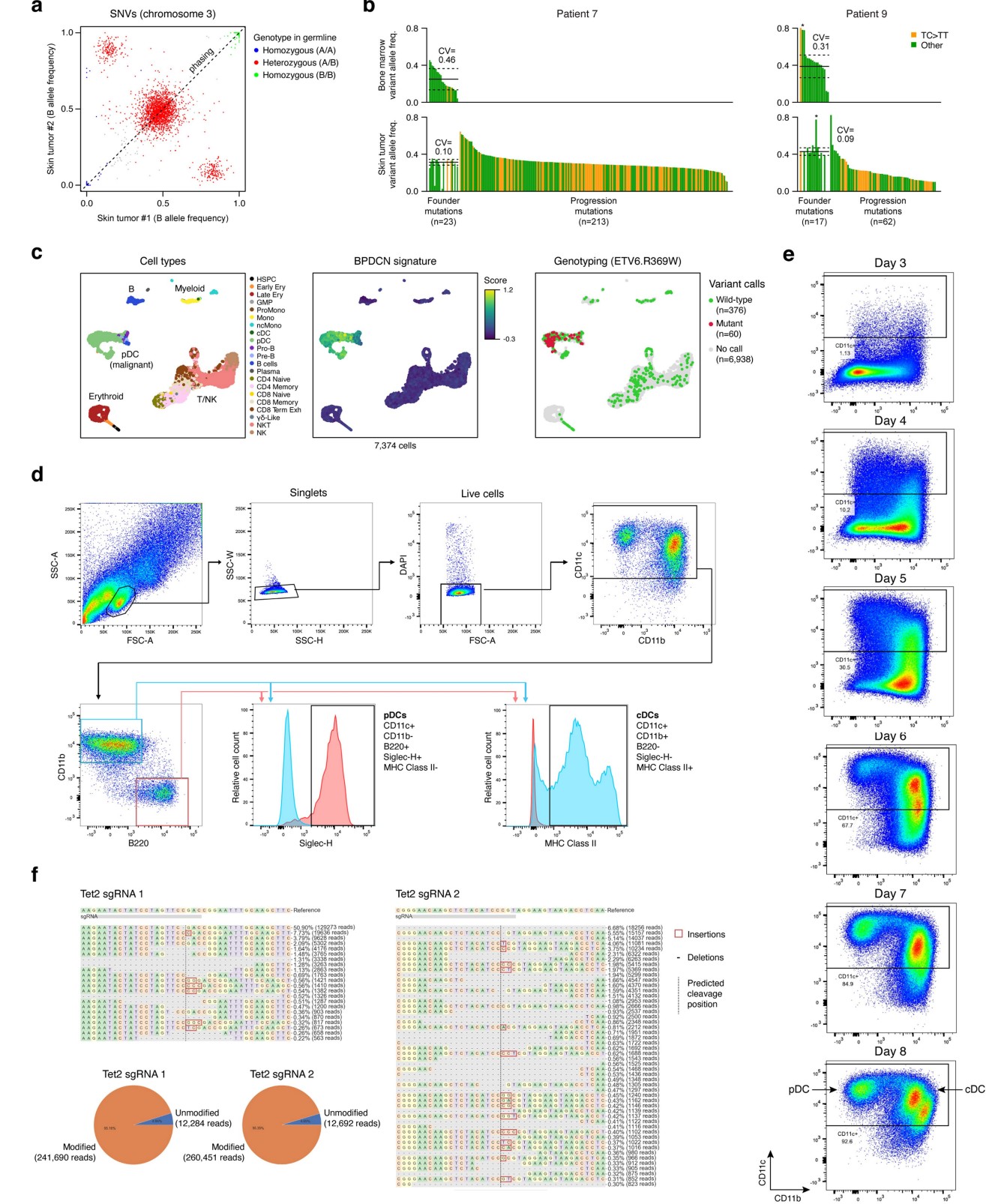

**Extended Data Fig. 10** | See next page for caption.

**Extended Data Fig. 10 | Order of acquisition of UV damage and functional evaluation in the HOXB8 model of pDC differentiation. a**, Scatterplot shows B (minor) allele frequencies of SNVs located on chromosome 3 in skin tumour #1 and #2 from Patient 10. SNVs are coloured according to their genotype in the matching germline sample. Loss of a region on chr3p (Extended Data Fig. 3a) affects different alleles, indicating separate events and convergent evolution in both skin tumours samples. Heterozygous SNVs were phased along the diagonal (shown in Fig. 4d). **b**, Barplots show VAFs for founder and progression mutations detected in bone marrow samples (top) and matched skin tumour samples (bottom) from Patients 7 ans 9. UV-associated TC > TT mutations are indicated in orange, other mutations in green. Average VAF (solid line) and coefficient of variation (CV, standard deviation divided by mean, dotted lines) of founder mutations are indicated. Asterisks indicate mutations that are affected by copy-number alterations (Patient 9) or are likely contaminants from blood (Patient 7) and are not included in this calculation. **c**, UMAPs of scRNA-seq performed on bone marrow cells (n=7,374 cells) from Patient 14. Cell type clusters (left), BPDCN signature scores (middle), and single cell genotyping results for the UV-specific (CC>TT) ETV6.R369W mutation (right) are shown. **d**, Flow cytometry gating strategy for the identification of pDCs (CD11c+/CD11b-/B220+) and cDCs (CD11c+/CD11b+/B220-) from representative HOXB8 cultures. **e**, Time course showing the dynamics of representative pDC and cDC differentiation from day 3 to day 8 following oestrogen withdrawal in the HOXB8 system. pDC and cDC populations become distinguishable on days 6-8. **f**, Amplicon sequencing showing successful CRISPR/Cas9 editing of *Tet2* in HOXB8 cells using two different gRNAs. Greater than 95% of reads for each guide are edited, including insertions (red boxes) and deletions (dashes) predicted to cause frameshift alterations. The predicted Cas9 cleavage site 3 nucleotides upstream from the NGG PAM site is indicated with a vertical dashed line. Related to Fig. 4.

# Reporting Summary

Nature Research wishes to improve the reproducibility of the work that we publish. This form provides structure for consistency and transparency in reporting. For further information on Nature Research policies, see our Editorial Policies and the Editorial Policy Checklist.

## Statistics

For all statistical analyses, confirm that the following items are present in the figure legend, table legend, main text, or Methods section.

| n/a | Confirmed | |
|---|---|---|
| ☐ | ☒ | The exact sample size (*n*) for each experimental group/condition, given as a discrete number and unit of measurement |
| ☐ | ☒ | A statement on whether measurements were taken from distinct samples or whether the same sample was measured repeatedly |
| ☐ | ☒ | The statistical test(s) used AND whether they are one- or two-sided *Only common tests should be described solely by name; describe more complex techniques in the Methods section.* |
| ☐ | ☒ | A description of all covariates tested |
| ☐ | ☒ | A description of any assumptions or corrections, such as tests of normality and adjustment for multiple comparisons |
| ☐ | ☒ | A full description of the statistical parameters including central tendency (e.g. means) or other basic estimates (e.g. regression coefficient) AND variation (e.g. standard deviation) or associated estimates of uncertainty (e.g. confidence intervals) |
| ☐ | ☒ | For null hypothesis testing, the test statistic (e.g. *F*, *t*, *r*) with confidence intervals, effect sizes, degrees of freedom and *P* value noted *Give P values as exact values whenever suitable.* |
| ☒ | ☐ | For Bayesian analysis, information on the choice of priors and Markov chain Monte Carlo settings |
| ☒ | ☐ | For hierarchical and complex designs, identification of the appropriate level for tests and full reporting of outcomes |
| ☐ | ☒ | Estimates of effect sizes (e.g. Cohen's *d*, Pearson's *r*), indicating how they were calculated |

*Our web collection on statistics for biologists contains articles on many of the points above.*

## Software and code

Policy information about availability of computer code

| Data collection | No software was used |
|---|---|
| Data analysis | All software and code are listed in the Methods section, including: BWA version 0.7.15; Picard version 2.5.0; GATK version version 4.0.0.0; Mutect2; MutationalPatterns; STAR version 2.6.0c; cellranger version 7.0.0; bcftools version 1.10.2; samtools mpileup; data.table; IronThrone-GoT version 2.1; Seurat version 4.1.1; Harmony; randomForest version 4.7-1.1; GeneSetEnrichmentAnalysis Molecular Signatures Database version 7.1; Custom R Scripts written for the analysis of scRNA-seq data will be on Github (https://github.com/petervangalen/Single-cell_BPDCN/) and and as DOI: 10.5281/zenodo.7746255 (https://zenodo.org/record/7746255#.ZBSOti-B0eY). |

For manuscripts utilizing custom algorithms or software that are central to the research but not yet described in published literature, software must be made available to editors and reviewers. We strongly encourage code deposition in a community repository (e.g. GitHub). See the Nature Research guidelines for submitting code & software for further information.

## Data

Policy information about availability of data

All manuscripts must include a data availability statement. This statement should provide the following information, where applicable:
- Accession codes, unique identifiers, or web links for publicly available datasets
- A list of figures that have associated raw data
- A description of any restrictions on data availability

The WES/WGS data are available in dbGaP (https://dbgap.ncbi.nlm.nih.gov/), accession number phs003228.v1.p1. For single-cell analyses, sequencing data and gene expression matrices are available in the Gene Expression Omnibus (https://www.ncbi.nlm.nih.gov/geo/), accession number GSE227690. Human genome reference hg19 is available at https://www.ncbi.nlm.nih.gov/data-hub/genome/GCF_000001405.13/.

# Field-specific reporting

Please select the one below that is the best fit for your research. If you are not sure, read the appropriate sections before making your selection.

☒ Life sciences ☐ Behavioural & social sciences ☐ Ecological, evolutionary & environmental sciences

For a reference copy of the document with all sections, see nature.com/documents/nr-reporting-summary-flat.pdf

# Life sciences study design

All studies must disclose on these points even when the disclosure is negative.

| | |
|---|---|
| Sample size | Sample size was based on tissue availability in this rare cancer. |
| Data exclusions | No data were excluded. |
| Replication | DNA mutations were confirmed by orthogonal methods on the same samples: e.g., exome capture and targeted capture/PCR amplification-based sequencing. Laboratory data were generated in at least two biologically independent experiments, each with at least three replicates. |
| Randomization | Randomization not relevant to this study as all data are sequencing-based approaches to human cancer tissue samples. Participants were known to have a diagnosis of this rare leukemia and consented to have their tumor tissue collected and sequenced. |
| Blinding | No blinding was performed. Sequencing was on samples from patients with known diagnoses of BPDCN or from normal healthy donors whose status was known to the research team. In laboratory experiments, the researchers were required to generate experimental models and conditions and were therefore aware of the status of each. Pathologists who evaluated the tissue samples were blinded to whether the patient had known skin and/or bone marrow involvement with BPDCN prior to seeing the tissue sections. |

# Reporting for specific materials, systems and methods

We require information from authors about some types of materials, experimental systems and methods used in many studies. Here, indicate whether each material, system or method listed is relevant to your study. If you are not sure if a list item applies to your research, read the appropriate section before selecting a response.

## Materials & experimental systems

| n/a | Involved in the study |
|---|---|
| ☐ | ☒ Antibodies |
| ☒ | ☐ Eukaryotic cell lines |
| ☒ | ☐ Palaeontology and archaeology |
| ☐ | ☒ Animals and other organisms |
| ☐ | ☒ Human research participants |
| ☒ | ☐ Clinical data |
| ☒ | ☐ Dual use research of concern |

## Methods

| n/a | Involved in the study |
|---|---|
| ☒ | ☐ ChIP-seq |
| ☒ | ☐ Flow cytometry |
| ☒ | ☐ MRI-based neuroimaging |

## Antibodies

| | |
|---|---|
| Antibodies used | CD11b Alexa Fluor 700 BioLegend 101222 M1/70 B336447;<br>CD11c PE/Cyanine7 BioLegend 117317 N418 B346714;<br>B220 APC/Cyanine7 BioLegend 103224 RA3-6B2 B321245;<br>Siglec-H PE BioLegend 129605 551 B248184;<br>MHC Class II PerCP/Cyanine5.5 BioLegend 107626 M5/114.15.2 B269461 |
| Validation | Examples per manufacturer's websites of flow cytometry in mouse cells. |

## Animals and other organisms

Policy information about studies involving animals; ARRIVE guidelines recommended for reporting animal research

| | |
|---|---|
| Laboratory animals | wild-type C57BL/6 mice, male consistent with extreme male bias of BPDCN |
| Wild animals | Provide details on animals observed in or captured in the field; report species, sex and age where possible. Describe how animals were caught and transported and what happened to captive animals after the study (if killed, explain why and describe method; if released, say where and when) OR state that the study did not involve wild animals. |

| Field-collected samples | *For laboratory work with field-collected samples, describe all relevant parameters such as housing, maintenance, temperature, photoperiod and end-of-experiment protocol OR state that the study did not involve samples collected from the field.* |
| --- | --- |
| Ethics oversight | *Identify the organization(s) that approved or provided guidance on the study protocol, OR state that no ethical approval or guidance was required and explain why not.* |

Note that full information on the approval of the study protocol must also be provided in the manuscript.

# Human research participants

Policy information about <u>studies involving human research participants</u>

| Population characteristics | Population is patients age 18 and over with BPDCN who presented to the Dana-Farber Cancer Institute for treatment and consented to tissue banking for research. All patients in this population were offered the choice to consent to a sample banking protocol if they chose. Healthy controls were volunteer donors consented to tissue banking protocols at Brigham and Women's Hospital. |
| --- | --- |
| Recruitment | Participants were not recruited. All patients with blood cancers at Dana-Farber Cancer Institute are offered participation in an excess tissue sample banking protocol for de-identified research. The research team is not aware of any additional self-selection biases that may have influenced the population, and it is generally in keeping with the total population of patients with BPDCN. Normal healthy donors were volunteer bone marrow donors age 18 and over who elected to participate in marrow donor registries without compensation. There are no known biases in the selection of the normal donor population. |
| Ethics oversight | Dana-Farber Cancer Institute and Brigham and Women's Hospital Institutional Review Boards (IRB) approved all studies. |

Note that full information on the approval of the study protocol must also be provided in the manuscript.

