## [Peer Review File · Nature]

Manuscript Title: Ultraviolet radiation shapes dendritic cell leukemia transformation in the skin

Reviewer Comments & Author Rebuttals

Reviewer Reports on the Initial Version:

Referee expertise:

Referee #1: single cell analysis, leukemia, clonal haematopoiesis

Referee #2: clonal haematopoiesis

Referee #3: cancer genomics, evolution, leukemia

Referee #4: dendritic cell neoplasms

Referees' comments:

Referee #1 (Remarks to the Author):

Griffin et al report sequencing of a rare form of leukemia, where they compare skin lesions with BM samples from the same individuals. They show that BPDCN is often accompanied by large clones in the BM that are ancestral to the BPDCN clones. They also show that the BPDCN clones have a UV signature supporting a cell of origin for transformation in the skin.

I found the paper to be clear, technically sound and well written. Furthermore, I fully agree with the notion that we should study rare cancers. However, I am not sure I grasp the clear insights here that offer a new approach to treating this leukemia. Instead, the authors position this paper as providing insight into clonal hematopoiesis (CH). Of note, according to the introduction they seem to refer to CH as the entity described by Jaisawal et al and Genovese et al, which reflect hematopoietic clonal growth without any bone marrow or blood abnormality. This CH framing is apparent for example in the manuscript title where BPDCN is not included, as well as several introduction and discussion points. These points may not be sufficiently supported by data. It is unclear what data is provided to support amending the "seed and soil" paradigm or to support tissue specific oncogenes. The paper does not address mechanisms of tissue tropism or local selective pressures, and it is unclear what is the relevance to CH and cardiovascular risk. I am also not sure I see how peripheral tissues shape the evolution of CH (other than the UV signature supporting a pDC cell of origin in the skin). While I understand that the ability to compare skin to BM is of interest (to address "distinct stages of disease evolution"), it is unclear how different it would be from the growing body of knowledge comparing CH clones to later arising common myeloid neoplasms, CH clones that remain after treatment for myeloid neoplasms, or CH co-occurring with myeloid neoplasms.

Furthermore, it is unclear to what degree the BM abnormalities in BPDCNs are representative of CH. BPDCNs are known to commonly have concurrent neoplastic BM disorders (MDS/CMML and AML), and a predilection for TET2 mutations (see references below). Many of the patients included here have significant cytopenia further supporting a concurrent bone marrow dysfunction. While the BM is noted to be uninvolved with BPDCN, it is unclear whether a close evaluation for dysplastic features was undertaken (especially for the more genetically aberrant/high VAF samples). The genetic information is also suggestive of a process that is further along than typical CH. The VAFs are high (often close to the entire sample), there are multiple gene mutations within the same clone and in 1 of 3 cases studied for CNVs also a large (chromosome arm) LOH event. Together with the cytopenia, these presentations may be more consistent with a concurrent MDS pattern than a functionally normal marrow as in CH. While cytopenia are often found in BPDCN and may be related to other causes such as splenomegaly, the conjunction of a genetically abnormal marrow may suggest bone marrow dysfunction at least in some cases. To my mind, this report is similar to, for example, a recent paper by Cohen Aubart et al (Blood, 2021) which looked at another rare clonal disorder (Erdheim-Chester disease) of myeloid origin, known to have significant overlap with myeloid neoplasms, which was found to harbor large TET2 mutated BM clones. Overall, I am not sure the broad claims about CH are justifiable given that BPDCN is a rare disorder that may not represent the majority of CH. Even so, the observation that myeloid malignancies often arise against a backdrop of a parent CH clone is well accepted, and therefore it is unclear how this current work adds to our understanding of this process.

Please find additional comments detailed below regarding the strength of the data supporting some of the claims:

1. The authors elected to perform the bulk of the sequencing with a targeted panel. The panels cited in the methods section seem to be quite small (19 genes for BM samples and 8 genes for skin samples, although this is in conflict with the main text indicating a 95 gene panel). Given the small number of cases and the rarity of the disease, I wonder whether WES of the entire cohort may be warranted. The WES is central to this paper's major novel claim (UV signature) as well as to the understanding of to what degree is the BM aberrant genetically. Standard WES is likely sufficient for most patients in terms of sequencing depth considering the high VAFs in the BM.
2. The enrichment in BPDCN in TET2, ASXL1 and RNA splice factor mutations is well known. I am not sure the data here on this small cohort is sufficient to support claim that "common age-related mutations could predispose to dendritic cell transformation". This needs a more careful treatment using sequential data, population studies or some form of control. Given how rare BPDCN is and how common CH mutations are, creating this linkage may be premature and may have untoward consequences.
3. The single cell data involves only two individuals, which is a weakness. The main claim that is made with regards to these data is that mutated cells are observed throughout most of the cell populations in the bone marrow. This is to be expected considering that the VAFs in bulk DNA sequencing of these mutations would support a near homogenous involvement in the bone marrow as depicted in the Fish plots in Fig 3. In fact, indicated wildtype cells are likely largely false negatives. For example patient 9, should have 90% mutated cells based on bulk DNA VAF. Given this study

design, the single-cell data add relatively limited new information. This may be further compounded by the low efficiency of genotyping (~7% summing information from all eight targeted loci in Fig 2c-d). The authors also make a claim about exclusion of mutated cells in mature T and B cells. This is often supported by only a few cells (<10 cells), without statistical evaluation. These data are interpreted as suggesting differentiation biases. Considering that lymphocytes may be long lived (especially as most of the cohort consists of elderly individuals), this may reflect different cellular life spans rather than a differentiation bias. Evaluating mutation frequencies in lymphoid progenitors may be more informative to differentiation biases.

4. The frequency of observing wildtype alleles is somewhat concerning (light gray color choice is a bit tricky for visualization). Considering the fish plots in Fig 3, and bulk VAF data, one would expect the large majority of TET2 transcripts to be mutated in these two samples. For example, for Patient 9, the bulk DNA VAF suggest that 70% of cells are compound heterozygotes with an even higher number for patient 10. In part this could reflect partial sampling of the alleles, and in part can be due to the fact that transcripts labeled as wildtype, actually contain a variant in a different locus on the same transcript. Analysis that takes into account clonal phasing may be more informative. In addition, direct examination of potential confounders such as difference of expression levels in different cell populations may be warranted. The ASXL1 variants may occur in a repetitive sequence which is prone for high rate of PCR artifacts. More broadly, the authors present XV-seq as an enhanced method. However, I am not sure this dataset makes the strongest case to that point. Genotyping aggregated across 3-5 targets/sample in Fig 2 only provided info for ~7% of cells. This seems to be a worse performance than the author's previous report. Given that the authors only assess mutation frequency in relation to broad cell types (HSPC, ery, myeloid, B and T cells as in panel e), I wonder whether an analysis as in Miles et al, 2020 with the tapestri platform + oligo-labeled antibodies may be a more appropriate method. Alternatively, a fairly straightforward sorting experiment, followed by targeted DNA sequencing may more readily address the question in Fig 2 without the technical confounders related to capture and target expression.

5. The UV signature is perhaps the most novel aspect of this study, as it links the cell of origin to a skin pDC. Of note, mutation rates are still lower than those seen in skin cancer, which may reflect shorter time in the skin pre-transformation. Perhaps expanding this experiment with WES beyond three individuals may help strengthen this claim. Furthermore, Given the paucity of available data, the authors may consider re-analyzing published data for UV signature (Menezes et al, Leukemia, 2014 [including exomes], Alayed et al, Am J Hematol, 2013 [including BM BPDCN], Stenzinger et al, Oncotarget, 2014, Beird et al, Blood Cancer Journal, 2019). I also note that signature decomposition may benefit from a confidence estimate. For example, the signature decomposition of a handful of mutations in the BM samples may not be very telling. This may also explain the strong MMR/MSI signature in the BM samples that is probably an artefact here given the low mutation number.

6. The authors show in Fig 4 that the single cell data identified 19 malignant cells in a pre-treatment BM sample. They argue for this as a potential novel diagnostic tool and an "important step towards single-cell multi-omics for diagnostic use in early detection, circulating tumor cell identification, or measurable residual disease (MRD) evaluation." I find this to be a fairly strong claim based on 19 cells within one individual. Any classifier is prone to over-fitting without proof of generalizability in independent samples. Even within this sample, the 19 cells are compared with only 39 normal pDCs.

It is also unclear what significance this would have in clinical management.

Referee #2 (Remarks to the Author):

In this manuscript from Griffin and colleagues, the authors seek to use a study of blastic plasmacytoid dendritic cell neoplasm (BPDCN) as a model for understanding evolution of a cancer from a pre-malignant state. The authors initially show that uninvolved bone marrow in patients with BPDCN has frequent clonal hematopoietic mutations with high variant allele frequencies. This finding is supported by other case reports in the literature. The authors then perform single cell RNA sequencing and show normal hematopoiesis and use a modified protocol for somatic mutation assessment, as this group and others have reported, to show that cells across the hematopoietic hierarchy harbor the clonal driver mutations. Next, the authors conduct exome and targeted sequencing of bone marrow and matched skin tumors. This reveals a higher burden of mutations in the skin tumors compared to the bone marrow samples and also reveals a signature associated with UV-induced mutagenesis in the BPDCN samples. Finally, the authors use the XV-seq method to try to track the origins of a bone marrow relapse in a patient with BPDCN and suggest that the origin may be from a transformed pDC-like cell.

This is an interesting manuscript that explores an important issue in the field of cancer biology, which is to define the evolution of a malignancy. Unfortunately, many of the conclusions made in this work are not entirely supported by the data and the authors fail to consider alternative possibilities underlying some of the observations. I will provide a few examples here:

- On pages 4-5, the authors use exome sequencing on paired samples and state, "malignant BPDCN skin tumors harbored a much higher overall burden of mutations (range 74-229), only a minority of which were found in matched bone marrows ... These findings confirm a direct clonal relationship between CH in the bone marrow and malignant BPDCN cells in the skin." This conclusion seems inappropriate given the evidence that is presented. The authors are comparing bone marrow with a variety of clones present to a skin BPDCN tumor that has a clonal origin. Exome and targeted sequencing both have detection limitations. How can the authors be sure that the observations are not simply attributable to the bottleneck present in the skin tumor emerging from a clone that has more readily detectable mutations?

- The above issue is also present to the analysis of UV induced mutations. This is interesting, but is this really causal in "shaping the evolution" as the authors suggest? The clonal cells could simply arrive in the skin and be subject to UV induced mutations that can be detected at higher VAFs, due to the clonal nature of BPDCNs.

- On page 7, the authors use XV-seq and show that tumor specific mutations were abundant in relapse pDC cells, but absent in hematopoietic compartments in the background. They then

conclude that the data confirms "the retrograde pathway of tumor progression." I am not sure that this statement can be so conclusively made using the somatic mutations that were profiled.

The authors have conducted an in depth and important single cell analysis of BPDCN here. This is certainly worth reporting to those who study this rare cancer in the hematology community. My concern is that the authors attempt to draw broad and sweeping conclusions about clonal evolution in cancer and how tissue microenvironments may be involved that are unsupported by the evidence shown. In addition, while interesting approaches like XV-seq are reported, these are similar to methods and studies that have already been described by this group and others (references 16, 17, and 44).

Referee #3 (Remarks to the Author):

Griffin et al, investigate the relationship between clonal hematopoiesis (CH) and blastic plasmacytoid dendritic cell neoplasm (BPDCN), a rare form of acute leukemia that often presents with malignant cells isolated to the skin. Using samples from a very unique cohort of 12 BPDCN patients the investigators study patterns of tumor phylogenies that underpin progression of CH to BPDCN, and how these are further represented in patients with subsequent bone marrow involvement. Additionally, using supervised RNA-seq classification frameworks the authors study cell type representation across stages of the disease (pre leukemic, transformation, disease progression). Overall this is an elegant and thought provoking study, that provides novel insights of the molecular underpinnings of BPDCN. The manuscript is a bit challenging to read at times, and given the density of the data and methods could benefit from streamlining the narrative.

The observations made are very interesting, the analysis approach is novel and the data analysis and interpretation is sound. The manuscript reveals intriguing evolutionary trajectories underlying BPDCN pathogenesis, informed by elegant analysis of scRNA, and molecular profiling of spatially and temporally separated specimens. The incorporation of mutation signature analysis and clonal reconstruction offers robust evidence for the timing and directionality of clonal dissemination during BPDCN transformation, and relapse.

1. The use case in this manuscript, which is CH to BPDCN reflects a very unique and rare in its clinical presentation disease entity. In contrast the title and abstract of the study is rather broad and suggestive that this observation (peripheral tissue selection of pre-malignant cells) be generalized. It may be more appropriate to align the title, abstract and discussion of the manuscript to the focus of the study.

2. The authors use the term pervasive CH following the observation of high VAF clones in patients with BPDCN. Not clear what the term pervasive eludes to as a function of CH, particularly in the context of BPDCN.

3. The observation of bi-allelic hits in TET2 is rather interesting and potentially novel. There seems to be enrichment of bi-allelic hits, which are mediated by two mutations or a mutation and an allelic

loss. Comparison with publicly available CH or AML datasets could verify this and provide potential insights on the implications of bi-allelic inactivation of TET2 in CH progression.

4. The data in Supplementary Table 1b are rather intriguing. Comparison of molecular findings in the bone marrow at diagnosis, relative to the skin and bone marrow at follow up reveals complex branching phylogenies with clones that are shared in the bone marrow, skin and follow up bone marrow samples, emerging subclones in the skin as well as evidence of clones that are confined in the bone marrow but not involved in transformation. This intriguing clonal structure is not formally presented in the main text. The manuscript could benefit by a more detailed and visual representation of these results.

5. With regards to cell type annotation from RNA-seq, the authors first perform a manual annotation of the healthy donors cells using a select list of gene markers. Then the authors train a RF classifier which takes as input the expression of the cell-type specific genes and outputs a probability of cell type assignment. This model is subsequently applied on the BM negative cells to assign each cell to the type where the probability of assignment is higher. The model is trained and applied in diverse cell types (negative BM cells, host cells after transplant) assuming that train and test data derive from the same underlying data distribution. However, output probabilities are more or less interpreted as similarity scores (i.e. pDC-like). Given the assumption that the cell type specific gene expression patterns are invariant on the condition, can the authors comment on the choice to use the classifier over a supervised classification informed by cell-type specific gene markers? Did the authors evaluate other classification approaches over the RF?

6. In relation to Figure 2b in the methods the authors mention that they present projection of patient cells into the UMAP of healthy donor cells was done by plotting each patient cell at the coordinates of the normal cell with the highest prediction score correlation. Therefore the cells shown are from the healthy donors and not of patients 9, 10.

7. Figure 3c might benefit from multi-sample clonality analysis

8. The authors use mutation signatures as barcodes to elegantly demonstrate that BPDCN tumors arise from a CH-derived clone in the skin, which accumulates UV-induced DNA damage during malignant progression. This analysis is further used to evaluate whether the cells that initiate disease relapse in the bone marrow and skin were there prior to initial therapy and cell stem transplantation. It is not clear however, whether the exposure to UV light provides sufficient evidence of a “tissue specific selective pressure” that shapes evolution of pre-malignant clones to cancer. Perhaps the authors can elaborate on this in their discussion.

9. The evaluation of scRNA seq data as a potential mechanism to detect rare skin-derived circulating tumor cells early is intriguing. How generalizable was this observation from the sc-RNA seq data? Did the authors evaluate samples from other patients? Was the identification of cells in patient 10 related to the patients BM involvement? The authors should highlight that the detection of gene-expression signatures could be further explored, however the validation of UV-induced mutations

required a priori knowledge of the mutations from the diagnostic specimen and would thus be less useful clinically.

Referee #4 (Remarks to the Author):

The authors demonstrate that BPDCN patients with skin involvement exhibit clonal hematopoiesis in their bone marrow and use this opportunity to evaluate premalignant to malignant evolution across different anatomic sites.

They use the natural history of BPDCN in the marrow and the skin to establish that tissue-specific selective pressures can shape the evolution of premalignant clones. They also illustrate the role of ultraviolet (UV) light-induced DNA damages acquired in the cutaneous site on some deteriorous evolution in advanced diseases.

They clearly explain the crucial role of clonal hematopoiesis as a model of cancer development that can be applied to various pathologic situations in oncologic and non-oncologic diseases. Authors also highlighted the importance of the local tissue-specific selective pressures on the pathologic development on the cellular level.

This work is original because BPDCN is a rare disease which is now well described but the use of deep sequencing of eXpressed Variants (XVseq) in marrow and skin tissues offers a unique opportunity to describe and analyse all the malignant process. Analyses and description are very pertinent and particularly well described.

The presentation of data and the methodology description is complete and appropriately presented in extended and supplemental data sets. The statistical analyses are extensively described without any specific problem.

Authors described and discuss their hypotheses with extensive and appropriate references. The authors have already published a lot of original data on this rare disease that are exploited here.

The manuscript is well construct with a very clear abstract that summarized key features of the study that are also clearly explained in the introduction and conclusion section.

This work largely merits publication. Due to the high quality of this manuscript I think it can be published with no further revision.

Prof. Eric DECONINCK

Author Rebuttals to Initial Comments:

Response to Referees' comments:

Referee #1 (Remarks to the Author):

Griffin et al report sequencing of a rare form of leukemia, where they compare skin lesions with BM samples from the same individuals. They show that BPDCN is often accompanied by large clones in the BM that are ancestral to the BPDCN clones. They also show that the BPDCN clones have a UV signature supporting a cell of origin for transformation in the skin.

I found the paper to be clear, technically sound and well written. Furthermore, I fully agree with the notion that we should study rare cancers. However, I am not sure I grasp the clear insights here that offer a new approach to treating this leukemia. Instead, the authors position this paper as providing insight into clonal hematopoiesis (CH). Of note, according to the introduction they seem to refer to CH as the entity described by Jaisawal et al and Genovese et al, which reflect hematopoietic clonal growth without any bone marrow or blood abnormality. This CH framing is apparent for example in the manuscript title where BPDCN is not included, as well as several introduction and discussion points. These points may not be sufficiently supported by data. It is unclear what data is provided to support amending the “seed and soil” paradigm or to support tissue specific oncogenes. The paper does not address mechanisms of tissue tropism or local selective pressures, and it is unclear what is the relevance to CH and cardiovascular risk. I am also not sure I see how peripheral tissues shape the evolution of CH (other than the UV signature supporting a pDC cell of origin in the skin). While I understand that the ability to compare skin to BM is of interest (to address “distinct stages of disease evolution”), it is unclear how different it would be from the growing body of knowledge comparing CH clones to later arising common myeloid neoplasms, CH clones that remain after treatment for myeloid neoplasms, or CH co-occurring with myeloid neoplasms.

Furthermore, it is unclear to what degree the BM abnormalities in BPDCNs are representative of CH. BPDCNs are known to commonly have concurrent neoplastic BM disorders (MDS/CMML and AML), and a predilection for TET2 mutations (see references below). Many of the patients included here have significant cytopenia further supporting a concurrent bone marrow dysfunction. While the BM is noted to be uninvolved with BPDCN, it is unclear whether a close evaluation for dysplastic features was undertaken (especially for the more genetically aberrant/high VAF samples). The genetic information is also suggestive of a process that is further along than typical CH. The VAFs are high (often close to the entire sample), there are multiple gene mutations within the same clone and in 1 of 3 cases studied for CNVs also a large (chromosome arm) LOH event. Together with the cytopenia, these presentations may be more consistent with a concurrent MDS pattern than a functionally normal marrow as in CH. While cytopenia are often found in BPDCN and may be related to other causes such as splenomegaly, the conjunction of a genetically abnormal marrow may suggest bone marrow dysfunction at least in some cases. To my mind, this report is similar to, for example, a recent paper by Cohen Aubart et al (Blood, 2021) which looked at another rare clonal disorder (Erdheim-Chester disease) of myeloid origin, known to

have significant overlap with myeloid neoplasms, which was found to harbor large TET2-mutated BM clones. Overall, I am not sure the broad claims about CH are justifiable given that BPDCN is a rare disorder that may not represent the majority of CH. Even so, the observation that myeloid malignancies often arise against a backdrop of a parent CH clone is well accepted, and therefore it is unclear how this current work adds to our understanding of this process.

We thank the Reviewer for their comments and for supporting the study of rare cancers. We also believe that the data presented here provide new insights into cancer evolution that will be of broad interest to the scientific community. We have performed substantial revisions to the manuscript in response to the Reviewer's very helpful critiques, which we attempt to summarize and address below.

A first critique relates to use of the term clonal hematopoiesis (CH) in reference to our patient cohort. We agree with the Reviewer and acknowledge that our use of this term in the initial submission lacked clarity. As the Reviewer rightly points out, the patients studied here by definition do not have Clonal Hematopoiesis of Indeterminate Potential (CHIP, per Jaiswal et al., Genovese et al., and Xie et al.)¹⁻³ as they have been diagnosed with a hematologic malignancy. Moreover, some patients also have cytopenias and/or complex genetics in the bone marrow, which further argues against classification as typical CHIP. While examination by a hematopathologist did not reveal morphologic dysplasia to support an overt myelodysplastic syndrome (we now specifically comment on this in revised Supplementary Table 1a), we agree that the overall picture suggests bone marrow dysfunction beyond what is seen in CHIP. In response to these helpful critiques, we have clarified our language to convey the premalignant nature of bone marrow clones in our cohort, and now include new single cell data that specifically resolves normal, premalignant, and malignant pDC-states. We have also removed overly broad references to CHIP/CH throughout the text and eliminated the term "clonal hematopoiesis" from the manuscript title. In parallel, we have crystallized our message to focus on the mutational evolution of clonal (pre-malignant) bone marrow-derived cells in the periphery (skin), and to study the role of site-specific mutational processes (UV radiation) during tumor development.

A second critique raises the question of novelty relative to other studies of myeloid neoplasms associated with underlying clonal hematopoiesis (as in the nice study of Erdheim-Chester Disease (ECD) cited by the Reviewer⁴). We agree that there are some parallels with prior work, namely that malignant cells are clonally related to bone marrow precursors harboring mutations in leukemia-associated genes (e.g., *TET2*). However, to our knowledge our study is the first to explore the evolution of clonal (pre-malignant) bone marrow-derived cells in secondary/peripheral tissue sites, including upon exposure to site-specific mutational processes. Our revised study uses a wealth of new phylogenomics, single cell, clinical, and functional data to associate UV radiation, a skin-specific mutagen, with progression of pre-malignant

bone marrow-derived pDCs in the skin. These data suggest tissue and cell-specific functions for *TET2* inactivation in disease evolution, including (i) expansion of hematopoietic clones in the bone marrow, and (ii) resistance to UV-induced cell death in pDCs in the skin. Moreover, we believe our concept of “retrograde dissemination” of UV-mutated malignant cells from the skin is distinct from paradigms in AML and many other myeloid neoplasms, which involve clonal progression to malignancy entirely within the bone marrow. Taken together, our data support a novel model of premalignant evolution and disease progression that unfolds across distinct anatomic sites and implicates tissue-specific mutational mechanisms.

Please find additional comments detailed below regarding the strength of the data supporting some of the claims:

R1Q1. 1. The authors elected to perform the bulk of the sequencing with a targeted panel. The panels cited in the methods section seem to be quite small (19 genes for BM samples and 8 genes for skin samples, although this is in conflict with the main text indicating a 95 gene panel). Given the small number of cases and the rarity of the disease, I wonder whether WES of the entire cohort may be warranted. The WES is central to this paper’s major novel claim (UV signature) as well as to the understanding of to what degree is the BM aberrant genetically. Standard WES is likely sufficient for most patients in terms of sequencing depth considering the high VAFs in the BM.

For bone marrow samples, the targeted sequencing method was a 95-gene leukemia panel (Rapid Heme Panel), which includes all of the recurrently mutated genes in myeloid neoplasms and BPDCN⁵. For skin tumors, the targeted sequencing method was a 282-gene pan-cancer panel (Oncopanel), which covers recurrently mutated genes in leukemia, BPDCN, and solid tumors, and is suitable for use with formalin fixed paraffin embedded (FFPE) archival tissue^{6,7}. In the Methods section cited by the Reviewer, the numbers indicated were meant to show sample numbers profiled by each method rather than the gene numbers. Our presentation of this was unclear, however, and we have fixed this in the revised Methods. We have also clarified all of the pertinent text, figure legends, methods, and citations related to the targeted sequencing assays employed, and added a table summarizing the sequencing methods used for each sample (Supplementary Table 1b).

We also appreciate the Reviewer’s suggestion to expand sequencing data in order to validate our major claims about tumor evolution and UV damage. In response to this helpful critique, we now include significantly more phylogenomics data and UV analysis to complement the WES from the initial submission. We now present complete phylogenies and mutational signature analysis for 5 patients from our cohort, including 12 new samples profiled with whole genome sequencing (3 from skin tumors, 6 bone marrow, and 3 paired normal tissue; presented in Fig. 1d-e). In addition, we provide new mutational signature analysis of 25 WES samples from 3

separate BPDCN datasets (EDF 9b). These data confirm and extend the clonal hierarchies, disease trajectories, and UV mutational signatures reported in the initial submission, as described in further detail in response to R1Q5 below.

R1Q2. 2. The enrichment in PBDCN in TET2, ASXL1 and RNA splice factor mutations is well known. I am not sure the data here on this small cohort is sufficient to support claim that “common age-related mutations could predispose to dendritic cell transformation”. This needs a more careful treatment using sequential data, population studies or some form of control. Given how rare BPDCN is and how common CH mutations are, creating this linkage may be premature and may have untoward consequences.

Thank you for this helpful comment. We agree that our suggestion in the initial submission regarding age-related mutations and predisposition to dendritic cell transformation was too broad. While our data and prior publications do suggest a link between *TET2* and dendritic cell differentiation^{8–10}, it is also true as the Reviewer points out that most patients with *TET2*-mutated clones, in the setting of CHIP or other myeloid neoplasms, do not develop dendritic cell neoplasms. In the revised manuscript, we have removed generalizations to CHIP/CH and instead focus on the process of premalignant evolution and leukemic transformation across anatomic sites and in response to UV radiation in the skin.

R1Q3. 3. The single cell data involves only two individuals, which is a weakness. The main claim that is made with regards to these data is that mutated cells are observed throughout most of the cell populations in the bone marrow. This is to be expected considering that the VAFs in bulk DNA sequencing of these mutations would support a near homogenous involvement in the bone marrow as depicted in the Fish plots in Fig 3. In fact, indicated wildtype cells are likely largely false negatives. For example patient 9, should have 90% mutated cells based on bulk DNA VAF. Given this study design, the single-cell data add relatively limited new information. This may be further compounded by the low efficiency of genotyping (~7% summing information from all eight targeted loci in Fig 2c-d). The authors also make a claim about exclusion of mutated cells in mature T and B cells. This is often supported by only a few cells (<10 cells), without statistical evaluation. These data are interpreted as suggesting differentiation biases. Considering that lymphocytes may be long lived (especially as most of the cohort consists of elderly individuals), this may reflect different cellular life spans rather than a differentiation bias. Evaluating mutation frequencies in lymphoid progenitors may be more informative to differentiation biases.

We thank the Reviewer for their critique and agree that insights provided by the scRNA-seq data presented in the initial submission were limited by only representing two patients. This prevented us from validating key observations about the distribution of founder and progression mutations across hematopoietic lineages, including pDCs. Moreover, our initial study only included one sample with marrow

involvement by malignant cells (Patient 10 relapse), preventing us from generating a generalizable BPDCN gene expression signature. As a result, our initial submission could not provide a detailed comparison of normal, premalignant, and malignant pDC transcriptomes/cell states. To address these weaknesses, we performed scRNA-seq with genotyping on many additional patients and samples, nearly tripling the size of our single cell dataset as summarized here:

	Initial Submission	Revision
Number of healthy controls	3	6
Number of BPDCN samples without marrow involvement	2	5
Number of BPDCN samples with marrow involvement	1	6
Total number of samples	6	17
Total number of cells (single cell transcriptomes)	30,770	87,011
Total number of genotyped cells	12,963	27,994
Total number of genotyped pDCs/BPDCN cells	3,726	8,198
Total number of genotyped mutations/sites (UV-associated)	15 (4)	40 (13)

Using this greatly expanded single cell and genotyping data, we were able to validate key observations from the initial submission and generate new insights, as follows:

(1) We validate the presence of founder mutations across hematopoietic precursors and differentiating myeloid and erythroid populations in five patients, increased from two patients in the submission (Fig. 2d). We also demonstrate the presence of progression mutations (i.e., those specific to BPDCN skin tumors) uniquely within malignant BPDCN cells.

(2) We more confidently show that founder mutations are present in a higher proportion of mutated HSPC/erythroid/myeloid cells when compared to B/T/NK cells ($p < 0.05$ in 4/5 patients by Chi-square test, Fig. 2e). This conclusion is now based on a much higher number of total cells, which were relatively sparse in our initial data as the Reviewer rightly pointed out. We agree it would be very interesting to assess mutation frequencies in lymphoid progenitors, but we could not reliably quantify the proportion of mutations in these cells due to their small numbers.

(3) We use our single cell data with genotyping to validate inferred clonal hierarchies from bulk sequencing. In Patient 10, we identify truncal ASXL1 mutations and two subclones harboring different pairs of TET2 mutations (EDF

6g). In Patient 9, we identify an expanded, marrow-specific subclone defined by chr 7q loss of heterozygosity (LOH) based on co-occurrence of 7q “B” alleles with TET2 and CUX1 mutations (EDF 6h).

(4) We now resolve normal, premalignant, and malignant pDCs using combined transcriptional and mutation profiling. We define a novel BPDCN gene signature and identify characteristic features of premalignant pDCs, including upregulation of TCL1A, a signature BPDCN transcription factor, and downregulation of chemokine receptors (e.g., CXCR4) involved in bone marrow homing (Fig. 2f-h, EDF 8a-b).

(5) We genotype a total of 13 UV-associated progression mutations/sites (increased from 4 in the initial submission) and show their localization within malignant BPDCN cells but not other hematopoietic cell types (Fig. 3f, EDF 9e). This further validates that the clonally expanded UV mutations identified by WES/WGS are present in tumor cells rather than bystander skin cells.

We also appreciate the Reviewers appropriate critiques regarding distinctions between wild-type and mutant calls. We agree that, in the case of heterozygous mutations, only the wild-type allele may be detected and the mutated allele may be missed. When we superimpose wild-type variant calls onto individual cells, they may appear wild-type even though they are not. This is a difficult issue of calling variants from sparse single cell datasets, and we note that other groups have faced similar challenges^{11,12}. In response to the Reviewer’s appropriate concerns, we addressed this issue in three ways. First, we now label figures with “Variant calls” to clarify that we are indicating the detection of mutated/wild-type transcripts, and not wild-type or mutant cell classifications (e.g., Fig. 2d, Fig 4e). Second, we include several mutation sites without a wild-type allele in our XV-seq analyses, such as the *CDKN2A* deletion, *TET2.Q1537**, *SETX*, *SMARCC1* (loss of heterozygosity), and *ZRSR2* and *RAB9A* (X-chromosome in male patient). This allows unambiguous discrimination between mutant and wild-type cells on the basis of a single variant call. Third, we have specifically highlighted the issue raised by the Reviewer in the Methods, as follows: *“In the case of heterozygous mutations, cells in which a wild-type transcript is detected are not necessarily wild-type cells, as the mutated allele may have been missed.”* We believe these changes should more accurately convey our single cell genotyping data in line with the Reviewer’s suggestions.

R1Q4. 4. The frequency of observing wildtype alleles is somewhat concerning (light gray color choice is a bit tricky for visualization). Considering the fish plots in Fig 3, and bulk VAF data, one would expect the large majority of TET2 transcripts to be mutated in these two samples. For example, for Patient 9, the bulk DNA VAF suggest that 70% of cells are compound heterozygotes with an even higher number for patient 10. In part this could reflect partial sampling of the alleles, and in part can be due to the fact that transcripts labeled as

wildtype, actually contain a variant in a different locus on the same transcript. Analysis that takes into account clonal phasing may be more informative. In addition, direct examination of potential confounders such as difference of expression levels in different cell populations may be warranted. The ASXL1 variants may occur in a repetitive sequence which is prone for high rate of PCR artifacts. More broadly, the authors present XV-seq as an enhanced method. However, I am not sure this dataset makes the strongest case to that point. Genotyping aggregated across 3-5 targets/sample in Fig 2 only provided info for ~7% of cells. This seems to be a worse performance than the author's previous report. Given that the authors only assess mutation frequency in relation to broad cell types (HSPC, ery, myeloid, B and T cells as in panel e), I wonder whether an analysis as in Miles et al, 2020 with the tapestri platform + oligo-labeled antibodies may be a more appropriate method. Alternatively, a fairly straightforward sorting experiment, followed by targeted DNA sequencing may more readily address the question in Fig 2 without the technical confounders related to capture and target expression.

Thank you for these suggestions. We appreciate the Reviewer's critique that a closer examination and clearer explanation of the frequency of wild-type calls in our data is warranted. We agree with the various potential confounders outlined by the Reviewer and have taken the following steps to address them:

(1) We have improved the color scheme in the updated visualization of single cell genotyping data of founder mutations (Fig. 2d) and throughout the manuscript.

(2) We now refer to "variant calls" of specific mutations, and not mutant/wild-type transcripts or cells (please see also response to R1Q3 above). Furthermore, we now provide specific clarification about this point in our Methods, as follows: "*In the case of multiple mutations within the same gene (as is observed for TET2 in BPDCN), transcripts may show a wild-type result at one site while still harboring a mutation in cis at a different position in the same transcript/allele.*"

(3) We have conducted an examination of the relationship between gene expression and genotyping efficiency. As the Reviewer points out, transcript expression ultimately corresponds to the abundance of cDNA from which a mutation site can be captured. Indeed, we see this effect in our data (Response Fig. 1 below and EDF 6d). To control for this, we always evaluate the ratio of mutated cells over total genotyped (mutated + wild-type) cells when determining the proportion of mutated cells (e.g., Fig. 2e). Thus, while gene expression may impact the likelihood of obtaining a genotyping call for a given mutation, it should not confound our conclusions regarding the proportion of mutated cells in any cell population.

Response Figure 1 | Dot plots show the proportion of cells in which a *TET2* transcript was captured at the indicated mutation site (x) vs. the mean *TET2* expression in that population (y). Cell types with <10 cells in the indicated sample are omitted. In most cases, there is a positive correlation between genotyping efficiency and expression, as expected. These analyses are available on https://github.com/petervangalen/Single-cell_BPDCN_5.4_Genotyping_efficiency.R.

(4) We clarify quality thresholds for variant calls and validate findings for the artifact-prone *ASXL1* mutation cited by the Reviewer. For our genotyping analysis, we used stringent quality thresholds that result in high-confidence mutation calls, including a requirement for transcripts to be supported by ≥ 3 reads with ≥ 3 -fold more wild-type than mutant calls or vice versa. Regarding the specific *ASXL1* mutation highlighted by the reviewer, this is indeed a challenging site due to homopolymeric sequence and known PCR artifacts. To address this concern, we replicated findings with different *ASXL1* enrichment primers in scRNA-seq libraries from Patient 10 Dx (from whom we analyzed 10,106 high-quality single-cell transcriptomes). In the first experiment, we detected mutated *ASXL1* transcripts in nine cells. In the second experiment, we detected mutated transcripts in eight cells. Seven of the cells overlapped between the two attempts, indicating concordance. Moreover, wild-type calls between the two experiments were perfectly concordant, indicating the absence of significant PCR artifacts and reproducibility of the single-cell genotyping pipeline. The agreement between these experiments, together with

the orthogonal targeted DNA sequencing that identified the same mutation, attests to the reliability of our mutation calls. We have added a description of this validation experiment to the Methods section and made the analysis available on Github:

(https://github.com/petervangalen/Single-cell_BPDCN/blob/main/13_Misc/13.3_ASXL1_concordance.R).

Moreover, we appreciate the Reviewer's comments regarding comparison of XV-seq to prior approaches, including our own¹³. We do believe that XV-seq represents conceptual and practical improvements that are of broader use to the scientific community, notably evaluating the regular single-cell RNA-seq data to select informative highly expressed variants. To clarify similarities and differences, including genotyping efficiencies, we have included new analyses in EDF 6a-e (please see response to R1C3 above) and incorporated the following detailed description into the revised Methods:

“Compared to prior methods by us and others^{13,14}, we incorporated a number of computational and experimental steps for increased sensitivity and specificity: (1) We considered all mutations detected by whole exome sequencing, including synonymous mutations and mutations affecting untranslated regions (UTR). These mutations do not result in changes in the protein sequence, but can be used to infer clonal relationships. (2) We quantified detection of these mutations in the regular scRNA-seq data prior to enrichment. For example, of the 186 mutations detected across samples for Patient 10, only 16 (8.6%) were detected in at least one transcript (Supplementary Table 2d). We found that detection in the regular scRNA-seq data is a good predictor of enrichment efficiency (Extended Data Fig. 6c). (3) We specifically considered loci of which only a single allele is present in the genomes of healthy and/or malignant cells. For these mutations, detection of the wild-type allele is as informative as the presence of the mutant allele (i.e. if the wild-type is detected, the mutant must be absent; for heterozygous mutations the mutant could remain undetected). In our dataset, this included a mutation in the X-chromosomal gene RAB9A (in a male patient), a focal deletion of CDKN2A/B which occurred in cells already carrying loss of chromosome 9, and 3' UTR mutations in SETX and SMARCC1 which also occurred in cells with loss of the other allele on chromosome 9 and chromosome 3. (4) Finally, we incorporated technical optimizations such as inclusion of dual indices, as outlined below.”

Finally, regarding the suggestion of a cell sorting experiment, we have obtained and analyzed data of sorted progenitor and malignant BPDCN cells of one patient from Batta et al¹⁵. Our reanalysis confirmed that founder-type mutations (e.g., with TET2 mutations) were present in all populations, including sorted HSPCs, while progression and UV-type mutations were present only in BPDCN cells (EDF 9b, shown below for convenience). Moreover, we have markedly expanded the size of

the single cell profiling and genotyping dataset for our cohort (as summarized in R1Q3 above). We believe this provides key validation about the distribution of mutations across hematopoietic lineages, akin to that which would be provided by the orthogonal methods suggested by the Reviewer (e.g., Tapestri).

EDF 9 | b, Heatmaps show mutational signature analysis of BPDCN patient samples analyzed by whole exome sequencing (only samples from Batta et al. are shown here). Blue heat indicates the predicted relative contribution of the mutational signature. Relative contribution of signature 7 (UV damage signature) is indicated. Samples from Batta et al. were generated from indicated bone marrow populations at diagnosis (n=3) and relapse (n=1), and a skin tumor sample, all from the same patient. The UV damage signature is detected in all samples containing malignant cells (relative contribution ≥ 0.38).

R1Q5. 5. The UV signature is perhaps the most novel aspect of this study, as it links the cell of origin to a skin pDC. Of note, mutation rates are still lower than those seen in skin cancer, which may reflect shorter time in the skin pre-transformation. Perhaps expanding this experiment with WES beyond three individuals may help strengthen this claim. Furthermore, Given the paucity of available data, the authors may consider re-analyzing published data for UV signature (Menezes et al, Leukemia, 2014 [including exomes], Alayed et al, Am J Hematol, 2013 [including BM BPDCN], Stenzinger et al, Oncotarget, 2014, Beird et al, Blood Cancer Journal, 2019). I also note that signature decomposition may benefit from a confidence estimate. For example, the signature decomposition of a handful of mutations in the BM samples may not be very telling. This may also explain the strong MMR/MSI signature in the BM samples that is probably an artefact here given the low mutation number.

We agree that the UV findings are key to our study and have added seven lines of evidence that link UV radiation to BPDCN evolution:

(1) We have added a mutational signature/UV analysis from three additional publicly available WES datasets as suggested by the Reviewer. This confirms the presence of UV mutational signatures in BPDCN and is presented in **EDF 9b**.

(2) We performed WGS on 12 samples (no WGS were included in the initial submission) and analyzed somatic variant signatures, which revealed the same UV mutational pattern. WGS was particularly helpful to increase the total number of assessed mutations, compared to WES, and to contribute power to the signature analysis of BM samples at diagnosis, as pointed out by the reviewer. Mutational signatures from these data are shown in new **Fig. 3b** (shown below for convenience).

Figure 3 | b, Heatmaps show mutational signature analysis of WGS and WES data for bone marrow (top), skin tumors at diagnosis (middle), and relapse samples (bottom). Blue heat indicates the relative contribution of each signature. The total number of single nucleotide variants (SNVs) is indicated on the right.

(3) We show additional visualizations of UV-specific dinucleotide mutations (CC>TT) by anatomic site (Fig. 3d, shown below), which highlights that these types of mutations represent the majority of dinucleotide mutations and are specific to skin samples.

(4) We now include evidence for transcription-coupled repair of UV-associated variants (TC>TT and CC>CT), which is a characteristic feature associated with DNA repair of UV damage on transcribed strands (Fig. 3e, shown below). This further validates UV radiation as the mechanism behind the observed mutational signatures.

(5) As requested by the Reviewer, we have added statistical analyses to the signature scores in Fig. 3 and EDF 9 as follows:

“We found a striking enrichment for ultraviolet (UV) light-induced DNA damage signatures (Signature 7) in 5/6 BPDCN skin tumors at diagnosis (relative contribution 0.49-0.65) and in 3/4 samples collected at relapse (0.52-0.57; Fig. 3b). In contrast, UV signatures were not found in six matching bone marrow samples (relative contribution 0-0.14, $P < 0.005$, Wilcoxon rank sum test; Fig. 3b, Extended Data Fig. 9a).”

“Consistent with this, UV-associated mutations comprised the majority of mutations specific to BPDCN skin tumors (43.1-55.6% TC>TT, 13.3-37.7% CC>CT, $n = 5$ from Patients 1, 7, 9, and 10) and relapse samples (32.0-56.7% TC>TT, 11.9-19.4% CC>CT, $n = 3$ from Patients 1 and 12) in our cohort (Fig. 3c,

Extended Data Fig. 9c-d). In contrast, only 5.3-15.4% of founder mutations in six matching bone marrow samples occurred in the TC>TT context ($P<0.001$, Wilcoxon rank sum test). Moreover, UV-specific CC>TT mutations were exclusively detected in BPDCN skin tumors and relapse samples ($n=82$, 71.1% of all dinucleotide mutations from Patient 1), but not in matched bone marrow (Fig. 3d). Finally, UV-associated TC>TT and CC>TT mutations were enriched on the non-template strand of coding genes (60.7% and 57.9%, $P<0.005$, Binomial test; Fig. 3e)."

(6) We visualized the presence of UV-type progression mutations in all 66,600 single cells from patient samples plotted by the random forest pDC prediction score (x-axis) and our BPDCN tumor gene expression signature score (y-axis). This analysis demonstrates near complete enrichment of UV-type dinucleotide mutations in malignant BPDCN cells (Fig. 3f, shown below).

Figure 3 | d, Barplot shows the number of dinucleotide variants detected in samples from Patient 1, grouped by UV-specific CC>TT and all other dinucleotide variants. Variants are only shown in the sample in which they were first detected. **e**, Barplot shows the number of UV-associated TC>TT and CC>CT variants in samples from Patient 1 separated according to their presence on the template (transcribed) versus non-template (non-transcribed) strand of annotated genes. **f**, Scatterplot shows single cells from all patient samples ($n=66,600$) according to their random forest pDC prediction score (x-axis) and BPDCN signature score (y-axis). Colors indicate combined UV-associated progression mutations as detected by XV-seq ($n=5$ TC>TT mutations, left; $n=6$ CC>CT mutations, right).

(7) As mentioned above, we have performed a new analysis of initial BPDCN skin lesion locations and found a striking enrichment at sun-exposed sites (Fig. 3g, shown below for convenience). This contrasts with AML that presents on any area of skin and is not enriched for sun-exposed sites. These clinical data connect UV exposure to patient clinical presentation and our new mechanistic data showing that *TET2* mutated pDCs having selective advantage in the setting of UV radiation.

Figure 3 | g, Schematic shows the location of index BPDCN skin lesions (left), progression BPDCN skin lesions (middle), and acute myeloid leukemia (AML) skin lesions (right). Grey shading indicates areas of chronic or intermittent UV exposure. Representative clinical photos are shown.

R1Q6. 6. The authors show in Fig 4 that the single cell data identified 19 malignant cells in a pre-treatment BM sample. They argue for this as a potential novel diagnostic tool and an "important step towards single-cell multi-omics for diagnostic use in early detection, circulating tumor cell identification, or measurable residual disease (MRD) evaluation." I find this to be a fairly strong claim based on 19 cells within one individual. Any classifier is prone to over-fitting without proof of generalizability in independent samples. Even within this sample, the 19 cells are compared with only 39 normal pDCs. It is also unclear what significance this would have in clinical management.

We were encouraged by the high agreement between transcriptional BPDCN signature scores and XV-seq genotyping data in identifying rare malignant BPDCN cells in the original submission (progression mutations were detected in 17/19 putative malignant cells, but not in 39 normal pDCs or in 10,055 other cells). However, we agree with the Reviewer that claiming this as a tool to aid clinical management is premature. We have removed any suggestion that the rare cells harboring BPDCN signatures and skin tumor-associated mutations are of potential clinical utility (early detection, or MRD quantitation). Instead, we make note of those rare cells (and additional cells identified during the revision) as evidence of circulating disease (Fig. 2h and EDF 8c-e). This helps to frame the central question about disease pathogenesis and order of mutation acquisition.

Referee #2 (Remarks to the Author):

In this manuscript from Griffin and colleagues, the authors seek to use a study of blastic plasmacytoid dendritic cell neoplasm (BPDCN) as a model for understanding evolution of a cancer from a pre-malignant state. The authors initially show that uninvolved bone marrow in patients with BPDCN has frequent clonal hematopoietic mutations with high variant allele frequencies. This finding is supported by other case reports in the literature. The authors then perform single cell RNA sequencing and show normal hematopoiesis and use a modified protocol for somatic mutation assessment, as this group and others have reported, to show that cells across the hematopoietic hierarchy harbor the clonal driver mutations. Next, the authors conduct exome and targeted sequencing of bone marrow and matched skin tumors. This reveals a higher burden of mutations in the skin tumors compared to the bone marrow samples and also reveals a signature associated with UV-induced mutagenesis in the BPDCN samples. Finally, the authors use the XV-seq method to try to track the origins of a bone marrow relapse in a patient with BPDCN and suggest that the origin may be from a transformed pDC-like cell.

This is an interesting manuscript that explores an important issue in the field of cancer biology, which is to define the evolution of a malignancy. Unfortunately, many of the conclusions made in this work are not entirely supported by the data and the authors fail to consider alternative possibilities underlying some of the observations. I will provide a few examples here:

We would like to thank the Reviewer for their comment that our manuscript is interesting and explores an important issue in cancer biology. Prompted by the Reviewer's critical feedback, we have generated extensive new data, analyses, and functional experiments, and now provide a more comprehensive and clear presentation of results. We believe that these additions have greatly improved the quality of our manuscript and address the Reviewer's concerns, as outlined below.

R2Q1. - On pages 4-5, the authors use exome sequencing on paired samples and state, "malignant BPDCN skin tumors harbored a much higher overall burden of mutations (range 74-229), only a minority of which were found in matched bone marrows ... These findings confirm a direct clonal relationship between CH in the bone marrow and malignant BPDCN cells in the skin." This conclusion seems inappropriate given the evidence that is presented. The authors are comparing bone marrow with a variety of clones present to a skin BPDCN tumor that has a clonal origin. Exome and targeted sequencing both have detection limitations. How can the authors be sure that the observations are not simply attributable to the bottleneck present in the skin tumor emerging from a clone that has more readily detectable mutations?

We thank the Reviewer for this critique. We agree that it is important to consider that malignant transformation may occur in the bone marrow but is first observed in the

skin due to a bottleneck event. We have added a new Fig. 3a (shown below for convenience) with accompanying text to clearly present two models of BPDCN development: (1) transformation of premalignant pDCs occurs in the marrow followed by spread to the skin, or (2) premalignant pDCs from the bone marrow home to the skin, undergo malignant transformation, and then disseminate “back” to the bone marrow.

Figure 3 | a, Illustration presenting two alternative models of clonal progression to malignancy in BPDCN. In Model 1, malignant cells (red) transform from clonal precursors (blue) in the bone marrow, followed by spread to the skin. In Model 2, malignant cells (red) transform from clonal bone marrow-derived cells (blue) in the skin, followed by retrograde dissemination back to bone marrow.

Indeed, attempting to resolve this question has been a major focus of our efforts for the revised manuscript. A key observation in favor of Model 2 is the presence of a strong UV mutational signature in many (but not all) BPDCN cases. In such cases, UV-associated mutations represent the majority of tumor-specific variants. The revised manuscript provides extensive validation of this finding using (i) whole genome sequencing of additional patients from our cohort, and (ii) analysis of datasets from 3 additional BPDCN studies. These data are presented in the revised Fig. 3 and EDF 9, and together indicate that the majority of mutations in many BPDCN tumors are acquired in the skin.

However, as the Reviewer indicates, UV signatures alone do not provide conclusive evidence that transformation occurs in the skin. Perhaps a bottlenecking event causes a malignant cell (that transformed in the marrow but was occult to pathology evaluation) to acquire UV mutations prior to clonal expansion in the skin and systemic dissemination. In the revised manuscript, we present several lines of evidence that argue against this interpretation as a common mechanism:

(1) Malignant cells and associated progression mutations (e.g., RAS mutations) are undetectable in the bone marrow of many BPDCN patients, including with sensitive technologies such as flow cytometry, targeted sequencing, and single cell genotyping

that were employed in this study. In cases where rare (occult) malignant BPDCN cells are detected, our analysis indicates that they already harbor UV mutations, indicating prior transit through the skin (EDF 8d-e, EDF 9e). Thus, we are unable to identify a population of malignant cells lacking UV mutations in the bone marrow of most patients.

(2) New analysis of clinical dermatology data shows that BPDCN index skin tumors typically involve sun exposed sites while lesions from AML patients do not (Fig. 3g). These findings disfavor Model 1 (i.e., transformation in the bone marrow followed by incidental acquisition of UV damage once present in the skin), as index skin tumors would not be biased towards sun-exposed sites.

(3) In some patients, it is evident that UV damage begins prior to genomic alterations associated with malignant progression. We track the order of mutation acquisition in an exemplary patient (Patient 10) with multiple biopsies from different anatomic sites and clinical time points (visualized in Fig. 4a-d shown below and EDF 10b). Phylogenomic analysis of this case showed a prominent UV signature and two distinct malignant subclones that were separable based on unique but convergent structural alterations. Both malignant subclones showed loss of the *CDKN2A* locus (chr 9p) but each harbored unique breakpoints, indicating distinct progression events (Fig. 4c). In addition, both malignant subclones showed loss of chr 3p loci (including *SETD2*) but each involved a different allele, also indicating distinct progression events (Fig. 4d). Analysis of shared mutations between the two malignant subclones showed that more than half (53.3%) were UV-associated mutations in the TC>TT context. This indicates that UV damage began *prior* to the loss of canonical tumor suppressor genes (e.g., *CDKN2A*, *SETD2*) often associated with transformation^{16,17}. Furthermore, in a second patient (Patient 14), we find evidence for a pathogenic UV-specific mutation in *ETV6*, a known leukemia driver gene.¹⁸ This also suggests that UV damage begins prior to the malignant transformation.

Together, we think these data provide support for Model 2 as a common mechanism in BPDCN. In this model, clonal bone-marrow derived pDCs or committed progenitors localize to the skin and subsequently undergo malignant transformation (Fig. 4j). Our results indicate that UV damage can precede malignant transformation, and nominate a clonal precursor in the skin as the likely origin of BPDCN in many cases. However, we also acknowledge that there is heterogeneity in BPDCN pathogenesis and that alternative models will likely apply in some cases.

Figure 4 | a, Subway plot shows the clonal dynamics and clinical features over the disease course of Patient 10. Whole-exome sequencing samples (n=5) are indicated by black dots. Connecting lines are colored for their percentage of UV-associated TC>TT mutations from green (0%) to orange (50%). Width of connecting lines indicate the total number of detected variants. Bottom plot shows bone marrow blast count (black line) from pathology assessment, and donor chimerism (gray line) following allogeneic stem cell transplant. **b**, Barplots show VAFs of somatic mutations detected in bone marrow and two skin tumor samples collected at diagnosis from Patient 10, grouped by the sample in which they were first detected. UV-associated TC>TT mutations are indicated in orange, other mutations in gray. Asterisks indicate mutations that are affected by copy-number alterations or are located on chromosome X. **c**, Genome plot shows normalized WES read coverage along a region on chromosome 9 for two skin tumor samples from Patient 10. Separate homozygous deletions affecting the *CDKN2A* tumor suppressor gene are indicated (red bars). **d**, Genome plot shows phased allele frequencies of heterozygous SNPs along a region on chromosome 3 for two skin tumor samples from Patient 10. Colored bars (blue, A allele lost; red, B allele lost) indicate that different alleles of a region harboring the *SETD2* gene were lost in each sample.

R2Q2. - The above issue is also present to the analysis of UV induced mutations. This is interesting, but is this really causal in "shaping the evolution" as the authors suggest? The clonal cells could simply arrive in the skin and be subject to UV induced mutations that can be detected at higher VAFs, due to the clonal nature of BPDCNs.

The Reviewer is correct that the acquisition of UV-induced mutations per se does not prove that these mutations are drivers of malignancy. In response to the Reviewer's question, we pursued the following analyses:

First, we assessed our sequencing dataset for evidence of UV damage as a causal mechanism for isolated mutations in genes associated with leukemia. As has previously been demonstrated for skin cancer, this is challenging unless mutations involve UV-specific CC>TT dinucleotide substitutions¹⁹. Among twelve such

dinucleotide mutations identified in our cohort, we identified one mutation in a known leukemia driver gene (ETV6 R369W, Patient 14; presented in Fig. 4e). Thus, for most cases in our cohort we could not definitively identify UV-induced alterations in oncogenes or tumor suppressors. This is perhaps not surprising given the challenge of this even in the setting of canonical UV-associated skin cancers¹⁹.

We also considered another hypothesis whereby UV exposure might select for premalignant pDCs. To address this possibility, Fig. 4f-i of the revision (shown below for convenience) includes new functional data on UV sensitivity and *TET2* loss-of-function mutations in BPDCN, which are present as founder events in the bone marrow of most patients. While control pDCs are highly sensitive to UV radiation, we find that *Tet2* KO pDCs are resistant to UV-induced cell death. This indicates a novel tumor suppressor role for *TET2* during DNA damage responses and may explain how clonal *TET2*-mutated pDCs in the skin are able to survive UV radiation prior to malignant transformation. We believe these data indicate a potential mechanism by which UV radiation influences BPDCN pathogenesis.

Figure 4 | f, Illustration shows *ex vivo* culture system of primary mouse dendritic cells. Transduction of an estrogen-responsive form of HOXB8 into bone marrow cells generates progenitor cells that can be stably propagated *in vitro*. Estrogen withdrawal triggers differentiation over a 6-8 day period into mature pDCs and cDCs. For UV experiments, cells are exposed to a single dose of UV (0, 100, 500 $\mu\text{J}/\text{cm}^2$) at Day 6 and further differentiated until Day 8. **g**, Flow cytometry for B220 (x-axis) and CD11b (y-axis) show the proportion of cDC (CD11b⁺, B220⁻) and pDC (CD11b⁻, B220⁺) populations in control and *Tet2* KO HOXB8 cultures on day 8 after estrogen withdrawal. Gating is on viable CD11c⁺ cells, as

per EDF 10d. **h**, Barplot shows total number of viable cells in control and *Tet2* KO HOXB8 cells on day 8 after estrogen withdrawal. UV exposure at the indicated doses was performed on day 6. Data include two control and two *Tet2* gRNAs performed in triplicate, and are representative of two independent experiments. **i**, Barplot shows proportion of viable cells classified as pDC or cDC by flow cytometry in control or *Tet2* KO conditions (n=2 gRNAs each performed in triplicate) at the indicated UV dose. Data are normalized to the 0 UV condition and are representative of two independent experiments.

Lastly, we highlight a new dermatologic analysis of index skin lesions in BPDCN patients. We find a striking enrichment for BPDCN index lesions at sun exposed sites (presented in Fig. 3g and below for convenience). This contrasts with skin lesions in AML (“leukemia cutis”) that presents in a more disseminated fashion on any area of skin. This finding indicates that UV exposure is associated with the anatomic distribution of BPDCN skin lesions.

Figure 3 | g, Schematic shows the location of index BPDCN skin lesions (left), progression BPDCN skin lesions (middle), and acute myeloid leukemia (AML) skin lesions (right). Grey shading indicates areas of chronic or intermittent UV exposure. Representative clinical photos are shown.

R2Q3. - On page 7, the authors use XV-seq and show that tumor specific mutations were abundant in relapse pDC cells, but absent in hematopoietic compartments in the background. They then conclude that the data confirms "the retrograde pathway of tumor progression." I am not sure that this statement can be so conclusively made using the somatic mutations that were profiled.

Thank you for highlighting the need for additional evidence to support our proposed model of “retrograde dissemination” of UV-mutated malignant cells. We believe that the main evidence in support of this model derives from our genome sequencing analysis of bone marrow and skin tumor samples at diagnosis and relapse, while single-cell RNA sequencing and XV-seq provide additional evidence and link individual mutations to transcriptional cell states. In the revised manuscript, we now present whole genome sequencing data of two patients (1 and 12) that include bone marrow samples at relapse, in addition to the sample included in the initial submission (Patient 10). The new dataset confirms that mutations that are first detected in skin tumors at diagnosis (presented in Fig. 1c-d and below for convenience) are also detected in relapse bone marrow samples. This includes UV-associated and UV-specific mutations presumed to be acquired in the skin (Fig. 3b-d). Furthermore, in our extended single-cell data, UV-associated progression

mutations are highly enriched in malignant BPDCN cells but not in other blood cell lineages (Fig. 3f and EDF 9e). The presence of UV mutations in malignant cells in the bone marrow at the time of relapse suggest their prior transit through the skin (i.e., “retrograde dissemination”). Considering results presented in response to R2Q1 and R2Q2, we conclude that cells collected from relapse bone marrow samples must have transited through the skin via a retrograde pathway of disease progression.

Figure 1 | c, Tumor phylogenies reconstructed from whole exome sequencing of samples for Patients 7, 9, and 10 (only Patient 10 is shown here). The number of detected somatic single-nucleotide variants (SNVs, red), insertions/deletions (green), and copy-number alterations (blue) are indicated. Dashed line indicates that SNVs could not be detected because of high donor DNA fraction following stem cell transplant (SCT). All somatic alterations above quality thresholds are represented (see Methods). **d**, Tumor phylogenies reconstructed from whole genome sequencing of samples of Patients 1 and 12. Somatic SNVs (red) and copy-number alterations (blue) were defined in the latest relapse sample, and assessed in prior skin tumor and bone marrow diagnosis samples.

The authors have conducted an in depth and important single cell analysis of BPDCN here. This is certainly worth reporting to those who study this rare cancer in the hematology community. My concern is that the authors attempt to draw broad and sweeping conclusions about clonal evolution in cancer and how tissue microenvironments may be involved that are unsupported by the evidence shown. In addition, while interesting approaches like XV-seq are reported, these are similar to methods and studies that have already been described by this group and others (references 16, 17, and 44).

We agree with the Reviewer’s assessment, and therefore have changed the tone of the text to avoid overstating the generalizability of these data to clonal evolution except as where supported by our findings. We maintain that this work is relevant and novel beyond BPDCN in the ability to analyze premalignant bone marrow plus malignant skin tumors at the same time point and use analytical methods to track

and order mutation acquisition. We have added whole genome sequencing, nearly tripled the number of single cell sequencing datasets, added clinical assessment of skin lesion geography, and performed new functional experiments linking *TET2* mutations to pDC clonal advantage in the setting of UV exposure.

Regarding XV-seq and its relationship to prior Genotyping of Transcriptomes efforts by us and others, we have added new data and analyses to help the reader place this technique in the context of available tools. While we agree with the Reviewer that mutation assessment in single-cell RNA libraries can and should continue to be improved, we now provide more concrete comparisons of this technique to prior efforts. We have also provided improved description and a GitHub-accessible computational pipeline for readers to apply to their own datasets (Methods and Data and software availability).

Our mean genotyping efficiency by XV-seq was 10.2% of cells (range 0.1-99.0%, see manuscript Supplementary Table 4 for more details). We integrated the detection of informative loci in scRNA-seq libraries to achieve higher genotyping rates, which shows that at least in some cases this approach is highly beneficial (e.g. *RAB9A*, *CDKN2A*, and *RPS24* in 37%, 20%, and 99% of cells, respectively) (EDF 6a). We compared genotyping efficiency by XV-seq to without enrichment, which was a median of 11.1-fold improved (EDF 6c). We have also added a comparison of variant frequency detected by XV-seq and PCR-based targeted DNA sequencing (EDF 6e). Finally, we posit that not every cell needs to be genotyped to determine the order of mutation acquisition. The single-cell genotyping results were in agreement with tumor phylogenies inferred from orthogonal assays and post-transplant donor/host calls (EDF 6f-h). We thank the Reviewer for directing us to describe our XV-seq results with more clarity and transparency.

Regarding general interest, we suggest BPDCN offers a unique situation to study tumorigenesis across tissue contexts and timepoints. This is more challenging to assess in other cancers because of mixed stages of malignancy in the same biopsy and/or uncertain lead times between pre-cancer and full malignancy. As a concrete example of potential general relevance, we point to two recent reports where a clonal UV signature was unexpectedly detected in both non-malignant and malignant hematopoietic cells. Machado et al. showed that UV signatures are detected in normal memory T cells from peripheral blood, but not in other lymphocyte subsets²⁰. Brady et al. showed that UV signatures define a subset of iAMP21 and hyperdiploid pediatric B-cell acute lymphoblastic leukemias²¹. Both papers found these data to be surprising and unexplained by normally understood models of hematopoietic cell migration or tumorigenesis. We believe that our study including our new functional data suggesting that UV and *TET2* mutations interact to select clonal outgrowth, put in the context of these studies that lacked functional experiments, argues that exposure to the skin/UV environment may influence evolutionary trajectories. Thus, we think this report has interest outside of hematologists who study BPDCN.

Referee #3 (Remarks to the Author):

Griffin et al, investigate the relationship between clonal hematopoiesis (CH) and blastic plasmacytoid dendritic cell neoplasm (BPDCN), a rare form of acute leukemia that often presents with malignant cells isolated to the skin. Using samples from a very unique cohort of 12 BPDCN patients the investigators study patterns of tumor phylogenies that underpin progression of CH to BPDCN, and how these are further represented in patients with subsequent bone marrow involvement. Additionally, using supervised RNA-seq classification frameworks the authors study cell type representation across stages of the disease (pre leukemic, transformation, disease progression). Overall this is an elegant and thought provoking study, that provides novel insights of the molecular underpinnings of BPDCN. The manuscript is a bit challenging to read at times, and given the density of the data and methods could benefit from streamlining the narrative.

The observations made are very interesting, the analysis approach is novel and the data analysis and interpretation is sound. The manuscript reveals intriguing evolutionary trajectories underlying BPDCN pathogenesis, informed by elegant analysis of scRNA, and molecular profiling of spatially and temporally separated specimens. The incorporation of mutation signature analysis and clonal reconstruction offers robust evidence for the timing and directionality of clonal dissemination during BPDCN transformation, and relapse.

Thank you for this positive assessment of our work and for finding the study elegant and thought provoking. We agree that the data density is high. We have worked to streamline the manuscript and feel that the readability and clarity is improved.

R3Q1. 1.The use case in this manuscript, which is CH to BPDCN reflects a very unique and rare in its clinical presentation disease entity. In contrast the title and abstract of the study is rather broad and suggestive that this observation (peripheral tissue selection of pre-malignant cells) be generalized. It may be more appropriate to align the title, abstract and discussion of the manuscript to the focus of the study.

We agree with the Reviewer that we should not attempt to make conclusions about all clonal hematopoiesis (CH) or leukemia evolution using these data. We have realigned the text as the Reviewer suggests, to be clearer on that point. Specifically, we focused the text more on BPDCN itself and on how UV exposure shapes malignant evolution. We have tried to communicate that the unique clinical and pathological characteristics of BPDCN allow us to demonstrate that cancer evolution may involve more than one tissue environment. In the discussion we also refer to two recent papers which reported that UV-induced DNA mutational signatures were unexpectedly found in clonally-expanded hematopoietic cells in other contexts (Machado et al. showing UV mutational signatures in normal memory T cells from peripheral blood²⁰; and Brady et al. showing clonal UV mutational signatures in a defined subset of pediatric B-cell acute lymphoblastic leukemias²¹).

We believe that our data, including our new functional studies reported in this revised manuscript suggesting that *TET2* mutations confer a cell type-specific advantage in the setting of UV radiation (Fig. 4f-i), argues that other clonal hematopoietic cells may also be exposed to site-specific selective pressures which could influence their evolution. Thus, we think there will be general interest in this report beyond the BPDCN scientific community.

Figure 4 | f, Illustration shows *ex vivo* culture system of primary mouse dendritic cells. Transduction of an estrogen-responsive form of HOXB8 into bone marrow cells generates progenitor cells that can be stably propagated *in vitro*. Estrogen withdrawal triggers differentiation over a 6-8 day period into mature pDCs and cDCs. For UV experiments, cells are exposed to a single dose of UV (0, 100, 500 $\mu\text{J}/\text{cm}^2$) at Day 6 and further differentiated until Day 8. **g**, Flow cytometry for B220 (x-axis) and CD11b (y-axis) show the proportion of cDC (CD11b⁺, B220⁻) and pDC (CD11b⁻, B220⁺) populations in control and *Tet2* KO HOXB8 cultures on day 8 after estrogen withdrawal. Gating is on viable CD11c⁺ expressing cells, as per EDF 10d. **h**, Barplot shows total number of viable cells in control and *Tet2* KO HOXB8 cells on day 8 after estrogen withdrawal. UV exposure at the indicated doses was performed on day 6. Data include two control and two *Tet2* gRNAs performed in triplicate, and are representative of two independent experiments. **i**, Barplot shows proportion of viable cells classified as pDC or cDC by flow cytometry in control or *Tet2* KO conditions (n=2 gRNAs each performed in triplicate) at the indicated UV dose. Data are normalized to the 0 UV condition and are representative of two independent experiments.

R3Q2. 2. The authors use the term pervasive CH following the observation of high VAF clones in patients with BDPCN. Not clear what the term pervasive eludes to as a function of CH, particularly in the context of BDPCN.

The Reviewer is correct that this word was neither clear nor necessary. We have rewritten the text and simply refer to the variant allele frequency (VAF).

R3Q3. 3. The observation of bi-allelic hits in *TET2* is rather interesting and potentially novel. There seems to be enrichment of bi-allelic hits, which are mediated by two mutations or a mutation and an allelic loss. Comparison with publicly available CH or AML datasets could verify this and provide potential insights on the implications of bi-allelic inactivation of *TET2* in CH progression.

We thank the reviewer for this excellent suggestion. We present new data showing that BPDCN has much higher frequency of two (potentially bi-allelic) mutations in *TET2* compared with CMML or AML (presented in Fig. 1g and below for convenience). Moreover, the functional data we now present suggests that bi-allelic or complete *TET2* inactivation specifically renders pDCs relatively resistant to UV-induced cell death, compared with other dendritic or myeloid cells (Fig. 4f-i, see above response to R3Q1). This suggests a functional link between *TET2* complete loss of function and pDC survival in the skin, connecting this specific genotype to BPDCN and providing a potential explanation for clinical observations.

Figure 1g. Barplots comparing the frequency of SNVs and insertion/deletions in founder and progression genes between two BPDCN cohorts, acute myeloid leukemia (AML), and chronic myelomonocytic leukemia (CMML).

R3Q4. 4. The data in Supplementary Table 1b are rather intriguing. Comparison of molecular findings in the bone marrow at diagnosis, relative to the skin and bone marrow at follow up reveals complex branching phylogenies with clones that are shared in the bone marrow, skin and follow up bone marrow samples, emerging subclones in the skin as well as

evidence of clones that are confined in the bone marrow but not involved in transformation. This intriguing clonal structure is not formally presented in the main text. The manuscript could benefit by a more detailed and visual representation of these results.

We share the reviewer's enthusiasm for the analysis of complex tumor phylogenies in BPDCN patients at different anatomic sites and across longitudinal timepoints, and now present these results more prominently in Fig. 1 of the manuscript. Moreover, the revised manuscript includes 13 new whole genome sequences from three patients of our cohort (including 4 from skin tumors, 6 bone marrow, and 3 paired normal tissues). These additions confirm our model of clonal evolution from normal to CH to BPDCN, within the same patient's BPDCN at distinct tissue sites and over time during disease progression. We have also emphasized the branches of the phylogenies in the marrow that do not lead to the transformation, as suggested by the Reviewer. The new representations are more visually intuitive and offer an improved picture of each patient's disease.

R3Q5. 5. With regards to cell type annotation from RNA-seq, the authors first perform a manual annotation of the healthy donors cells using a select list of gene markers. Then the authors train a RF classifier which takes as input the expression of the cell-type specific genes and outputs a probability of cell type assignment. This model is subsequently applied on the BM negative cells to assign each cell to the type where the probability of assignment is higher. The model is trained and applied in diverse cell types (negative BM cells, host cells after transplant) assuming that train and test data derive from the same underlying data distribution. However, output probabilities are more or less interpreted as similarity scores (i.e. pDC-like). Given the assumption that the cell type specific gene expression patterns are invariant on the condition, can the authors comment on the choice to use the classifier over a supervised classification informed by cell-type specific gene markers? Did the authors evaluate other classification approaches over the RF?

Thank you for this important comment. We have added a more detailed description of our procedures to the Methods section and released Github scripts (3_RandomForest.R on https://github.com/petervangalen/Single-cell_BPDCN/). We recently compared four computational methods for reference-guided cell type classification from scRNA-seq of normal and malignant bone marrow samples²². We tested the following methods: Random Forest, cellHarmony, Seurat TransferData, and scPred, and found high concordance between the four methods. We agree with the reviewer that a supervised classification of patient cells informed by cell-type specific marker genes would be a valid alternative strategy, and highlight our evaluation of marker genes in random forest-based cell type predictions (EDF 5b). One feature of the Random Forest classification is that it provides a matrix of class probabilities when choosing *prediction type* = "prob", which allows visualizations such as Fig. 3f and EDF 7d. Given these considerations, we believe the random forest approach is appropriate.

R3Q6. 6. In relation to Figure 2b in the methods the authors mention that they present projection of patient cells into the UMAP of healthy donor cells was done by plotting each patient cell at the coordinates of the normal cell with the highest prediction score correlation. Therefore the cells shown are from the healthy donors and not of patients 9, 10.

We apologize this was not clearly stated. We have edited the figure legend to clarify that the cells are projected by transcriptional similarity to healthy donor cells:

“Figure 2b. UMAP shows density of bone marrow cells from healthy donors (n=6) and BPDCN patients (n=11) projected by transcriptional similarity to cells defined in healthy donors from 2a, and colored by two-dimensional kernel density estimation.”

R3Q7. 7. Figure 3c might benefit from multi-sample clonality analysis.

We thank the Reviewer for this suggestion. Manual construction of tumor phylogenies was relatively straightforward for the patients in the cited figure (now Fig. 1e, extended to five patients in the revised submission), due to the clonal nature and high VAF of shared mutations in most samples (data for Patient 10 shown Response Fig. 2 below). Based on the Reviewer’s suggestion, we performed a multi-sample clonality analysis (sciClone, <https://github.com/genome/sciclone>)²³ to confirm the clonal relationships in this patient. Results validated the phylogenies we determined manually.

Response Figure 2 | left, Variant allele frequencies of all variants detected in five samples from Patient 10. Mutations are grouped based on which samples they are detected in. **right**, Venn diagram highlighting the overlap in mutations between samples, using the same colors.

R3Q8. 8. The authors use mutation signatures as barcodes to elegantly demonstrate that BPDCN tumors arise from a CH-derived clone in the skin, which accumulates UV-induced

DNA damage during malignant progression. This analysis is further used to evaluate whether the cells that initiate disease relapse in the bone marrow and skin were there prior to initial therapy and cell stem transplantation. It is not clear however, whether the exposure to UV light provides sufficient evidence of a “tissue specific selective pressure” that shapes evolution of pre-malignant clones to cancer. Perhaps the authors can elaborate on this in their discussion.

We thank the Reviewer for their description of this work as elegant. The Reviewer identifies a critical conclusion in this manuscript that we have now better supported with additional data. We have spent considerable time adding experiments in this area and updating the text, including identifying caveats and areas for future work as the Reviewer suggests. Please see similar questions from Reviewers #1 and #2 and the appended responses.

To summarize our new data and thought process on this point, we provide additional data that establishes the order of acquisition of mutations as founder (present in premalignant clonal marrow that is not involved by BPDCN), then clonal high VAF UV mutation acquisition, then progression (e.g., *RAS*, *CDKN2A* loss, 3p/*SETD2* loss). Ordering of mutations in this granular fashion was not present in the original manuscript. However, this order of events alone does not prove that UV mutations “shape” the malignancy. Could UV mutations simply be a marker of transit through the skin? The high variant allele fraction and clonal nature of individual UV-type mutations suggests that a single premalignant cell, likely of the pDC lineage and harboring founder mutations, arrives in the skin, then acquires a UV DNA damage signature, and only then acquires progression mutations and the fully malignant phenotype. This alone we think is an important finding. Nonetheless, in response to questions about UV influencing evolution, we also pursued a functional experiment.

We now provide experimental data in a dendritic cell differentiation system showing that wild-type pDCs are very sensitive to UV radiation-induced cell death (compared to other DC and myeloid populations), but that *TET2* mutation offers a relative survival advantage specifically in pDCs in the setting of UV exposure (Fig. 4f-i, see above in answer to R3Q1). This is coupled with published data and our own findings that *TET2* mutation can expand pDCs relative to other DC subsets. Together with the clinical epidemiology of BPDCN, which often arises in a patient already diagnosed with CMML or MDS (diseases enriched for *TET2* mutations), these data suggest that certain CH mutations may increase the propensity of a pDC lineage cell to transform to BPDCN.

We have also performed a new clinical analysis of the initial skin lesion location in BPDCN patients and found a striking enrichment for the first presentation to be a single lesion at a sun-exposed site (presented in Fig. 3g and below for convenience). This is in contrast to other myeloid malignancies including AML involving the skin (“leukemia cutis”) that present on any area of skin and often in multiple lesions

simultaneously, not enriched for sun exposed sites. These data connect the UV DNA mutational signature to patient clinical presentation and to our new mechanistic data for *TET2* mutated pDCs having a selective advantage in the setting of UV radiation. Given these new data in the revised manuscript, we think our conclusion that UV shapes the evolution of BPDCN is sufficiently supported.

Figure 3 | g, Schematic shows the location of index BPDCN skin lesions (left), progression BPDCN skin lesions (middle), and acute myeloid leukemia (AML) skin lesions (right). Grey shading indicates areas of chronic or intermittent UV exposure. Representative clinical photos are shown.

R3Q9. 9. The evaluation of scRNA seq data as a potential mechanism to detect rare skin-derived circulating tumor cells early is intriguing. How generalizable was this observation from the sc-RNA seq data? Did the authors evaluate samples from other patients? Was the identification of cells in patient 10 related to the patients BM involvement? The authors should highlight that the detection of gene-expression signatures could be further explored, however the validation of UV-induced mutations required a priori knowledge of the mutations from the diagnostic specimen and would thus be less useful clinically.

In the revised manuscript, we have expanded the single-cell sequencing dataset from 6 patient samples to 17 patient samples (six healthy controls, five samples in which clinical evaluation did not show bone marrow involvement, and six samples with bone marrow involvement; Supplementary Table 3a). This enabled us to generate a gene expression signature of malignant BPDCN cells, and apply this signature to identify rare putative malignant cells in patients without marrow involvement. Using this revised analysis, we show tumor signature scores in putative malignant cells in 3/5 samples “without marrow involvement”: 2/4,593 cells (0.04%) from Patient 9, 19/10,106 cells (0.19%) from Patient 10, and 2/6,862 (0.03%) from Patient 12). Furthermore, in Patients 10 and 12, we detected progression mutations (including UV signature mutations) in the rare putative malignant cells using XV-seq, lending support to their malignant origin and our overall model. These findings are illustrated in Fig. 2g-h (shown below) and EDF 8c-e. Nonetheless, we agree with the Reviewer that with small numbers of patients and detected cells, we should not make too broad of claims and have removed references to potential clinical use of this technology. And the Reviewer is also correct that tumor-associated mutations must be known a priori to facilitate XV-seq, which we clarify in the methods and in the text:

“We next sought to map mutations that were identified a priori by our phylogenomic analysis onto hematopoietic differentiation hierarchies.”

Regarding the further exploration of gene expression, based on the signature of malignant BPDCN cells and founder/progression mutations, we were able to compare normal, premalignant, and malignant pDCs (EDF 7e-g). In premalignant pDCs, this showed upregulation of *TCL1A*, a signature BPDCN transcription factor, and downregulation of chemokine receptors (e.g., *CXCR4*) involved in bone marrow homing/retention (EDF 8a-b).

Figure 2 | g, Heatmap shows expression of BPDCN signature genes (rows, n=45) that contrast pDCs from six healthy donors (left, n=203 cells) and malignant BPDCN cells from six samples with bone marrow involvement (right, n=14,209). Malignant cells were downsampled to 30 cells per sample with genotyping information, if available. Top annotation bars indicate sample (colors as in panel f) and BPDCN signature score. Bottom heatmaps indicate founder and progression mutations analyzed by XV-seq in four of the six patient samples. **h**, Heatmaps shows expression of BPDCN signature genes in premalignant pDCs (left, n=91/495 cells with genotyping information shown) and malignant BPDCN cells (right, n=23) from samples without bone marrow involvement. Bottom heatmaps indicate founder mutations in both cell populations and progression mutations restricted to malignant BPDCN cells.

Referee #4 (Remarks to the Author):

The authors demonstrate that BPDCN patients with skin involvement exhibit clonal hematopoiesis in their bone marrow and use this opportunity to evaluate premalignant to malignant evolution across different anatomic sites.

They use the natural history of BPDCN in the marrow and the skin to establish that tissue-specific selective pressures can shape the evolution of premalignant clones. They also illustrate the role of ultraviolet (UV) light-induced DNA damages acquired in the cutaneous site on some deleterious evolution in advanced diseases.

They clearly explain the crucial role of clonal hematopoiesis as a model of cancer development that can be applied to various pathologic situations in oncologic and non-oncologic diseases. Authors also highlighted the importance of the local tissue-specific selective pressures on the pathologic development on the cellular level.

This work is original because BPDCN is a rare disease which is now well described but the use of deep sequencing of eXpressed Variants (XVseq) in marrow and skin tissues offers a unique opportunity to describe and analyse all the malignant process. Analyses and description are very pertinent and particularly well described.

The presentation of data and the methodology description is complete and appropriately presented in extended and supplemental data sets. The statistical analyses are extensively described without any specific problem.

Authors described and discuss their hypotheses with extensive and appropriate references. The authors have already published a lot of original data on this rare disease that are exploited here.

The manuscript is well constructed with a very clear abstract that summarized key features of the study that are also clearly explained in the introduction and conclusion section.

This work largely merits publication. Due to the high quality of this manuscript I think it can be published with no further revision.

Prof. Eric DECONINCK

Thank you for these very kind words and the extremely positive and enthusiastic evaluation of our work.

References

1. Jaiswal, S. *et al.* Age-related clonal hematopoiesis associated with adverse outcomes. *N. Engl. J. Med.* **371**, 2488–2498 (2014).
2. Genovese, G. *et al.* Clonal hematopoiesis and blood-cancer risk inferred from blood DNA sequence. *N. Engl. J. Med.* **371**, 2477–2487 (2014).
3. Xie, M. *et al.* Age-related mutations associated with clonal hematopoietic expansion and malignancies. *Nat. Med.* **20**, 1472–1478 (2014).
4. Cohen Aubart, F. *et al.* High frequency of clonal hematopoiesis in Erdheim-Chester disease. *Blood* (2020) doi:10.1182/blood.2020005101.
5. Kluk, M. J. *et al.* Validation and Implementation of a Custom Next-Generation Sequencing Clinical Assay for Hematologic Malignancies. *J. Mol. Diagn.* **18**, 507–515 (2016).
6. Garcia, E. P. *et al.* Validation of OncoPanel: A Targeted Next-Generation Sequencing Assay for the Detection of Somatic Variants in Cancer. *Arch. Pathol. Lab. Med.* **141**, 751–758 (2017).
7. Sholl, L. M. *et al.* Institutional implementation of clinical tumor profiling on an unselected cancer population. *JCI Insight* **1**, e87062 (2016).
8. Ostrander, E. L. *et al.* Divergent Effects of Dnmt3a and Tet2 Mutations on Hematopoietic Progenitor Cell Fitness. *Stem Cell Reports* **14**, 551–560 (2020).
9. Togami, K. *et al.* Sex-Biased ZRSR2 Mutations in Myeloid Malignancies Impair Plasmacytoid Dendritic Cell Activation and Apoptosis. *Cancer Discov.* (2021) doi:10.1158/2159-8290.CD-20-1513.
10. Izzo, F. *et al.* DNA methylation disruption reshapes the hematopoietic differentiation landscape. *Nat. Genet.* **52**, 378–387 (2020).
11. Nam, A. S. *et al.* Single-cell multi-omics of human clonal hematopoiesis reveals that DNMT3A R882 mutations perturb early progenitor states through selective hypomethylation. *Nat. Genet.* **54**, 1514–1526 (2022).
12. Van Egeren, D. *et al.* Reconstructing the Lineage Histories and Differentiation Trajectories of Individual Cancer Cells in Myeloproliferative Neoplasms. *Cell Stem Cell* **28**, 514–523.e9 (2021).
13. van Galen, P. *et al.* Single-Cell RNA-Seq Reveals AML Hierarchies Relevant to Disease Progression and Immunity. *Cell* **176**, 1265–1281.e24 (2019).
14. Nam, A. S. *et al.* Somatic mutations and cell identity linked by Genotyping of Transcriptomes. *Nature* **571**, 355–360 (2019).
15. Batta, K. *et al.* Divergent clonal evolution of blastic plasmacytoid dendritic cell neoplasm and chronic myelomonocytic leukemia from a shared TET2-mutated origin. *Leukemia* **35**, 3299–3303 (2021).
16. Mar, B. G. *et al.* SETD2 alterations impair DNA damage recognition and lead to resistance to chemotherapy in leukemia. *Blood* **130**, 2631–2641 (2017).
17. Carrasco Salas, P. *et al.* The role of CDKN2A/B deletions in pediatric acute lymphoblastic leukemia. *Pediatr. Hematol. Oncol.* **33**, 415–422 (2016).
18. Hock, H. & Shimamura, A. ETV6 in hematopoiesis and leukemia predisposition. *Semin. Hematol.* **54**, 98–104 (2017).

19. Hodis, E. *et al.* A landscape of driver mutations in melanoma. *Cell* **150**, 251–263 (2012).
20. Machado, H. E. *et al.* Diverse mutational landscapes in human lymphocytes. *Nature* **608**, 724–732 (2022).
21. Brady, S. W. *et al.* The genomic landscape of pediatric acute lymphoblastic leukemia. *Nat. Genet.* **54**, 1376–1389 (2022).
22. DePasquale, E. A. K. *et al.* Single-Cell Multiomics Reveals Clonal T-Cell Expansions and Exhaustion in Blastic Plasmacytoid Dendritic Cell Neoplasm. *Front. Immunol.* **13**, 809414 (2022).
23. Miller, C. A. *et al.* SciClone: inferring clonal architecture and tracking the spatial and temporal patterns of tumor evolution. *PLoS Comput. Biol.* **10**, e1003665 (2014).

Reviewer Reports on the First Revision:

Referees' comments:

Referee #1 (Remarks to the Author):

Thank you for the opportunity to review this revised manuscript by Griffin and colleagues. I find the revision responsive, and that the claims are better supported by data. This was achieved by both limiting the scope of the text to the investigation carried, as well as additional valuable data. In particular, I appreciate the increased focus on the UV mutational aspect as a key discovery in this paper.

I would like to suggest a couple of discussion points to potentially increase the appeal to a broader audience.

- The authors observe less mutation burden than typical skin cancers. Can the authors speculate as to why that could be? Is this the location of pDC deeper in the skin? Or perhaps the shorter time pDCs spend in the skin prior to transformation?
- Related question with potential translational impact, would the find of UV signature suggest that these cancers may potentially benefit from CBI (higher neoantigen load)?
- Can the authors speculate as to why TET2 loss would be protective from UV damage/cell death? Or at least highlight that this is a key question for future studies?

Referee #3 (Remarks to the Author):

The authors present a largely improved manuscript. The content is measured to the data provided, and further experimental data were generated to support some of the key observations in the manuscript. The data presented elegantly track the evolutionary history of BPDCN, and the study incorporates thorough genomic and single cell profiling of the corresponding cohort to evidence the proposed phylogenetic trajectories.

Some minor comments:

The title of the manuscript remains general, and could benefit by a more explicit statement relating to BPDCN.

Paragraph 1 Introduction needs to include relevant citations on the section regarding "clonally expanded precursors harboring pre-leukemic mutations giving rise to differentiated immune populations that circulate throughout the body and the relationship with inflammation and pathology".

How did mutation VAF's compare in patients with BM involvement? From Figure 1b it is not clear

whether the VAF distributions between the two subsets are significantly different. For this analysis the authors should account and correct for loss of heterozygosity at the TET2 locus, and cancer cell fraction.

In the single cell analysis section, the authors should acknowledge that the number of cells supporting the presence of rare pDCs in patients without marrow involvement are limited and warrant further validation.

Clusters in Figure 2F are hard to discern - consider revising figure, and adjunct text.

Last the authors should acknowledge that whilst UV selection is a potential model, the data in the study do not definitively demonstrate that this is the case.

Referee #5 (Remarks to the Author):

Griffin et al investigate the pathogenesis of BPDCN, a rare but aggressive hematological malignancy of the myeloid lineage.

They study 16 cases of BPDCN, 9 without and 7 with bone marrow involvement (including multiple relapse samples), using WGS, WES, targeted gene sequencing, single cell transcriptomics +/- paired mutation calling, phylogenetic and mutational signature analyses. They present evidence that BPDCNs develop from premalignant clonal expansions akin to clonal hematopoiesis (CH) or clonal cytopenia of unknown significance (CCUS), which seed skin with mutation-bearing plasmacytoid dendritic cells (pDCs) (a process that also occurs physiologically with pDCs derived from normal hematopoietic stem cells). These premalignant pDCs are then driven to transformation through the acquisition of additional oncogenic mutations, many of which are not commonly seen in other myeloid malignancies such as AML.

The authors also provide experimental evidence using HoxB8-immortalised mouse hematopoietic progenitors that TET2 mutations, very common in BPDCN, protect pDCs from UV light-induced cell death, proposing that this may be a key conduit in the clonal evolution of these aggressive malignancies.

The study, which is well conducted and presented, gives important new insights into the pathogenesis of BPDCNs. Despite their rarity, these cancers are well recognized and the work will be of interest to Nature's readership.

Reviewers' comments to the initial submission were insightful and helpful. The authors have addressed these well and as a result the manuscript appears much more robust/comprehensive and provides support for many of the authors' findings/claims.

With regards to novelty, this primarily relates to the identification of a requirement for two distinct geographies/environments (bone marrow and skin) in the evolution of BPDCN, with UV-mutagenesis and TET2-loss playing critical roles in this process.

Below are some comments/suggestions aiming to help further improve the manuscript:

1. I agree with Reviewer 1 that the absence of the term BPDCN from the title is perplexing. In fact the authors provide data that leukemia cutis (i.e. skin AML) differs to BPCND in terms of its skin distribution and mutational cargo. This makes the current title “Ultraviolet radiation shapes leukemia transformation in the skin” potentially confusing. A different title should be considered.

2. Linking the development of BPDCN to cancer dissemination more generally (“seed and soil” etc) may not be appropriate. The authors report a process that exploits a physiological mechanism of pDC dissemination to the skin, where they can be subjected to a mutagenic process (UV) that does not commonly affect blood cells resident in hematopoietic tissues. Cancer dissemination/metastases relies on a process of spread and adaptation to a new environment/site, with the cancer cell moving to a site where it does not normally reside in and subsequently adapting to this new environment. This is not to say that the process described here has no broader relevance - in my view it reveals how evolving cancer cells can harness any mechanism that facilitates their evolution/transformation, including an unusual mutagenic process (e.g. equivalent to what happens with chemotherapy-induced mutations driving treatment-related myeloid neoplasms). In my view, emphasizing this rather than the dissemination argument would be more accurate.

3. The authors frame pDCs as the cells of origin of BPDCN. As normal pDCs are terminally-differentiated, it is possible/probable that the cell of origin of BPDCN is an abnormal pDC-like cell or progenitor. It would be helpful for readers if this could be discussed.

4. The experimental data using HOXB8 cells provides a potential basis for the role of TET2 bi-allelic loss in the genesis of BPDCN, i.e. that it confers resistance to UV light and so enables UV-exposed pDCs to survive and “benefit” from the UV-mediated mutations. However, epigenetic rewiring mediated by TET2-loss in the context of UV may itself contribute to the malignant phenotype (e.g. by downregulating apoptotic or other pathways). The recent paper by Jain et al (Nature 2023) on TET2’s role in CAR-T cell behavior, is an example of how TET2 loss can drive a proliferative state in association with reduced effector function and acquisition of subsequent somatic mutations. This could also be discussed.

Author Rebuttals to First Revision:

Response to Reviewers

Referee #1 (Remarks to the Author):

Thank you for the opportunity to review this revised manuscript by Griffin and colleagues. I find the revision responsive, and that the claims are better supported by data. This was achieved by both limiting the scope of the text to the investigation carried, as well as additional valuable data. In particular, I appreciate the increased focus on the UV mutational aspect as a key discovery in this paper.

I would like to suggest a couple of discussion points to potentially increase the appeal to a broader audience.

- The authors observe less mutation burden than typical skin cancers. Can the authors speculate as to why that could be? Is this the location of pDC deeper in the skin? Or perhaps the shorter time pDCs spend in the skin prior to transformation?

We note in the discussion that pDCs are not found in large numbers in normal skin, but that they can be recruited in the setting of inflammation. We have added text to results and discussion highlighting important questions for future research that broadly encompasses these important mechanistic questions proposed by the reviewer.

- Related question with potential translational impact, would the find of UV signature suggest that these cancers may potentially benefit from CBI (higher neoantigen load)?

This is a very interesting point and a question that we have also considered. We added this question to the future directions statement in the discussion.

- Can the authors speculate as to why TET2 loss would be protective from UV damage/cell death? Or at least highlight that this is a key question for future studies?

We agree and have highlighted this in the discussion as a key question for future studies.

Referee #3 (Remarks to the Author):

The authors present a largely improved manuscript. The content is measured to the data provided, and further experimental data were generated to support some of the key observations in the manuscript. The data presented elegantly track the evolutionary history of BPDCN, and the study incorporates thorough genomic and single cell profiling of the corresponding cohort to evidence the proposed phylogenetic trajectories.

Some minor comments:

The title of the manuscript remains general, and could benefit by a more explicit statement relating to BPDCN.

We agree and have changed the title to be more specific: "Ultraviolet radiation shapes dendritic cell leukemia transformation in the skin."

Paragraph 1 Introduction needs to include relevant citations on the section regarding "clonally expanded precursors harboring pre-leukemic mutations giving rise to differentiated immune populations that circulate throughout the body and the relationship with inflammation and pathology".

We added references to the abstract/introductory paragraph including several to support this statement.

How did mutation VAF's compare in patients with BM involvement? From Figure 1b it is not clear whether the VAF distributions between the two subsets are significantly different. For this analysis the authors should account and correct for loss of heterozygosity at the TET2 locus, and cancer cell fraction.

This is an important point, and we thank the reviewer for suggesting we clarify it. Our intent in Figure 1b was to show that founder-type mutations (e.g., in *TET2* and *ASXL1*) do not have different VAF between patients with and without overt bone marrow involvement. This supports the conclusions that these founder mutations are distributed throughout hematopoietic compartments and not restricted to the pDC lineage. We have now performed a statistical test, as suggested by the reviewer, to emphasize that the VAFs are similar between the two groups. We also corrected for LOH as suggested by the reviewer. We added this to the text associated with Figure 1b and provide the primary data underlying this calculation in a Primary Data File.

In the single cell analysis section, the authors should acknowledge that the number of cells supporting the presence of rare pDCs in patients without marrow involvement are limited and warrant further validation.

We added a sentence to the single cell analysis section acknowledging that these cells are rare and that their clinical relevance requires further investigation.

Clusters in Figure 2F are hard to discern - consider revising figure, and adjunct text.

We agree with the reviewer. We moved this former Figure 2f to an extended data figure, and we also added annotations to the UMAP clusters to provide more clarity.

Last the authors should acknowledge that whilst UV selection is a potential model, the data in the study do not definitively demonstrate that this is the case.

We added this caveat to the discussion section and characterize our model as one possible mechanism of BPDCN pathogenesis that requires additional validation.

Referee #5 (Remarks to the Author):

Griffin et al investigate the pathogenesis of BPDCN, a rare but aggressive hematological malignancy of the myeloid lineage.

They study 16 cases of BPDCN, 9 without and 7 with bone marrow involvement (including multiple relapse samples), using WGS, WES, targeted gene sequencing, single cell transcriptomics +/- paired mutation calling, phylogenetic and mutational signature analyses. They present evidence that BPDCNs develop from premalignant clonal expansions akin to clonal hematopoiesis (CH) or clonal cytopenia of unknown significance (CCUS), which seed skin with mutation-bearing plasmacytoid dendritic cells (pDCs) (a process that also occurs physiologically with pDCs derived from normal hematopoietic stem cells). These premalignant pDCs are then driven to transformation through the acquisition of additional oncogenic mutations, many of which are not commonly seen in other myeloid malignancies such as AML.

The authors also provide experimental evidence using HoxB8-immortalised mouse hematopoietic progenitors that TET2 mutations, very common in BPDCN, protect pDCs from UV light-induced cell death, proposing that this may be a key conduit in the clonal evolution of these aggressive malignancies.

The study, which is well conducted and presented, gives important new insights into the pathogenesis of BPDCNs. Despite their rarity, these cancers are well recognized and the work will be of interest to Nature's readership.

Reviewers' comments to the initial submission were insightful and helpful. The authors have addressed these well and as a result the manuscript appears much more robust/comprehensive and provides support for many of the authors' findings/claims.

With regards to novelty, this primarily relates to the identification of a requirement for two distinct geographies/environments (bone marrow and skin) in the evolution of BPDCN, with UV-mutagenesis and TET2-loss playing critical roles in this process.

Below are some comments/suggestions aiming to help further improve the manuscript:

1. I agree with Reviewer 1 that the absence of the term BPDCN from the title is perplexing. In fact the authors provide data that leukemia cutis (i.e. skin AML) differs to BPCND in terms of its skin distribution and mutational cargo. This makes the current title “Ultraviolet radiation shapes leukemia transformation in the skin” potentially confusing. A different title should be considered.

We agree and have changed the title to be more specific: “Ultraviolet radiation shapes dendritic cell leukemia transformation in the skin.”

2. Linking the development of BPDCN to cancer dissemination more generally (“seed and soil” etc) may not be appropriate. The authors report a process that exploits a physiological mechanism of pDC dissemination to the skin, where they can be subjected to a mutagenic process (UV) that does not commonly affect blood cells resident in hematopoietic tissues. Cancer dissemination/metastases relies on a process of spread and adaptation to a new environment/site, with the cancer cell moving to a site where it does not normally reside in and subsequently adapting to this new environment.

This is not to say that the process described here has no broader relevance - in my view it reveals how evolving cancer cells can harness any mechanism that facilitates their evolution/transformation, including an unusual mutagenic process (e.g. equivalent to what happens with chemotherapy-induced mutations driving treatment-related myeloid neoplasms). In my view, emphasizing this rather than the dissemination argument would be more accurate.

We appreciate these thoughtful comments and agree that our data are most in keeping with a theme of cancer cells harnessing mechanisms that facilitate their evolution. We removed the specific references to “seed and soil” and further emphasized the evolutionary aspect of the model throughout the manuscript.

3. The authors frame pDCs as the cells of origin of BPDCN. As normal pDCs are terminally-differentiated, it is possible/probable that the cell of origin of BPDCN is an abnormal pDC-like cell or progenitor. It would be helpful for readers if this could be discussed.

We changed the language in several places in the manuscript to say pDC or pDC-like cell, or pDC or pDC precursor to avoid implying that a terminally differentiated pDC is the only possible cell of origin. We agree that the exact cellular target for transformation – suggested here as likely a

dendritic lineage committed or restricted cell but remaining incompletely defined - is an important topic for future work.

4. The experimental data using HOXB8 cells provides a potential basis for the role of TET2 bi-allelic loss in the genesis of BPDCN, i.e. that it confers resistance to UV light and so enables UV-exposed pDCs to survive and “benefit” from the UV-mediated mutations. However, epigenetic rewiring mediated by TET2-loss in the context of UV may itself contribute to the malignant phenotype (e.g. by downregulating apoptotic or other pathways). The recent paper by Jain et al (Nature 2023) on TET2’s role in CAR-T cell behavior, is an example of how TET2 loss can drive a proliferative state in association with reduced effector function and acquisition of subsequent somatic mutations. This could also be discussed.

We agree that the mechanism of how TET2 loss creates a selective advantage in the setting of UV is an important topic for future work. We added a statement to the discussion to emphasize that importance.